# A genus-wide interaction atlas across NS4B orthologues identifies a conserved role for UFMylation in orthoflavivirus replication

Sreejith Rajasekharan[1], Viviana Andrea Barragan Torres [2], Yago Cortes Pinheiro Gomes[2], Lucas Wilken [1], Katharina Overhoff[1], Yen-Chia Lin[1], Shunmoogum A. Patten [2,3], Satoru Watanabe[4], Laurent Chatel-Chaix [2,3,5] & Pietro Scaturro [1,6] ✉

Orthoflavivirus infections represent an increasing public health burden, with several members of the genus emerging or re-emerging globally. Several pre-clinical studies identified the non-structural protein 4B (NS4B), as the most promising target for the development of potent direct-acting antivirals. However, its functional roles in viral replication are still elusive. Here, we employ an integrated proteomic approach to systematically identify cellular targets of NS4B across eight prototypic orthoflaviviruses and characterize their influence on the human proteome. Using this approach, we mapped high-confidence NS4B-interacting human proteins across the genus, underlying potentially divergent and convergent mechanisms of host adaptation across orthoflaviviruses spanning diverse pathologies and vector preferences. Among these, we unveil a novel function for UBA5, the E1-activating enzyme of the UFMylation pathway, in orthoflavivirus replication. Mechanistically, we map associations of distinct viral proteins with multiple members of the UFMylation pathway, which are selectively recruited to sites of viral replication to promote infectious particle production. Finally, we demonstrate that pharmacological inhibition of UFMylation exerts potent antiviral activity in vitro and in vivo. This integrative study provides a rational framework for a system-level understanding of orthoflavivirus NS4B effector functions and sheds light on a conserved and unconventional role for UFMylation in ortho-flavivirus replication.

The genus *Orthoflavivirus* includes several clinically relevant emerging and re-emerging arboviruses, such as dengue virus (DENV), Zika virus (ZIKV), Japanese encephalitis virus (JEV), yellow fever virus (YFV), Powassan virus (POWV), Usutu virus (USUV), tick-borne encephalitis virus (TBEV) and West Nile virus (WNV)[1]. Orthoflaviviruses are enveloped viruses with a positive-sense single-stranded RNA genome encoding for a polyprotein that is co- and post-translationally processed into three structural (capsid, pre-membrane, envelope) and seven nonstructural proteins (NS1, NS2A, NS2B, NS3, NS4A, NS4B, NS5)[2]. During infection, orthoflaviviruses hijack and rewire multiple

[1]Leibniz Institute of Virology, Hamburg, Germany. [2]Centre Armand-Frappier Santé Biotechnologie, Institut National de la Recherche Scientifique, Laval, Québec, Canada. [3]Center of Excellence in Orphan Diseases Research-Fondation Courtois (CERMO-FC), Québec, Canada. [4]Programme in Emerging Infectious Diseases, Duke-NUS Medical School, Singapore, Singapore. [5]Swine and Poultry Infectious Diseases Research Centre (CRIPA), Québec, Canada. [6]German Centre for Infection Research (DZIF), Partner Site Hamburg-Lübeck-Borstel-Riems, Hamburg, Germany. ✉e-mail: pietro.scaturro@leibniz-liv.de

host machineries to create a microenvironment conducive to viral RNA replication, including the biogenesis of replication organelles (ROs) within the endoplasmic reticulum (ER), where genome replication takes place[2]. The mature non-structural protein 4B (NS4B) is a small integral transmembrane protein that plays an essential, yet poorly characterized, role in viral replication via interactions with viral and host proteins[3]. NS4B is involved in the remodeling of the ER[4], biogenesis of the viral ROs[4–6], and modulation of mitochondria morphodynamics[7].

Recently, NS4B has attracted renewed interest as small molecule inhibitors targeting its interactions with the viral protease/helicase NS3 have demonstrated highly potent antiviral activity in mice and primate infection models, with some currently being evaluated in human clinical trials[8–11]. The NS4B–host interactome has been previously characterized for individual viral species (DENV[7,12], WNV[13] and ZIKV[14]), identifying associations with cellular proteins involved in immune responses[15–17], mitochondria–ER contact sites (MERCs)[7,14], ER stress[15,18] and autophagy[18]. However, the high heterogeneity of these data with respect to cellular backgrounds, expression systems and proteomic pipelines, has precluded the mapping of conserved or distinct specificities among orthoflaviviruses, which could help elucidating the molecular mechanisms underlying the broad range of diseases caused by this important group of arboviruses. Furthermore, only a limited number of NS4B-interacting proteins have been functionally or phenotypically characterized to date[3]. Altogether, we still know surprisingly little on the specific cellular machineries usurped by orthoflavivirus NS4B, and the molecular mechanisms underlying species-specific dysfunctional programs.

Here, we leverage an integrative systems biology approach, combining affinity-purification coupled with mass spectrometry and global proteomic profiling, to identify human proteins targeted by the NS4B of eight (re-)emerging orthoflaviviruses. This pan-viral NS4B interaction atlas, hereafter referred to as NS4Bome, elucidates hitherto unreported host-binding specificities. In addition to highlighting cellular factors that are selectively targeted by each viral species, this genus-wide atlas further categorizes cellular processes and proteins consistently hijacked across orthoflaviviruses. Among these, we identify a conserved pro-viral role for UFMylation – an emerging post-translational modification – in orthoflavivirus replication via multipartite interactions between viral proteins and UFMylation pathway components. Importantly, we demonstrate that pharmacological inhibition of UFMylation exerts potent antiviral activity in vitro and in vivo, improving disease outcomes in developing animals.

Collectively, these results provide a rational framework for the functional characterization of the most versatile orthoflavivirus protein and demonstrate the effectiveness of this approach in streamlining target identification efforts for the rational design of a new class of broad-spectrum antiviral inhibitors.

## Results

### A genus-wide map of orthoflavivirus NS4B human targets identifies strain-, species-specific and pan-orthoflavivirus associations with the human proteome

We employed a multi-layered proteomic approach, combining affinity purification coupled with quantitative mass spectrometry (AP-MS) and global proteomics (Fig. 1a), to map the extended orthoflavivirus NS4Bome and identify cellular target whose abundance is modulated by persistent viral protein expression. To this end, the C-terminally HA-tagged 2k-NS4B of eight prototypic orthoflaviviruses (DENV, ZIKV, JEV, WNV, YFV, TBEV, POWV and USUV) were stably expressed in human choriocarcinoma trophoblast JEG-3 cells. To account for possible differences between historical and contemporary isolates, the NS4B of two DENV serotype 2 (DENV2) strains were included (Supplementary Fig. 1). JEG-3 cells expressing the related *Flaviviridae* hepatitis C virus (HCV) NS4B, ZIKV capsid, *Gaussia* luciferase or empty vector were

used as control for non-specific binding (Fig. 1a). Subcellular distribution and expression levels of HA-tagged NS4B were validated by immunofluorescence analysis, immunoblotting and analysis of intensity-based absolute quantification (iBAQ) values from the proteomic dataset, confirming comparable expression levels and a good separation of NS4B interactomes of individual species (Supplementary Fig. 1a–e). For each cell line, four replicates of affinity-purified complexes (NS4Bomes) and whole cell lysates (effectomes) were collected and analyzed by quantitative mass-spectrometry, identifying a total of 538 NS4B-interacting proteins and 514 host proteins differentially expressed upon stable NS4B expression (Figs. 1b, 2a and Supplementary Data 1, 2).

To prioritize reproducible, bait-specific protein-protein interactions (PPIs), we analyzed four replicates for each bait protein using very stringent cut-off criteria (log2 (fold-change) ≥ 5; two-sided Welch's t-test p-value ≤ 0.05; S0 = 4, FDR ≤ 0.01, $n = 250$). Collectively, this extended NS4Bome revealed interactions between 538 human proteins and 8 NS4B orthologues (Figs. 1b, 2a) with normalized intensities spanning 15 orders of magnitudes and normally distributed across the whole dynamic range of the human proteome (Supplementary Fig. 2a). The total number of PPIs displaying very high fold-enrichment levels (Log2(fold-difference vs. G.luc[HA] control) > 5) varied significantly across baits, with up to ten-fold difference between the lowest (DENV-NS4B, 43 interactors) and the highest (WNV-NS4B, 386 interactors) number of PPI observed for individual species (Fig. 2a). Notably, these differences did not correlate linearly with either the expression levels of the baits, or the total number of preys identified (Supplementary Fig. 1c, Fig. 2a), suggesting a certain degree of divergence of NS4B-host-interactions within the *Orthoflavivirus* genus.

The systematic nature of the orthoflavivirus NS4Bome allows for the first time the identification and classification of human cellular targets into a core group of NS4B interacting proteins (broadly conserved, highly enriched interacting proteins), an extended group orthoflavivirus targets (proteins within pathways broadly targeted by the majority of the viral species) and a distinctive group of proteins selectively enriched in individual or a small number of species (sub- or species-specific binding profiles).

### The core human interactome of orthoflavivirus NS4B

To define the core NS4Bome, we extracted from the extended NS4Bome proteins displaying very high enrichment levels (log2(fold-enrichment) > 5) and significantly bound by at least 6 out of 8 viral species (Fig. 2a, Supplementary Fig. 2b). In total we mapped 74 host proteins within this class of interactors, including factors previously reported to interact with individual orthoflavivirus NS4Bs, such as BSG[14], CHP1[14], STT3A[19], SEC61G[12], VDAC1/2[7] or TMEM41B[14,20], as well as novel host interactors, such as GPX8, UNC50 and STX17 (Supplementary Fig. 2c, d). Beside interactions with discrete cellular proteins, the core NS4Bome revealed enrichment of distinctive cellular processes, including the translocon complex (SEC61B, SSR1, SSR3), the very-long chain fatty acid biosynthetic pathway (HSD17-B1-11-12, DHCR2 and DHCR7, FADS2 and FAR1) as well as mitochondrial respiration (COX6A1, NDUFA11, ATP5L) and dynamics (FIS1, MFN2). Furthermore, the oligosaccharyltransferase complex and signal recognition particle–dependent co-translational targeting to membranes and lipid biosynthesis were significantly overrepresented pathways, highlighting an evolutionary conserved targeting of processes involved in protein synthesis and lipid metabolism (Supplementary Fig. 2b, c, d).

Collectively, these proteins likely support evolutionary conserved mechanisms of host-adaptation across the *Orthoflavivirus* genus, providing cues into the key mediators of NS4B functions and a list of valuable targets for the rational design of pan-viral host-targeting antivirals.

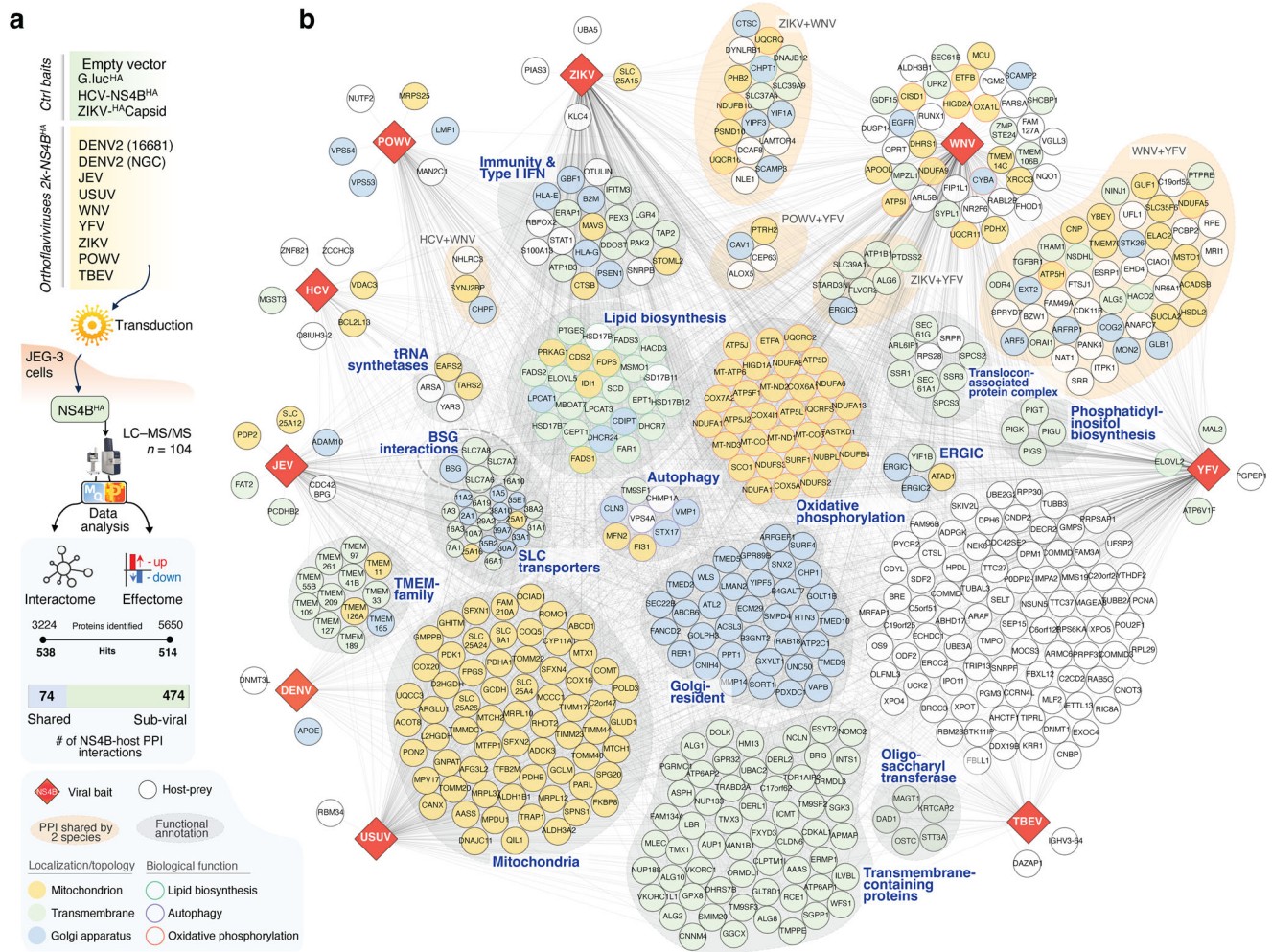

**Fig. 1 | A genus-wide atlas of the orthoflavivirus NS4B interactome. a** Systematic comparison of interactomes and host proteome changes (Effectome) of the orthoflavivirus NS4Bs and four control proteins (ZIKV capsid, *Gaussia* luciferase, HCV-NS4B and naïve JEG-3 cells), using eight orthologues of pathogenic orthoflaviviruses (DENV, YFV, ZIKV, JEV, WNV, USUV, TBEV, POWV). The numbers of unique and shared (significantly-enriched in at least 6 viral baits) host interactions across the orthoflavivirus NS4B proteins ("NS4Bome", Interactome) and significantly modulated proteins in NS4B-expressing cells (Effectome) is shown. **b** Combined virus–host protein–protein interaction network of orthoflavivirus

NS4Bs measured by AP–MS. Shared interacting proteins amongst two species are denoted by the orange shadows. Selected biological functions and processes are denoted in grey shades. Only high-confidence interactors are shown (Log2(Fold-change) ≥ 5; $p$ value ≤ 0.01). Interactions between viral and host proteins are indicated by grey lines. Colored circles and nodes represent a manually-curated selection of gene-ontology annotations and organellar distribution, respectively. ERGIC ER-Golgi intermediate compartment, SLC solute carrier; BSG basigin; TMEM transmembrane proteins; tRNA transfer RNA ($n$ = 4 biological replicates).

## The extended human interactome of orthoflavivirus NS4B

To identify pathways underlying functionally conserved mechanisms of host-adaptation across orthoflaviviruses, we additionally performed upset and gene ontology (GO) enrichment analysis on the extended NS4B PPI network. This approach revealed a higher number of shared interactions and targeted cellular pathways between WNV, YFV and ZIKV NS4B when compared to other members of the genus, suggesting a higher degree of functional conservation across these species (Fig. 2a). Furthermore, this analysis revealed significant enrichment of interactors associated with mitochondrial membrane organization and autophagy, consistent with the intricate relationship between orthoflaviviruses, the ER and mitochondria[7,21–23] (Fig. 2b, c; and Supplementary Data 1). Notably, proteins involved in mitochondrial membrane organization, mitochondrial respiration and respiratory electron transport were amongst the most enriched cellular targets of the global PPI network, suggesting a distinctive recruitment of processes involved in oxidative phosphorylation via multiple interactions with mitochondrial membrane proteins.

Additionally, the extended NS4Bome illuminated unexpected interactions of NS4B with pathways involved in intracellular protein transport (COPII-mediated vesicular transport) and intercellular interactions (basigin interactions), residing in ER-distal organelles including the Golgi apparatus and the trans-Golgi network (Fig. 2c, Supplementary Fig. 2d). Importantly, it consolidates and expands the interconnections between NS4B and mitochondria-resident processes[7,24], highlighting specific targeting of proteins involved in the mitochondrial respiratory chain.

## Host processes and complexes targeted by individual orthoflaviviral NS4B proteins

In addition to proteins and pathways broadly targeted by the majority of orthoflavivirus NS4B, the extended NS4Bome enabled the identification of species-specific PPIs displaying highly selective binding profiles, providing a rational framework for mechanistic and functional follow-up studies on individual species. Examples of virus-specific interactions include proteins involved in protein sorting at the Golgi

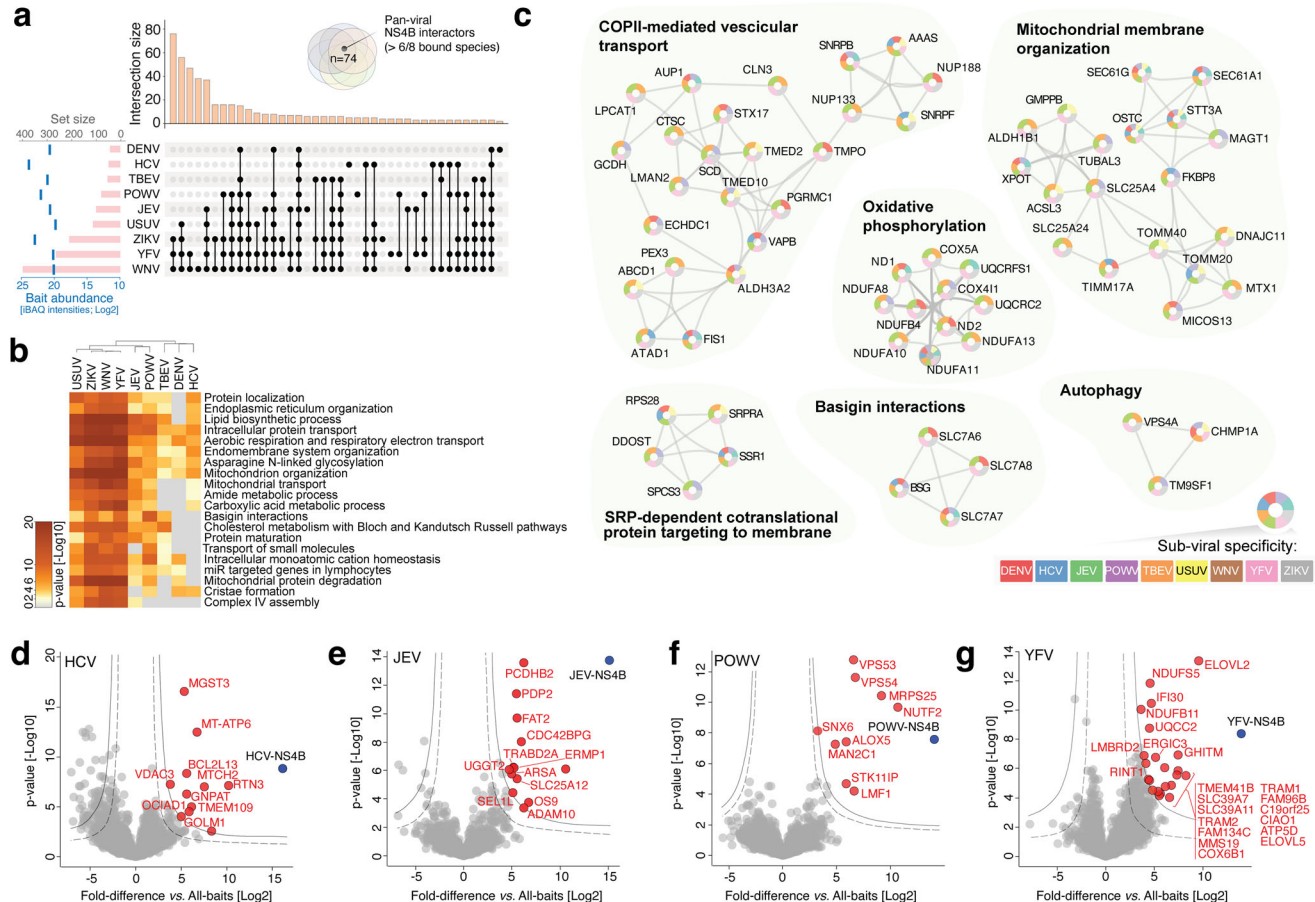

**Fig. 2 | Orthoflavivirus NS4Bs display pan- and sub-viral specificities. a** Upset plot of the orthoflavivirus NS4B-human interactome shows the total number of protein–protein interactions (set size) and the size of the intersecting pairs among the individual NS4B interactomes (intersection size). Every possible intersection of protein pairs is represented in the histograms. The median expression level across three to four biological replicates of each of the viral baits in the AP–MS dataset is overlayed on the set size ("Bait abundance"). The colored Venn diagram displays the "core" NS4B interactome of host proteins interacting with at least 6 out of 8 orthoflavivirus NS4Bs (Log2(Fold-change) ≥ 5; p-value ≤ 0.01). **b** Metascape enrichment analysis of all statistically significant host-interacting proteins from NS4Bome interaction networks shown in Fig. 1B. Top20 most significant terms are

shown. **c** Protein–protein Interaction (PPI) enrichment analysis to identify densely connected network components. Pathway and process enrichment analysis has been applied to each component independently, and the top 6 most enriched are shown. Colored pie-charts denote viral-bait specificity across orthoflaviviruses. **d–g** Volcano plots of selected species-specific NS4B host-interactors of HCV (d), JEV (e), POWV (f) and YFV (g). Significant interactors are determined using a permutation-based FDR and the resulting high-confidence (solid line) and low-confidence (dashed lines) thresholds are displayed in the plot. Significant high-confidence interactors selectively enriched in individual species when compared to all the others (Log2(Fold-change) vs. All-baits ≥ 5; p-value ≤ 0.01) are displayed in red, viral baits are displayed in blue (n = 4 biological replicates).

apparatus (VPS53 and VSP54, POWV), pyruvate dehydrogenase complex activity (PDP2, JEV) and ER membrane curvature (FAM134C, YFV) (Fig. 2d–g; and Supplementary Data 1). Interestingly, comparative analysis of PPIs between historical and contemporary strains of DENV2 (16681 and New Guinea C) also identified a number of isolate-specific PPIs, providing potential cues into differential host-responses and the replicative fitness observed in vitro[25], and validating a large proportion of host interactors shared by both isolates such as TMEM165, ATP5L and NDUFA11 (Supplementary Fig. 1e).

To chart virus–host interactions into effector functions of viral proteins, we next analyzed the global proteomes of JEG-3 cells persistently expressing individual NS4Bs ("effectome"; Fig. 1a; and Supplementary Data 2). This approach identified 514 host proteins whose relative abundance was significantly modulated by expression of at least one NS4B orthologue. Intersecting individual NS4B interactomes and their corresponding effectomes, highlighted a subgroup of NS4B interacting proteins displaying altered steady state expression levels, providing direct links between NS4B-interactions and down-regulation of selected host proteins (Supplementary Fig. 2e–m). These include

proteins previously associated with interferon (IFN) -dependent restriction of viral replication[26] (IFITM3; HCV and YFV), providing a mechanistic basis for active counteraction of innate immune responses. This approach also revealed several host proteins involved in the ER-associated degradation machinery (i.e., DERL2; USUV and C6orf120; ZIKV, YFV, WNV) and selective components of the electron transport chain (i.e., MT-ND3; ZIKV, YFV, WNV), highlighting novel host proteins bound by NS4B and specifically targeted for degradation (Supplementary Fig. 2e–m and Supplementary Data 2).

Altogether, the orthoflavivirus NS4Bome provides an accurate cartography of NS4B-associated effector functions, providing an evidence-based framework to categorize and characterize broadly conserved as well as species-specific virus–host PPIs involved in orthoflavivirus replication.

**Targeted RNAi-screening uncovers NS4B-interacting host factors impacting ZIKV replication**

To assess the functional relevance of the newly identified NS4B-interacting host factors in viral replication, we selected 58 interactors

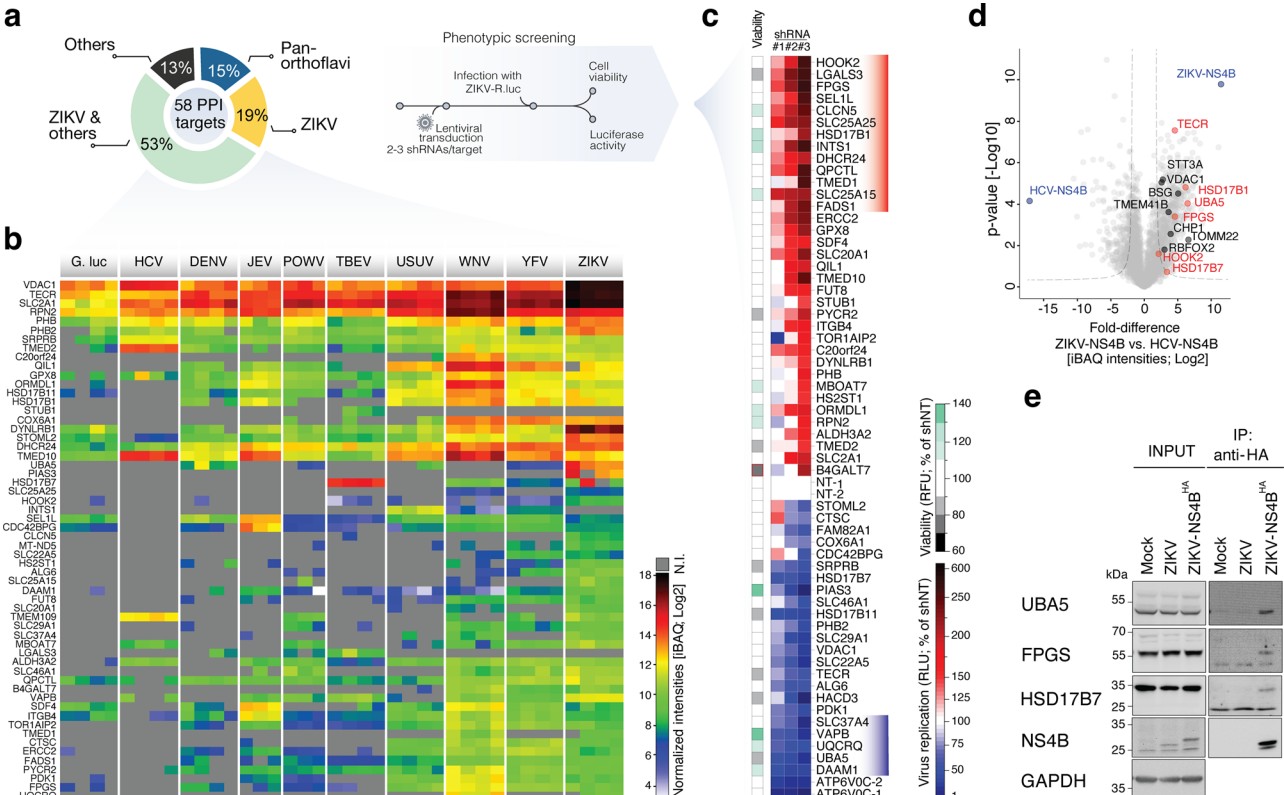

**Fig. 3 | Functional validation of orthoflavivirus NS4B host-interacting proteins identifies novel host-restriction and -dependency factors. a** Specificity of shortlisted NS4B interacting host proteins across NS4B orthologues selected for functional validation and experimental setup of the phenotypic screen to assess their functional relevance. **b** Intensity-based absolute quantification (iBAQ) of protein abundance of orthoflavivirus NS4B-interacting host proteins across baits. The profile of 58 host proteins selected for RNA interference (RNAi) screening are shown. (N.I. not identified). **c** RNAi screening of NS4B-binding proteins to identify host factors involved in ZIKV replication. JEG-3 cells were transduced with lenti-viruses encoding shRNAs targeting each of the 58 NS4B -interacting host proteins (2–3 shRNAs/gene), and infected with a full-length ZIKV reporter strain expressing *Renilla* luciferase at 72 hpt. The extent of virus replication was determined by luciferase activity at 48 hpi. The results of each biological replicate for one shRNA per gene are shown as a heatmap (relative to shNT; complete dataset in Supplementary Fig. 3). Newly identified host-restriction and -dependency factors are highlighted in red and blue, respectively (cut-off criteria: 50% difference in viral replication with two out of three shRNAs or >75% difference with one out of three shRNAs). Mean cell viability for each shRNA is also shown as a heatmap (cut-off criteria: cell viability ≥75% of shNTs; n = 3). B4GALT7 shRNA is slightly toxic (cell viability 74%, highlighted in red). **d** Volcano plots comparing interactors of ZIKV-NS4B and HCV-NS4B (two-sided Welch's t-test, log2(fold change) ≥ 2.5, FDR-corrected Welch's t-test $p ≤ 0.01$) (n = 4 biological replicates). A selected group of previously reported interactors of ZIKV-NS4B is shown in black. Newly identified host-interactors functionally- and orthogonally-validated in this study are shown in red. Viral baits are shown in blue. **e** Co-immunoprecipitation of ZIKV-NS4B–HA with endogenous host proteins. Cell lysates of JEG-3 cells infected with wild-type ZIKV (NT) or NS4B-HA-tagged ZIKV (ZIKV-NS4B[HA]) proteins were used for anti-HA immunoaffinity purification and probed with the indicated antibodies against newly-identified NS4B-interacting host proteins. Representative blots are shown (n = 3 independent experiments).

for further phenotypic characterization in loss-of-function analysis. Candidate host-proteins were selected from each of the categories described above, spanning species-, sub- and pan-viral specificities (Fig. 3a, b). ZIKV was selected for functional follow-up experiments because of the specific cellular background closely resembling one of the known tissues targeted in vivo (placenta). JEG-3 cells were transduced with individual short hairpin RNA (shRNA)-encoding lenti-viruses and infected with a ZIKV H/PF/2013 luciferase reporter virus[27] (Fig. 3a). This approach identified 14 anti-viral and 5 pro-viral genes whose silencing significantly enhanced or impaired ZIKV replication, respectively, in the absence of overt cytotoxicity (Fig. 3c, red and blue shades; Supplementary Fig. 3a, b and Supplementary Data 3). Inter-estingly, these newly identified host dependency and host restriction factors include ZIKV-NS4B–interacting proteins displaying both higher (i.e., UBA5 and HSD17B1) and lower (HOOK2 and HSD17B7) enrich-ments than previously described and validated interactors (Fig. 3d, red and black circles, respectively), corroborating the value of the curated NS4Bome in detecting a wide-range of functionally relevant associa-tions. Co-immunoprecipitation followed by western blotting (co-IP) confirmed that multiple proteins specifically associate with several

orthoflavivirus NS4Bs (HOOK2, TECR, HSD17B1, HACD3; Supplemen-tary Fig. 3c), largely mirroring AP-MS data (Fig. 3b). To further validate NS4B interactions in the context of productive viral infection, we engineered a ZIKV infectious clone carrying an internal HA-tag within NS4B (ZIKV-NS4B[HA]) (Supplementary Fig. 4a). We confirmed that the insertion of the HA-tag only slightly reduced replication levels (Sup-plementary Fig. 4b), and validated epitope tag stability and its co-localization with NS4B in infected cells (Supplementary Fig. 4c, d). Co-IP of endogenous host proteins in ZIKV-NS4B[HA]-infected JEG-3 cells confirmed specific NS4B interactions with HSD17B7, FPGS and UBA5 (Fig. 3e). Altogether, these experiments identify a new set of bona fide ZIKV-NS4B interactors functionally promoting or restricting viral replication.

## UBA5 exerts a UFMylation-dependent pro-viral role in ZIKV replication

Among the strongest ZIKV-NS4B interactors, we identified ubiquitin-like modifier-activating enzyme 5 (UBA5), an E1 activating enzyme that specifically catalyzes the first step of UFMylation[28]. UFMylation is an emerging post-translational modification involved in regulation of the

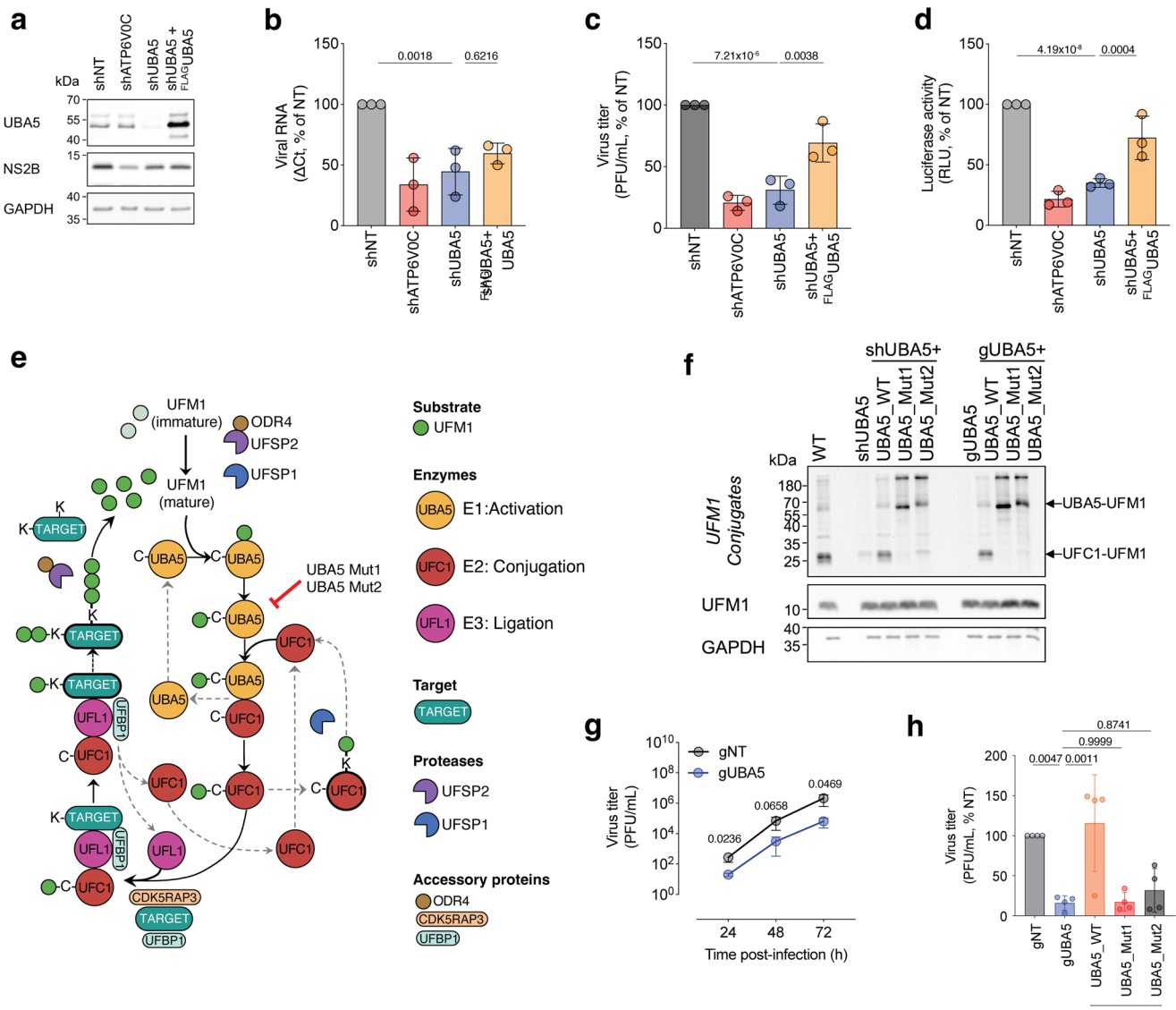

**Fig. 4 | UBA5 pro-viral function in viral replication is UFMylation-dependent.** UBA5 knock-down JEG-3 cells (shUBA5) or control cells (shNT) were infected with ZIKV H/PF/2013 wild-type strain (a-c) or ZIKV H/PF/2013 expressing *Renilla* luciferase (d) (MOI = 0.1). At 24 hpi, abundance of intracellular viral proteins (**a**) and viral RNA (**b**) were determined by western blotting and RT-qPCR (**b**), respectively. Infectious particle production was determined by PFU assay on cell culture supernatants (**c**), while viral replication was determined by luciferase activity (**d**). **e** Schematic representation of the UFMylation pathway and its main players. **f** Western blot analysis of UFMylation pathway activity after the complementation of UBA5 knock-out (KO) JEG-3 cells with wild-type (WT) or UFMylation-null UBA5 mutants. **g** Kinetics of infectious particle release in wild-type (WT) or UBA5 KO JEG-3 cells. Cells were infected with ZIKV H/PF/2013 (MOI = 0.01), supernatants collected at different times post-infection and analysed by PFU assay. **h** UBA5 KO JEG-3 cells complemented with either wild-type (WT) or UFMylation-null UBA5 mutants (L397R, Mut1; and M401R, Mut2)[37], were infected with ZIKV H/PF/2013 (MOI = 0.01) and virus titers measured in the supernatants at 48 hpi by PFU assay. **b–d**, **g** and **h**, $n = 3$ biological replicates; bars (or circles in g) represent the mean and error bars represent the standard deviation of the mean. Statistical significance was determined by two-way ANOVA with Tukey's multiple comparisons test (**b, c**), two-tailed unpaired t-test on $\log_{10}$-transformed PFU/mL values with Welch's correction and Holm–Šidák correction for multiple comparisons (**g**) or one-way ANOVA with Dunnett's multiple comparisons test (**h**). **a**, **f** representative immunoblots from $n = 3$ biological replicates are shown.

DNA damage response[29–31], ER stress[32], antiviral RIG-I signalling[33], autophagy[34], and protein synthesis through regulation of the ER translocon[35,36].

To validate the pro-viral role of UBA5 in ZIKV replication, we generated stable UBA5 knock-down JEG-3 cells (Fig. 4a) and confirmed a significant reduction in viral RNA replication (Fig. 4b) and infectious particle production (Fig. 4c) using wild-type ZIKV H/PF/2013 as well as a luciferase-reporter ZIKV (Fig. 4d). Importantly, ectopic expression of FLAG-tagged UBA5 in UBA5-deficient cells rescued ZIKV replication, corroborating the specificity of the observed pro-viral effects.

UFMylation involves the covalent conjugation of ubiquitin-like fold modifier 1 (UFM1), a 9.1 kDa protein, to lysine residues of target proteins, to modify their functions. This post-translational modification involves the E1 activase, UBA5, the E2 conjugase, UFC1, and the E3 ligase complex, UFL1/UFPB1. The pathway-specific protease UFSP2 mediates UFM1 maturation and recycling[37] (Fig. 4e).

To assess whether UBA5 plays a pro-viral role dependent on UFMylation pathway activity, we generated clonal UBA5 knock-out JEG-3 cells (JEG-3 gUBA5) completely devoid of residual UFMylation, as confirmed by the absence of UBA5 and UFC1 higher molecular UFM1-reactive conjugates (UFC1-UFM1 and UBA5-UFM1) (Fig. 4f).

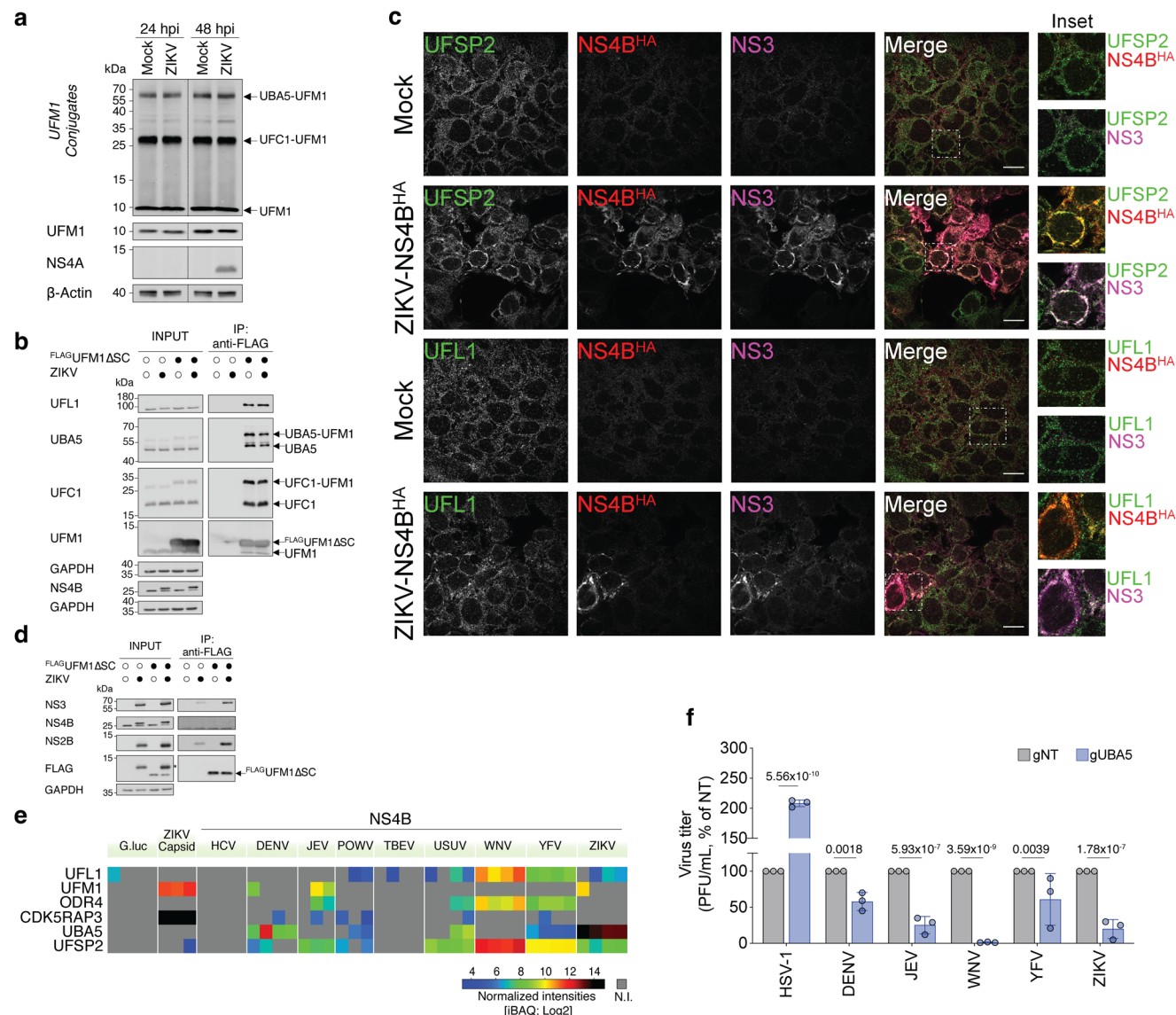

**Fig. 5 | Multipartite interactions between orthoflaviviral and UFMylation pathway members. a** Mock- or ZIKV-infected JEG-3 cells (ZIKV H/PF/2013, MOI = 0.1) were harvested at 24 and 48 hpi and subjected to non-reducing SDS-PAGE and western-blotting using antibodies against UFM1, NS4A and β-actin (representative of n = 2 independent experiments). **b** Immunoblot analysis of anti-FLAG immunoprecipitated extracts (right panels) and inputs (left panels) from mock- or ZIKV-infected (ZIKV H/PF/2013, MOI = 0.01) JEG-3 cells stably expressing ᶠᴸᴬᴳUFM1ΔSC. Eluates were stained with antibodies specific to UFMylation pathway members as indicated on the left (representative of n = 2 independent experiments) **c** Subcellular distribution of UFMylation pathway members upon ZIKV infection (UFSP2 and UFL1). JEG-3 cells infected with a ZIKV infectious molecular clone carrying an internal HA tag within NS4B (ZIKV-NS4Bᴴᴬ, MOI = 5) were fixed 24 hpi and stained with anti-UFSP2 (top two rows) or anti-UFL1 (bottom two rows), anti-HA and anti-NS3 antibodies. Scale bar = 10 μm. A representative experiment of n = 3 is shown. **d** Immunoblot analysis of anti-FLAG immunoprecipitated extracts (right

panels) and inputs (left panels) from mock- and ZIKV-infected JEG-3 cells stably expressing ᶠᴸᴬᴳUFM1ΔSC. Eluates were stained with antibodies specific to viral proteins as indicated on the left (representative of n = 2 independent experiments). Asterisk denote signal from prior anti-NS2B incubation. **e** Heat map of all detected UFMylation pathway members across the extended NS4B PPI networks. Intensity-based absolute quantification (iBAQ) of protein abundance of UFMylation-related host proteins across baits. N.I. not identified. **f** Replication of multiple orthoflaviviruses is inhibited in UBA5 KO cells. Control (gNT) or UBA5 KO (gUBA5) JEG-3 cells were infected with ZIKV, DENV2, JEV, WNV and YFV. At 48 hpi, infectious titers in the supernatant were determined by PFU assay. Herpes simplex virus 1 (HSV-1) was used as control. Bars represent the mean and error bars represent the standard deviation of the mean (n = 3 biological replicates). Statistical significance was determined by two-way ANOVA with Sidak's multiple comparisons test.

Strikingly, ZIKV replication was even more severely impaired compared to UBA5 knock-down cells (Fig. 4g), and was fully rescued by ectopic expression of wild-type UBA5 (Fig. 4h). Interestingly, two distinct UBA5 mutants severely impaired in their ability to sustain functional UFMylation due to selective defects in UFM1 activation (UBA5_Mut1) or UFC1 binding[38] (UBA5_Mut2), failed to rescue viral replication, demonstrating a functional role for UFMylation in viral replication. Altogether, these results unequivocally confirm a pro-

viral role of UBA5 in viral replication via a UFMylation-dependent mechanism.

**Multiple UFMylation pathway members are recruited to viral replication compartments via interactions with several viral proteins**

To further characterize the impact of ZIKV infection on UFMylation, we next analyzed the migration profile of UFM1-reactive species under

non-denaturing conditions (Fig. 5a). In agreement with previous reports[38], we identified all the major UFM1-conjugates, including free UFM1, UFC1-UFM1 and UBA5-UFM1, alongside a few additional UFM1-reactive species. However, their relative abundance was not significantly affected by ZIKV infection (Fig. 5a). To further investigate potential perturbations within the UFMylation pathway, we next characterized the interactions of UFM1 with the core components of this pathway. As expected, FLAG-tagged mature UFM1 ([FLAG]UFM1ΔSC) was found to form complexes with UBA5, UFC1 and UFL1. However, none of these interactions was significantly modulated by ZIKV infection (Fig. 5b). Furthermore, analysis of crude fractions from mock- or ZIKV-infected cells, revealed no major differences in the degree of UFMylated RPL26 at the ER, suggesting comparable levels of UFMylation at the ER (Supplementary Fig. 5a). We next assessed potential co-opting of UFMylation components by viral replication complexes, characterizing the subcellular distribution of several pathway members in ZIKV-infected cells. Upon infection, both UFL1 and UFSP2, dramatically redistributed to viral replication sites and co-localized with NS4B and the viral protease/helicase NS3 (Fig. 5c). Notably, analysis of UFM1 eluates from infected cells, revealed a significant enrichment of both NS3 and its co-factor NS2B (Fig. 5d). Collectively, these results suggest that multiple UFMylation components are actively recruited to sites of viral replication via multipartite interactions with distinct elements of the viral replicase.

## The UFMylation pathway exerts a conserved pro-viral function across the *Orthoflavivirus* genus

Having identified multiple interactions between viral proteins and members of the UFMylation pathway, we retrieved the interaction profiles of all detected UFMylation pathway members across our entire NS4Bome. This analysis revealed that, albeit with lower affinities, the NS4B proteins of other orthoflaviviruses such as DENV and YFV, also associate with UBA5 (Fig. 5e). Surprisingly, we identified additional associations of nearly all orthoflavivirus NS4Bs with at least one UFMylation pathway component, such as UFL1 (USUV, WNV, YFV, ZIKV), ODR4 (YFV, WNV) and UFSP2 (USUV, WNV, YFV, ZIKV, JEV) (Fig. 5e), suggesting a conserved targeting of the pathway through selectively divergent mechanisms. Unexpectedly, also the capsid protein of ZIKV (one of the control baits in our NS4Bome) associated very strongly with UFM1 itself and with CDK5RAP3, a substrate adaptor that directs UFMylation to the ribosomal protein RPL26[39] (Fig. 5e).

To determine whether UFMylation is functionally required across the entire viral genus, we tested the replication fitness of multiple orthoflaviviruses in UBA5 KO JEG-3 cells (Fig. 5f). These experiments revealed a significant inhibition of viral replication across a broad range of viral species including DENV, YFV, JEV, ZIKV and WNV (40-99% inhibition compared to gNT control, Fig. 5f). In agreement with similar reports on other DNA viruses, such as Epstein-Barr virus (EBV)[39], replication of herpes simplex virus 1 (HSV-1) was significantly up-regulated in UBA5 KO cells (Fig. 5f), likely as a result of reduced type I IFN responses[33,40].

Altogether, these results demonstrate that UFMylation exerts a broadly-conserved pro-viral role in orthoflavivirus replication, through multipartite interactions between multiple UFMylation pathway members and several viral proteins.

## Pharmacological inhibition of UFMylation results in antiviral activity

Several UFMylation inhibitors are being investigated for therapeutic applications, particularly in cancer and other diseases[41-44]. Among these, we evaluated the antiviral potential of DKM 2-93, a selective small-molecule inhibitor that covalently binds UBA5 catalytic cysteine, thereby inhibiting UFMylation[45] (Fig. 6a, Supplementary Fig. 6a). DKM 2-93 significantly inhibited ZIKV replication in the low micromolar range without overt cytotoxicity ($EC_{50}$ = 12,92 μM; $CC_{50}$ = 81,23 μM;

Fig. 6b). These effects were characterized by a reduction in both intracellular viral protein abundance (Fig. 6c) and infectious particle production by 10-fold (Fig. 6d), and could be phenocopied in multiple human cell lines (Fig. 6f, g and Supplementary Fig. 6b)

To identify the step(s) of the replication cycle targeted by UBA5 inhibition, we performed time-of-addition experiments, adding the compound before (pre-treatment), during (co-treatment) or after (post-treatment) virus absorption, and measured ZIKV replication after 48 h. DKM 2-93 retained significant and comparable anti-ZIKV inhibitory activity when added up to 3 h after infection, suggesting a role of UFMylation in a post-entry step of viral replication (Fig. 6e). To determine whether DKM 2-93 affects viral RNA (vRNA) synthesis and/or translation, we used a modified ZIKV subgenomic replicon (sgZIKV-R2A)[27]—a self-replicating vRNA lacking the coding sequences of structural proteins and therefore unable to produce viral particles—and evaluated the effect of the drug on luciferase expression following in vitro transcription and electroporation of these RNAs (Fig. 6f, g). Under these conditions, intracellular luciferase activity measured over time exclusively reflects efficient vRNA translation and replication, independent of viral entry and virus assembly or release. Luciferase activity at 4 h post electroporation was not impacted by drug treatment, thus excluding DKM 2-93 effects on viral translation (Fig. 6f). Similarly, DKM 2-93 treatment did not significantly affect ZIKV replication up to 96 h post-electroporation (Fig. 6g). In contrast, treatment of cells infected with a full-length ZIKV-R2A resulted in significant reduction of viral replication at late stages of infection (Fig. 6b), suggesting a selective inhibition of virion assembly or virus release. Collectively, these results indicate that functional UFMylation is critically required for late stages of the viral replication cycle, independent from viral entry, viral RNA translation and viral RNA replication.

## UFMylation modulates viral replication via an IFN-independent mechanism

To shed light on potential mechanisms underlying UFMylation-dependent pro-viral functions, we next assessed the impact of UBA5 inhibition on cellular transcriptional responses. We analyzed transcriptomic changes in either control (shNT) or UBA5 knock-down cells (shUBA5), following mock or ZIKV infection (Supplementary Fig. 5). Gene set enrichment analysis of transcripts significantly regulated by UBA5 silencing, revealed an overall moderate impact on mRNA abundance, with only 15 and 11 transcripts robustly up- and down-regulated, respectively (|Log2(Fold-change)|>2; Supplementary Fig. 5b). Among these, we observed an upregulation of genes involved in mitochondrial protein import (*TIMM17A*) and pre-mRNA processing (*U2AF1L5*, and *RBMY1F*) and down-regulation of genes involved in heme metabolism (*HMOX1*), cellular stress responses (*TXNIP*) and RNA binding (*RBMY1E*) (Supplementary Data 4, Supplementary Fig. 5b, c). These changes were largely observed also in ZIKV-infected cells, suggesting a lack of correlation with UFMylation-dependent pro-viral functions (Supplementary Fig. 5d, e). As expected, ZIKV infection robustly stimulated innate immune responses, as inferred by the analysis of an extended panel of 48 known IFN-stimulated genes (ISGs), including *RIG-I*, *IFIT1*, *IFIT3* and *IFNL2* (Supplementary Fig. 5f). Interestingly, silencing of UBA5 did not exacerbate innate immune responses in ZIKV infected cells, in agreement with recent reports describing UFMylation as a positive post-translational regulator of type I IFN responses[34,46].

Altogether, these results suggest that the inhibition in viral replication observed UBA5-deficient cells is associated with a mild transcriptomic signature and is not due to exacerbated activation of innate anti-viral immune signalling.

## UFMylation modulates mitochondrial respiration and morphodynamics

Orthoflaviviruses, including DENV and ZIKV, profoundly alter mitochondrial respiration in a time-dependent manner, correlating with

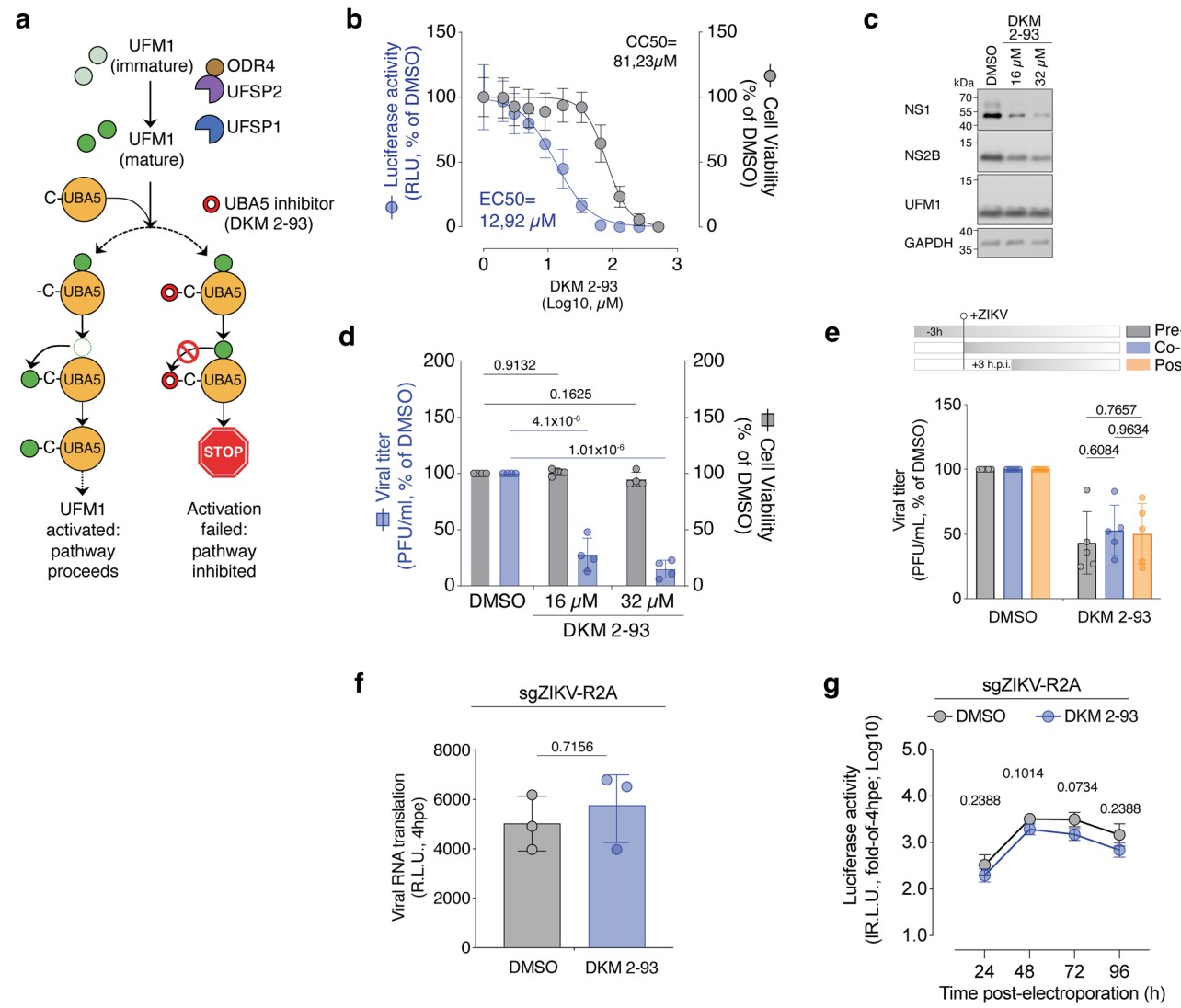

**Fig. 6 | Pharmacological inhibition of the UFMylation pathway impairs ZIKV infectious particle production. a** Schematic representation of DKM 2-93 inhibitory activity on UFMylation. DKM 2-93 competitively binds to the catalytic cysteine of UBA5, and prevents the activation of UFM1. **b** Dose-response curve of DKM 2-93 ZIKV antiviral activity in JEG-3 cells. Cells were treated with increasing concentrations of the inhibitor for 24 h, then infected with a *Renilla* luciferase ZIKV H/PF/2013 reporter virus (MOI = 0.1), and treated with the inhibitor for another 24 h. Cell viability and virus replication were determined at 24 hpi by resazurin and luciferase assays, respectively. **c** Western blot analysis of ZIKV-infected cells upon DKM 2-93 treatment, stained with anti-NS1 and anti-GAPDH specific antibodies (representative of n = 3 biological replicates). **d** Antiviral activity of DKM 2-93. JEG-3 cells were infected with ZIKV (ZIKV H/PF/2013, MOI = 0.01), and treated with either DMSO or two concentrations of DKM 2-93. Virus titers were determined by PFU assay 48 h.p.i. **e** Time-of-addition analysis of DKM 2-93 antiviral activity. JEG-3 cells were treated

3 h before (pre-; grey bar), during (co-; blue bar) or or 3 h after (post-; orange bar) virus absorption (ZIKV H/PF/2013, MOI = 0.01) with DKM 2-93 (32 μM). Virus titers were determined by PFU assay 48 h.p.i. **f**, **g** Huh7 cells were electroporated with in vitro transcribed RNA of a ZIKV subgenomic replicon expressing *Renilla* luciferase (sgZIKV-R2A), and treated with DKM 2-93 (32 μM) immediately thereafter. Luciferase activity was measured 4 h post electroporation to assess effects on viral RNA translation (**f**), and up to 96 h post electroporation to assess effects on viral RNA replication (**g**). **b**, **d–g**, n = 3 (**c**, **f**), n = 4 (**d**, **g**), and n = 5 (**e**) biological replicates; bars (circles for b and g) represent the mean and error bars represent the standard deviation of the mean. Statistical significance was determined by one-way ANOVA with Dunnett's multiple comparisons test (**d**), two-way ANOVA with Tukey's multiple comparisons test (**e**); two-way ANOVA with Sidak's multiple comparisons test (**f**) or multiple unpaired two-tailed t-test on log$_{10}$-transformed PFU/mL values with Welch correction and Holm–Šidák correction for multiple comparisons (**g**).

changes in the stoichiometry of the electron transport chain and the abundance of Krebs cycle metabolites[24]. In addition, NS4B expression induces mitochondrial elongation by counteraction of the fission machinery, ultimately promoting virus replication[7,47]. Interestingly, our NS4Bome identified oxidative phosphorylation (OXPHOS) and mitochondrial processes-associated factors amongst the most enriched NS4B interactors broadly targeted by the majority of ortho-flaviviruses (Fig. 2c), suggesting a possible functional link between UFMylation and mitochondria. To determine whether UFMylation inhibition perturbs the mitochondrial network, we analyzed several

morphological features of the mitochondrial network in UBA5 KO (JEG-3 gUBA5) and control (JEG-3 gNT) cells (Fig. 7a–g). UBA5 depletion was associated with a significant increase in length (Fig. 7d, e), branching (Fig. 7f) and connectivity (Fig. 7g) of the mitochondrial network, without affecting their area or overall perimeter (Fig. 7b, c). Comparable results were obtained when JEG-3 cells were treated with DKM 2-93 (Supplementary Fig. 7a–g), further corroborating the specificity of this UBA5 inhibitor and the notion that UBA5 negatively regulates mitochondria elongation. Considering that enhanced elongation of mitochondria was reported to promote DENV and ZIKV replication in

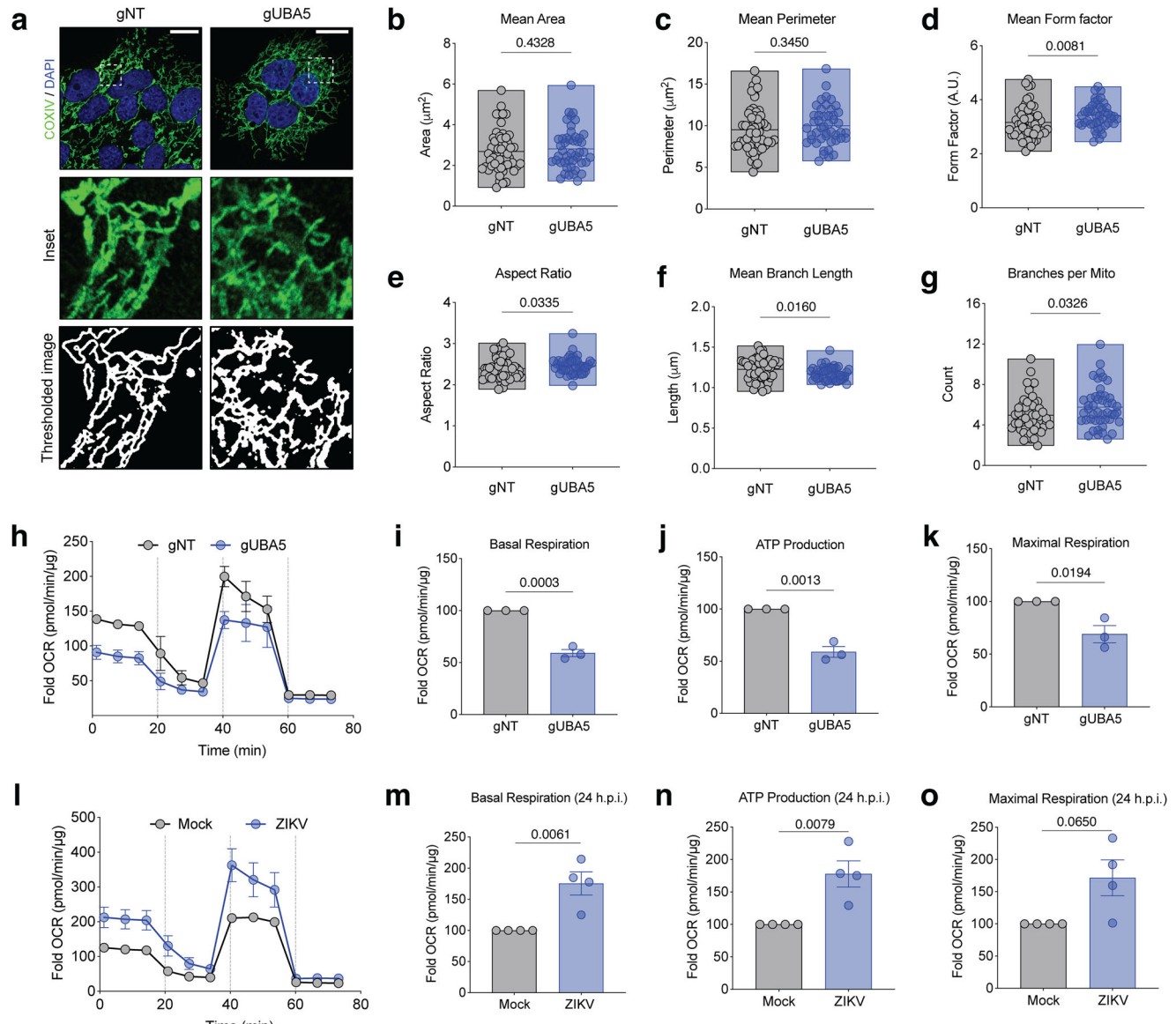

**Fig. 7 | UBA5 is essential for mitochondrial morphodynamics. a** Control (gNT) and UBA5 KO (gUBA5) JEG-3 cells cultured for two days were imaged by confocal microscopy. Representative images of mitochondria labeled with anti-COXIV (green) and the nuclear stain DAPI (blue). Scale bar, 20 μm. Quantitative analysis of mitochondrial morphology using the Mitochondrial Analyzer plugin in ImageJ/Fiji. **b–g** represent the mean area which calculates the average size of individual mitochondria per cell (**b**), the mean perimeter (**c**), the mean form factor (perimeter² / 4π × area) which reflects the mitochondrial shape, with higher values indicating more elongated structures and values closer to 1 indicating rounder mitochondria (**d**), the aspect ratio, the ratio of the major axis to the minor axis with higher AR > 1.5– 2.0 indicates elongated mitochondria associated with fusion (**e**), the mean branch length, the average length of individual mitochondrial branches within the network (**f**), and the branches per mitochondria (**g**) in UBA5 KO JEG-3

cells. **h–k** Knock-out of UBA5 impairs mitochondrial respiration. The Oxygen Consumption Rate (OCR) of UBA5 knock-out JEG-3 cells was measured at the indicated time points using the Seahorse technology. OCR values were first normalized to total protein content (μg per condition) and then to the mean basal OCR of control cells in each independent experiment (**h**). The basal respiration (**i**), ATP production (**j**), and maximal respiration (**k**) were quantified from the mitochondrial respiration profile. **l–o** ZIKV infection modulates mitochondrial respiration. Data from *n* = 3 biological replicates are shown in all panels. **b–g** floating bars (min. to max.), the middle line of the corresponds to the mean; **h, l** each circle represents the mean and error bars represent the standard deviation of the mean; and panels **i–k**, and **m–o**, the bars represent the mean and error bars represent the standard error of the mean. Statistical significance was determined by two-sided Mann-Whitney test (**b–g**) or unpaired two-sided t-test (**i–k, m–o**).

liver cells[7,47], these results suggest that the proflaviviral role of UBA5 is not linked to its activity on mitochondria morphodynamics.

We next investigated the impact of UBA5 depletion and pharmacological inhibition on multiple parameters of mitochondrial respiration in Seahorse assays, by measuring the oxygen consumption rates (OCR) while sequentially adding selective inhibitors of the electron transport chain of the inner mitochondrial membrane (Fig. 7h–k). JEG-3 gUBA5 cells showed a significant reduction in basal respiration

(Fig. 7i), ATP-linked respiration (Fig. 7j), and maximal respiration (Fig. 7k). These effects were fully phenocopied by treatment of JEG-3 cells with DKM 2-93, suggesting a direct modulation of mitochondrial respiration by UFMylation (Supplementary Fig. 7). In contrast, ZIKV infection elicited a reciprocal increase in all OCR parameters at 24 hpi in line with previous observations in liver cells[47], confirming an up-regulation of mitochondrial respiration at early stages of ortho-flavivirus infections (Fig. 7l–o).

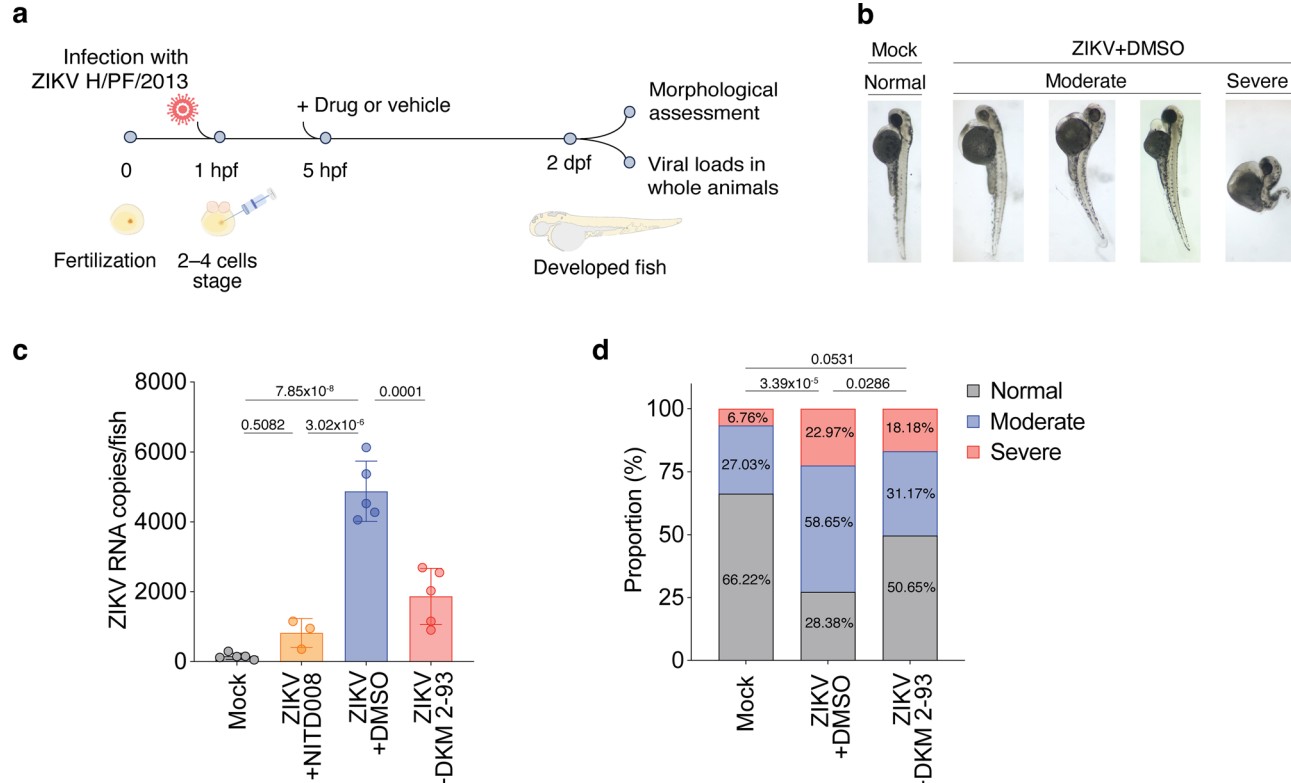

**Fig. 8 | A selective inhibitor of UBA5 displays potent antiviral activity in a zebrafish-based model of ZIKV infection. a** Experimental scheme of ZIKV infection in zebrafish. **b** Representative images of zebrafish larvae at 2 days post-fertilization (dpf). **c** ZIKV RNA levels in zebrafish larvae at 2 dpf determined by ddPCR (*n* = 5 biological replicates). Bars represent the mean and error bars represent the standard deviation of the mean. Statistical significance was determined by one-way ANOVA followed by Tukey's post hoc test. **d** Phenotype proportions of zebrafish larvae at 2 dpf were examined across conditions as previously described[48] (*n* = 3 independent experiments): Mock-infected (*n* = 74), ZIKV-infected (*n* = 74), and ZIKV-infected treated with 20 μM DKM 2-93 (*n* = 77). Statistical significance was determined by one-way ANOVA followed by Tukey's post hoc correction (**c**) and two-sided pairwise Fisher's exact test with FDR correction (**d**).

Together these results establish a novel connection between UFMylation and mitochondrial respiration, suggesting that UBA5 may contribute to ZIKV-induced stimulation of mitochondrial respiration to promote viral replication.

**UFMylation inhibition exerts antiviral activity in vivo**

We have recently established an in vivo zebrafish model of ZIKV infection, which phenocopies several patho-physiological features associated with ZIKV infections in humans, such as microcephaly as well as neural progenitor cells infection and depletion[48]. This model capitalizes two major advantages: (i) the rapid development of the embryonic/larval brain, which allows phenotypic characterization of host and viral determinants involved in ZIKV-induced neuropathogenesis, and (ii) time-effective quantitative assessment of antiviral drug efficacy in a large number of animals (2–3 days)[48]. To assess the antiviral potential of UFMylation pathway inhibition in vivo, zebrafish embryos were infected with ZIKV at the 2-4 cell stage and subsequently treated with non-cytotoxic concentrations of DKM 2-93 for two days, i.e. during the first stages of brain development (Fig. 8a). Treatment with NITD008, a pan-flaviviral NS5 polymerase inhibitor[49], was used as control. The inhibition of UFMylation by DKM 2-93 significantly reduced ZIKV vRNA levels in whole larvae (pool of 6-10 larvae per condition per independent experiment) at 48 h post-infection/fertilization compared to DMSO-treated animals (Fig. 8b, c). Importantly, the parallel analysis of ZIKV-induced morphological defects (such as abnormal morphology, curved tail and oedema) on 225 larvae revealed that this reduction in viral replication correlated with a significant reversion of virus-induced comorbidities in the infected larvae (from 58,65% to 31,17%, and from 22,96% to 18,18% of animals displaying

moderate and severe phenotypes, respectively) when compared to DMSO-treated animals (Fig. 8d).

Collectively, these results demonstrate significant antiviral activity of DKM 2-93 in vivo and corroborate the functional relevance of the UFMylation pathway in ZIKV pathogenesis during organism development. This further highlights the therapeutic potential of the UFMylation pathway as a novel host-targeting antiviral strategy to control viral infections in vivo and attenuating neurovirulence in the developing brain.

## Discussion

In this study, we characterize for the first time the extended genus-wide orthoflavivirus NS4B protein-protein interaction network (NS4Bome). This integrative approach, enabled accurate mapping of cellular hubs and pathways broadly targeted by NS4B across orthoflaviviruses, as well as the identification of virus species-specific interactions underlying potentially divergent mechanisms of host adaptation across eight prototypic orthoflaviviruses, spanning diverse pathologies (e.g. encephalitic *vs.* hemorrhagic) and vector preferences (e.g. mosquitoes *vs.* ticks). This extended PPI network revealed broadly conserved targeting of cellular pathways involved in oxidative phosphorylation, protein trafficking, co-translational targeting to membranes and mitochondrial membrane organization, generalizing previous reports on individual species of the *Orthoflavivirus* genus[12–14] (Supplementary Data 5). Importantly, it corroborated the importance of mitochondrial functions modulation during orthoflavivirus infections via interactions with NS4B[7,50]. Furthermore, we identified several host proteins involved in ER-associated degradation (e.g., DERL2, C6orf120), which is in line with previous reports from us and others on

Valosin-containing protein (VCP), an NS4B interactor and positive regulator of orthoflaviviral infection, which is a key factor in proteostasis through targeting of retrotranslocated misfolded ER proteins to the proteasome[18,51–56]. Functional RNAi-based phenotypic screening on a subset of NS4B host-interacting proteins, led to the identification of 5 novel host dependency and 14 host restriction factors, that significantly modulate viral replication (Fig. 3, Supplementary Fig. 3). Interestingly, one of the newly identified restriction factors, SEL1L, was previously reported as host-dependency factors for JEV in a different cellular background[18]. This difference could reflect divergent roles for this host protein in JEV and ZIKV replication, or result from cell type-dependent specificities. Our ongoing studies will further elucidate the relationship between each of these orthoflavivirus NS4Bs and their targets, uncovering additional mechanisms by which each species manipulate the host cell.

Here, we focused on the interaction between UBA5 and ZIKV-NS4B, and show that UBA5 plays an important pro-viral function via a UFMylation-dependent mechanism (Figs. 4–7). UBA5 is the E1 activase of UFMylation, an emerging ubiquitination-like post-translational modification which culminates with the covalent attachment of UFM1 onto lysine residues of target proteins, thereby modulating their activity[37]. This chain of events involves transfer of UFM1, a small 9.1 kDa protein, through an E2 conjugase (UFC1) and E3 ligase (UFL1), and a number of accessory proteins involved in recycling and conjugation of the substrate (UFSP1, UFSP2, ODR4, UFBP1 and CDK5RAP3). In this study, through a combination of genetic, biochemical and pharmacological approaches, we provide multiple lines of evidence demonstrating a pro-viral role for UBA5 in orthoflavivirus replication (Figs. 4, 6, 7, 8). Notably, we show that orthoflaviviruses, most likely via NS4B and other orthoflavivirus proteins, such as NS2B-3 and capsid, actively recruit UFMylation pathway members to sites of viral RNA replication via multipartite interactions (Fig. 5c, e). This suggests that the UFMylation pathway is physically and functionally hijacked by orthoflaviviruses to sites of viral replication. We note that during the revision of this manuscript, Schmidt and colleagues have also independently reported that UFMylation exerts a pro-viral role in orthoflavivirus infectious particle production, and identified interactions of UFL1 with multiple structural and non-structural proteins of DENV and ZIKV[57]. In agreement to our observations, this study also did not observe any upregulation in innate immune responses, confirming that alternative mechanisms are responsible for the pro-viral function of UFMylation in orthoflavivirus replication. Interestingly, in our study no significant changes in global UFMylation activity could be observed (Fig. 5a, b, Supplementary Fig. 5a), suggesting that only selected host or viral targets might be differentially UFMylated. Additional experiments using analytical methods with inherently higher sensitivity could shed light onto the molecular targets underlying the pro-viral function of UFMylation.

UFMylation has been implicated in diverse cellular processes including ribosomal activity and translation stalling at ribosomes[36,58], ER-phagy[59], cell cycle regulation[60] and DNA damage response[31]. Furthermore, it has been shown to influence viral infections, such as hepatitis A virus (HAV) and EBV, by modulating host translation[61] and innate immune signaling[40]. Notably, UFMylation of 14-3-3ε was reported to facilitate RIG-I binding, translocation to mitochondria-associated membranes and interferon induction, underscoring its regulatory role in antiviral immunity[40]. The activation of the RIG-I/MAVS signaling correlated with a recruitment of UFL1 to this inter organelle interface[62], implying a functional interplay between the UFMylation pathway and this sub-cellular compartment. Interestingly, ER-mitochondria contact sites (ERMCs) are disrupted in DENV/ZIKV infected hepatocarcinoma cells[7,24], suggesting that this morphological alteration might contribute to the hijacking of the UFMylation pathway by orthoflaviviruses.

While these studies show that UFMylation regulates host protein functions during viral infection, here we report a novel pro-viral mechanism involving active recruitment of the UFMylation machinery to viral replication sites, independent of viral RNA translation or innate immune activation. Indeed, using RNA-seq to broadly map UFMylation responses on global cellular transcription, we corroborated a positive regulation of multiple ISGs by UBA5, but no overt activation of canonical innate immune responses was observed in ZIKV-infected UBA KO cells (Supplementary Fig. 5). Furthermore, leveraging a combination of genetically modified viral genomes and time-of-addition studies, we ruled out effects of UFMylation on viral entry, viral RNA replication or translation, supporting a role for UFMylation in virus assembly, virion release or viral spread (Fig. 6b–g).

Interestingly, using a combination of unbiased multiparametric image-based analysis of the mitochondrial network and accurate measurements of mitochondrial respiration, we observed a profound modulation of mitochondrial activity and network morphology in UFMylation-deficient cells (Fig. 7). Pharmacological or genetic inhibition of UFMylation resulted in increased branching of the mitochondrial network and profound inhibition of basal and maximal respiration, as well as ATP production (Fig. 7h–k; Supplementary Fig. 7h–k). Strikingly, ZIKV-infected cells reversely mirrored these effects, displaying a strong increase in energy and mitochondrial respiration at 24 h post-infection (Fig. 7h–k). Mitochondria elongation and branching generally correlate with enhanced respiration. However, mitochondria hyperfusion has also been observed as a feedback mechanism in the presence of a respiratory deficit. Thus, there is a possibility that UBA5 KO and drug-treated cells exhibit an increase in mitochondria branching and elongation in response to the observed alteration in mitochondrial respiration. Collectively, these observations point towards a functional role of UFMylation as a positive regulator of mitochondrial respiration, at stages where virus-infected cells display a high energy demand to create a conducive environment to viral replication. These observations are in agreement with studies by us and others, reporting an overall increase in multiple steps of mitochondrial respiration in hepatic cells[46]. Furthermore, earlier studies supported an NS4B-mediated modulation of mitochondria morphodynamics in orthoflavivirus-infected cells[7], suggesting that mitochondria elongation is likely a conserved, cell-type-independent feature across multiple orthoflaviviruses. In light of these observations, we speculate that UFMylation might contribute to the alteration of ER-mitochondria contact sites (ERMC) via enhanced UFMylation of host or viral substrates proximal to viral replication sites. More work is needed to establish the causal relationship between mitochondrial respiration and UFMylation, and the key viral or cellular effectors involved.

In addition to identifying interactions between UBA5 and NS4B of DENV and ZIKV virus, we also mapped multipartite interactions between several viral proteins (NS2B/3 and capsid) and different components of the UFMylation machinery (UFM1 and CDK5RAP3, respectively) (Fig. 5). Furthermore, leveraging the unique perspective offered by our extended NS4Bome, we mapped associations of several orthoflaviviruses NS4Bs, including DENV, WNV, YFV, JEV and USUV, with additional members of the UFMylation machinery such as UFSP2, the protease responsible for recycling of UFM1 from target proteins, and ODR4, its scaffolding co-factor[63] (Fig. 5e). These observations, together with the pro-viral activity of UFMylation across a broad panel of orthoflaviviruses (Fig. 5f), argue for an evolutionary-conserved requirement of UFMylation across the entire viral genus, via interactions with different pathway members downstream of UBA5. Importantly, recruitment of UFMylation pathway members to sites of RNA replication (Fig. 5c), without detectable changes in general UFMylation levels (Fig. 5a) or the abundance of intermediate UFM1 conjugates at the ER (Fig. 5b, Supplementary Fig. 5a), strongly support the notion of an increased local demand of UFMylation at sites of viral replication.

Arguably, in agreement with a recent report[57], these effects seem particularly relevant at late stages of infection, as they affect infectious particle production, independently of viral entry, viral RNA translation and viral RNA replication (Fig. 6c–g). More work is needed to dissect whether assembly, trafficking or specific infectivity of virus particles are affected. Interestingly, COPII-mediated vesicular transport and co-translational protein targeting to membranes were amongst the most enriched cellular functions in our extended NS4B interaction atlas (Fig. 2b). Since UFMylation has recently been associated with COPII recruitment, anterograde transport[64] and specific protein-RNA interactions[65], it is tantalizing to speculate that UFMylation regulates trafficking of virions or protein–RNA interactions involved in genome packaging during virion assembly, providing further cues into poorly characterized mechanisms regulating orthoflavivirus assembly or release. Additional studies are required to elucidate the molecular drivers underlying UFMylation-mediated pro-viral functions, and identify potential viral or host proteins differentially UFMylated upon infection.

Leveraging a recently reported zebrafish model of ZIKV infection, alongside mechanistic and phenotypic studies in vitro, we provide proof-of-principle that pharmacological inhibition of UFMylation exerts antiviral activity in vitro and in vivo. To our knowledge, this is the first report describing UFMylation as a tractable pathway for therapeutic or prophylactic treatment of viral infections. Notably, treatment of ZIKV-infected zebrafish embryos with a small molecule inhibitor of UBA5 significantly reduced viral loads, concomitantly with less severe morphological manifestations of the infection. Interestingly, while the use of UBA5 $^{-/-}$ lines for further mechanistic experiments is associated with technical challenges (it would require the co-injection of ZIKV particles and gRNA-Cas9 complexes at the 1-4 cell embryonic stage), UBA5 deficiency in zebrafish was shown to impact mitochondria functions and peripheral and central nervous systems development[66]. Improved chemotypes or alternative strategies selectively targeting this pathway might serve as highly potent broad-spectrum antiviral inhibitors and provide valuable model systems to characterize the relevance of UFMylation in ZIKV pathogenesis.

Collectively, this study provides a comprehensive network to rationalize NS4B effector functions in human cells, shedding light on conserved and species-specific activities of orthoflavivirus NS4B. Furthermore, it identifies a conserved role for UFMylation across orthoflaviviruses, exemplifying the use of genus-wide PPI networks to identify actionable host targets for the development of broad-spectrum antivirals.

## Methods

### Plasmids

The pWPI_puro constructs of the orthoflavivirus 2k-NS4B, *Gaussia* luciferase (GLuc), HCV-NS4B, and ZIKV-capsid, all C-terminally tagged with HA were synthesized as gBlocks fragments (Integrated DNA Technologies), and cloned into pWPI_puro vector using *BamHI* and *SpeI* sites. The pLKO constructs encoding non-targeting shRNA (plasmid SHC001; shNT-1), shRNA targeting the known virus restriction factor ATP6V0C (shATP6V0C-1), shRNA targeting UBA5 (clone ID: TRCN0000008018; shUBA5) and two shRNAs targeting UFM1 (shUFM1_1: TRCN0000232599 and shUFM1_2: TRCN 0000037004) were obtained from the MISSION TRC lentiviral library (Sigma-Aldrich).

A shRNA- and gRNA-resistant, N-terminally FLAG-tagged UBA5 ($^{FLAG}$UBA5) and a guide RNA-resistant, N-terminally FLAG-tagged mature UFM1 ($^{FLAG}$UFM1ΔSC) were synthesized as gBlocks fragments (Integrated DNA Technologies). $^{FLAG}$UBA5 shRNA- and gRNA-resistant variants were generated by shuffling the codons in the shRNA and gRNA targeting region of the wildtype UBA5 (shRNA: $^{901}$CCT CAG TGT GAT GAC AGA AAT$^{921}$ and gRNA: $^{223}$GCC GTA GCA ATA GTA GGT GT$^{242}$) using the online tool 'Synonymous Mutation Generator' (http://jong2.

pythonanywhere.com) and manually correcting the Codon Adaptation Index. shRNA resistant UBA5: $^{901}$CCC CAA TGC GAC GAT CGC AAC$^{921}$ and gRNA resistant UBA5: $^{223}$GCT GTC GCC ATC GTC GGC GT$^{242}$. UFM1 is translated in the cells in its precursor immature form that contains a serine followed by a cysteine amino acid residues at its C-term end. The cellular proteases UFSP2 cleaves these two residues during the UFM1 maturation process leaving the glycine residue as the neo C-term. Upon conjugation to target substrates, this glycine residue of the cleaved mature UFM1 forms an iso-peptide bond with a lysine residue of the target protein. UFM1ΔSC denotes the mature UFM1. The $^{FLAG}$UFM1ΔSC was also made gRNA resistant in a similar way as $^{FLAG}$UBA5. gRNA resistant UFM1: $^{15}$CTT CAA AAT AAC CCT CAC CA[34]. $^{FLAG}$UBA5 was PCR-amplified using the primers UBA5_Fwd: 5′-CGC GCC GGA TCC GCC ACC ATG-3′ and UBA5_Rev: 5′-TAG TTT ACT AGT CTA CAT ATT CTT CAT TTT GGC CAT GAG GTC TTC CAA GC-3′, and cloned into the pWPI_puro vector using *BamHI* and *SpeI* sites. This construct served as a template to generate UBA5 mutants (UBA5_Mut1 and UBA5_Mut2) via overlap extension PCR, using the primers UBA5_Fwd, UBA5_Mut1_Fwd: 5′-GAA ACG AGA AGG TGT TCG TGC AGC CAG TCT TCC T-3′, UBA5_Mut1_Rev: 5′-AGG AAG ACT GGC TGC ACG AAC ACC TTC TCG TTT C-3′ and UBA5_Mut2_Rev: 5′-GTA GTT TAC TAG TCT ACA TAT TCT TCC TTT TGG CCA TCC TGT CTT CCA AGC TTT CAC CAG-3′. Mature UFM1 was cloned similarly by inserting *BamHI*- and *SpeI*-digested gBlock fragments into the pWPI vector. The pWPI construct of $^{FLAG}$UFM1ΔSC was used a template for the generation of the UFMylation deficient UFM1 mutant $^{FLAG}$UFM1ΔGSC and de-UFMylation resistant UFM1 mutant $^{FLAG}$UFM1G83A constructs using primers Fwd: 5′- AGC TTT GTT TAA ACG GCG CGG ATC CGC CAC CAT G-3′ and Rev: 5′- GCC CGT AGT TTA CTA GTT TAT TAC ACC CGA TCT CTA GGA ATA ATC CG-3′ (for $^{FLAG}$UFM1ΔGSC) or Rev: 5′- GCC CGT AGT TTA CTA GTT TAT GCC ACC CGA TCT CTA GGA ATA ATC CG-3′ (for $^{FLAG}$UFM1G83A). The mutants were cloned using *BamHI* and *SpeI* sites.

To generate the CRISPR-Cas9 gRNA targeting UBA5 (gUBA5), UFM1 (gUFM1) and UFSP2 (UFSP2) the oligos 5′-CAC CGA AGC AGC AGA ACA TAC TCT G-3′ and 5′-AAA CCA GAG TAT GTT CTG CTG CTT C-3′ (for UBA5), 5′- CAC CGC TTT AAG ATC ACG CTG ACG T-3′ and 5′-AAA CAC GTC AGC GTG ATC TTA AAG C-3′ (for UFM1), and 5′-CAC CGA ATA AGA GGA GGC TTG ATT-3′ and 5′- AAA CAA TCA AGG CCT CCT CTT ATT C-3′ (for UFSP2) were annealed and cloned into the pLentiCRISPRv2 plasmid, following previously described protocols[67,68]. Control non-targeting gRNA (gNT) pLentiCRISPRv2 and empty vector pLentiCRISPRv2 were previously described (Addgene).

The pFK_ZIKV_HPF2013[27] construct was used as a template to generate a pFK-based ZIKV H/PF/2013 construct carrying an HA epitope within NS4B (pFK_ZIKV-NS4B$^{HA}$), between nucleotides 66 and 67 (corresponding to the site between the 22$^{nd}$ and 23$^{rd}$ amino acid residues of ZIKV NS4B protein). A two-step overlap extension PCR with the primers NS4A_SacII_Fwd: 5′-AAG CCG CGG CGG CCC AAT GCG CGG AG-3′, NS4B_HA_Rev: 5′-AGC GTA ATC TGG AAC ATC GTA TGG GTA TGA TCC CTC CTC TCT CCT TCC CAT TAG-3′, NS4B_HA_Fwd: 5′-TAC CCA TAC GAT GTT CCA GAT TAC GCT GGA TCA GGG GCA ACC ATA GGA TTC TCA ATG GAC-3′ and NS5_EcoRI_Rev: 5′-AGT AGA ATT CCA GGG CCG ACA TCT GGT TC-3′, was performed to introduce the HA epitope with the NS4B. The amplicon was cloned using *SacII* and *EcoRI* sites. The reporter construct pFK_ZIKV_R2A was previously described[27].

### Antibodies

The rabbit polyclonal anti-human UBA5 (12093-1-AP, WB: 1:1'000), HSD17B7 (14854-1-AP; WB, 1:1'000), HACD3 (28572-1-AP; WB, 1:1'000), mouse anti-Calnexin (66903-1-Ig; WB, 1:1'000; IF, 1:500), ChromoTek monoclonal rat anti-HA (7c9; 1:500) antibodies, and secondary anti-rat horseradish peroxidase (HRP)-conjugated antibodies (SA00001-15; WB, 1:1'000), were purchased from Proteintech. The rabbit monoclonal anti-human UFM1 (ab109305; WB, 1:1'000), UFC1 (ab189252; WB, 1:3'000; IF, 1:500), and UFSP2 (ab192597; WB, 1:1'000; IF, 1:500)

antibodies were obtained from Abcam. The rabbit polyclonal antibody recognizing human UFL1 protein (NBP1-90691; WB, 1:1'000; IF, 1:500) was purchased from Novus Biologicals. The mouse monoclonal antibody recognizing ZIKV NS1 protein (GTX634158), and polyclonal rabbit anti-ZIKV NS2B (GTX133318), NS4A (GTX133704), and NS4B (GTX133311) antibodies (WB, 1:1'000), were purchased from GeneTex. The generation of the rat polyclonal anti-NS3 antibody by Medimabs (Montréal, Canada) was previously described (WB, 1:3'000; IF, 1:500)[51]. The mouse monoclonal antibodies recognizing FLAG (A8592; WB: 1:5000), FPGS (HPA050488; WB, 1:1000), HOOK2 (HPA043519; WB, 1:1000), TECR (HPA056488; WB, 1:1000), and HSD17B1 (HPA021032; WB, 1:1000), and secondary anti-mouse (A4416) and anti-rabbit (A6154) HRP-conjugated antibodies (WB, 1:10'000), along with agarose-conjugated anti-HA (A2095) and anti-FLAG (A2220) beads, were purchased from Sigma-Aldrich. The mouse monoclonal antibodies recognizing HA (2999; WB, 1:1000; IF, 1:100), the rabbit monoclonal anti-human COXIV (4850) antibody, and HRP-conjugated anti-human GAPDH (51332; WB, 1:5000) were purchased from Cell Signaling. The anti-human β-Actin monoclonal antibodies (sc-47778; WB, 1:5000) was purchased from Santa Cruz Biotechnology. Secondary antibodies conjugated to Alexa Fluor 488, Alexa Fluor 568, and Alexa Fluor 647, for anti-rabbit (A21206), anti-mouse (A10037, A21235), and anti-rat antibodies (A21247, A21208), respectively, were purchased from Invitrogen (IF, 1:500).

## Cell Culture

JEG-3 cells were cultured in Roswell Park Memorial Institute 1640 medium (RPMI-1640; Sigma-Aldrich), supplemented with L-glutamine and sodium bicarbonate. HEK293T, VeroE6, HeLa, Huh7, and MRC-5 cells were maintained in Dulbecco's Modified Eagle Medium (DMEM; Sigma-Aldrich). Both media were supplemented with 10% fetal bovine serum (FBS; Capricorn) and 1% penicillin–streptomycin (Sigma-Aldrich). Cells were maintained at 37 °C in a humidified incubator with 5% $CO_2$. All cell lines were authenticated by STR profiling (Eurofins Genomics).

## Generation of stable cell lines

Stable JEG-3 cell lines expressing C-terminally HA tagged 2k-NS4B of eight different orthoflaviviruses (DENV2, JEV, USUV, WNV, YFV, ZIKV, POWV and TBEV), *Gaussia* luciferase (GLuc), HCV-NS4B, and ZIKV-capsid were generated by lentiviral transduction. Stable JEG-3 cell lines expressing short hairpin RNAs (shRNAs) targeting either ATP6V0C (shATP6V0C), UBA5 (shUBA5) or UFM1 (shUFM_1 and shUFM1_2), as well as a non-targeting control (shNT), were also generated by lentiviral transduction. JEG-3 cells (30,000 cells per 500 μL per well in a 24-well plate) were transduced with lentiviruses encoding the respective shRNAs at a multiplicity of infection (MOI) of 3. Twenty-four hours post-transduction, the medium was replaced with fresh complete RPMI supplemented with 1 μg/mL puromycin (Sigma-Aldrich). Three days later, once all cells in the non-transduced control well were eliminated, puromycin-resistant cells were expanded. To generate JEG-3_shUBA5 cells reconstituted with shRNA-resistant FLAG-tagged UBA5 (JEG-3_shUBA5 + ^FLAG^UBA5), lentiviral transduction was performed using the corresponding overexpression construct. JEG-3 cells overexpressing FLAG-tagged mature UFM1 (JEG-3_^FLAG^UFM1ΔSC), UFMylation deficient UFM1 mutant (Mut1, JEG-3_^FLAG^UFM1ΔGSC) and de-UFMylation resistant UFM1 mutant (Mut2, JEG-3_^FLAG^UFM1G83A) were generated similarly by transducing cells with lentiviruses carrying the relevant overexpression plasmid. Knockout JEG-3 cell lines were generated using CRISPR-Cas9. Cells were transduced with lentiviruses encoding guide RNAs targeting UBA5 (JEG-3 gUBA5), a non-targeting control (JEG-3 gNT). Following knockout validation by western blotting, single-cell clones were obtained by serial dilution in 96-well

plates. Clonal populations were monitored for two weeks and subsequently expanded.

## Lentivirus production

Lentiviral particles were produced by co-transfecting HEK293T cells with the appropriate pLKO-, pWPI-, or pLentiCRISPRv2-based transfer plasmids and packaging vectors (pMD2-VSV-G and either psPAX2 or pCMV-GagPol). Subconfluent monolayers of HEK293T cells were seeded in 6-well plates or 10-cm² dishes, and co-transfected with the packaging plasmids pMD2-VSV-G and psPAX2 (for shRNA transfer plasmids) or pCMV-GagPol (for overexpression plasmids), and the transfer plasmid encoding the gene of interest. Plasmids were mixed at a 1:0.5:1.5 ratio (transfer:packaging:envelope), and polyethylenimine (PEI) was added at a PEI:DNA ratio of 3:1. The DNA–PEI complexes were incubated at room temperature for 20 min before being added dropwise to cells. Culture media were replaced 16 h post-transfection. Lentivirus-containing supernatants were collected at 48, 56, and 72 h post-transfection, pooled, 0.45 μm-filtered and stored at −80 °C until use. Viral titers were estimated by colony-forming unit (CFU) assay on HeLa cells as previously described[69].

## Lentiviral shRNA Library Production and shRNA Screen

Three different shRNAs targeting each of the 58 MS hits and 4 controls (non-targeting shRNA-encoding plasmids SHC001 and SHC002–shNT-1 and shNT-2, and plasmids encoding shRNA targeting the known host dependency factor ATP6V0C–ATP6V0C-1 and ATP6V0C-2) were selected from the MISSION TRC lentiviral library (Sigma-Aldrich; a total of 178 shRNAs; Supplementary Data 1). Lentiviruses encoding individual shRNAs were produced in HEK293T cells by co-transfection with the packaging plasmids psPAX2 and pMD2-VSV-G, and the shRNA-encoding pLKO plasmids. Lentiviruses were harvested at 48, 56, and 72 h post-transfection. For quality control, ~25% of the shRNA library was randomly selected to measure lentiviral titers. JEG-3 cells were seeded at 20,000 cells in 500 μL per well in 24-well plates for the screen. Cells were transduced with lentiviruses (MOI ≈ 3; one shRNA/well) in triplicate wells and three independent biological replicates. Three days post-transduction, cells were infected with the ZIKV H/PF/2013 reporter strain expressing *Renilla* luciferase (ZIKV-R2A, MOI = 0.1). Forty-eight hours post-infection, wells were washed once with PBS and luciferase activity measured as previously described[70]. Luminescence values for the target shRNAs were normalized to those of the non-targeting shRNAs, and hits were selected based on the criteria of a 50% difference in viral replication with two out of three shRNAs or >75% difference with one out of three shRNAs. Cell viability was assessed by a resazurin-based cell viability assay at 72 h post-transduction on plates transduced with lentiviruses in parallel. Individual shRNAs were considered toxic if the cell viability (% of shNT-1 or shNT-2) was ≤ 75%.

## Virus stock production

Stocks of dengue virus (strain UVE/DENV-2/2018/RE/47099), Japanese encephalitis virus (strain UVE/JEV/2009/LA/CNS769), West Nile virus (UVE/WNV/UNK/CF/Ar B 3573/82), Yellow fever virus (strain Asibi), Zika virus (strain H/PF/2013), and herpes simplex virus 1 (strain F) were produced in Vero E6 cells. Stocks of ZIKV H/PF/2013 reporter strains, either expressing *Renilla* luciferase (ZIKV-R2A), carrying an HA epitope within NS4B (ZIKV-NS4B^HA^) were produced by electroporating VeroE6 cells with in vitro-transcribed viral RNA. In both cases, culture medium was replaced the day after infection or electroporation with DMEM supplemented with 2% fetal bovine serum (FBS) and 15 mM HEPES (no HEPES for HSV-1). Supernatants were collected 4–6 days post-infection, clarified through 0.45 μm filters, aliquoted, and stored at −80 °C. Viral titers of infectious clones were determined by plaque assay; while

titers of reporter viruses were determined by 50% tissue culture infectious dose (TCID$_{50}$) assay as previously described[70].

## In vitro transcription (IVT) and electroporation of RNA

Plasmid DNA (10 µg) encoding the transcript was linearized with XhoI (New England Biolabs) and purified using the phenol–chloroform extraction method. The ethanol-precipitated DNA was air-dried and resuspended in nuclease-free water. In vitro transcription was carried out using the RiboMAX™ Large Scale RNA Production System-T7 (Promega) with linearized plasmid DNA as template. Reactions (100 µL) were assembled with 3.125 mM each of rATP, rCTP, and rUTP; 0.625 mM rGTP; 1X T7 transcription buffer; 1:10 diluted T7 enzyme mix; and 1 mM Anti-Reverse Cap Analog (ARCA; New England Biolabs), and incubated for 30 min at 37 °C. Subsequently, 2.5 mM rGTP was added, and the reaction was incubated for an additional 6 h at 37 °C. Transcription was terminated by treatment with 10 U RQ1 RNase-free DNase (Promega) for 30 min at 37 °C. RNA was extracted with acidic phenol–chloroform, precipitated with isopropanol, and resuspended in nuclease-free water. RNA concentration was determined using a Nanodrop, and integrity was assessed by agarose gel electrophoresis.

For electroporation of ZIKV full-length RNA (ZIKV H/PF/2013 reporter strain expressing Renilla luciferase) and subgenomic replicons into Huh7 (or Vero E6 cells for virus stock production), cells were resuspended at a density of $1 \times 10^7$ cells/mL ($1.5 \times 10^7$ cells/mL for Vero E6 cells) in Cytomix buffer (120 mM KCl, 0.15 mM CaCl$_2$, 10 mM potassium phosphate buffer pH 7.6, 25 mM HEPES pH 7.6, 2 mM EGTA, 5 mM MgCl$_2$) supplemented with 2 mM ATP and 5 mM glutathione. For each reaction, 200 µL of resuspended cells were mixed with 5 µg RNA (400 µL cells and 10 µg RNA for virus stock production), and electroporated using a Gene Pulser system (Bio-Rad) in a 0.2-cm gap cuvette at 975 µF and 270 V (0.4-cm cuvette for 400 µL volume). Cells were immediately transferred to 6 mL complete DMEM, and 500 µL of the suspension was seeded per well in a 24-well plate. For virus stock production, cells were pooled from two electroporations into 25 mL DMEM and seeded into a 15-cm dish.

## Virus titrations

Plaque assays were performed by seeding Vero E6 cells ($2 \times 10^5$ cells in 500 µL per well) in 24-well plates 24 h prior to infection. Confluent monolayers were infected with serial tenfold dilutions of virus-containing supernatants and incubated for 1 h at 37 °C. Following infection, the inoculum was removed and replaced with Minimum Essential Medium (MEM; Gibco) supplemented with 2% FBS and 1.5% carboxymethylcellulose (Sigma-Aldrich). At 4 days post-infection, cells were fixed with 5% formaldehyde for 30 min at room temperature, washed, and stained with 1% crystal violet solution (in PBS with 20% ethanol) for 30 min. After rinsing with water, plaques were counted, and viral titres were calculated as plaque-forming units (PFU) per millilitre.

Antibody-based TCID$_{50}$ (50% tissue culture infectious dose) assays were performed by seeding Vero E6 cells ($2.5 \times 10^4$ cells in 180 µL per well) in 96-well plates 24 h prior to infection. Tenfold serial dilutions of virus supernatants were prepared in-plate (six replicates per dilution). At 4 days post-infection, cells were fixed with 5% formaldehyde for 30 min at room temperature, washed, and permeabilised with 0.5% Triton X-100 in PBS for 10 min. Cells were then blocked with 2.5% milk in PBS containing 0.25% Tween-20 (PBS-T) for 30 min at room temperature. After a brief wash, cells were incubated with a protein A–purified anti-orthoflavivirus envelope protein monoclonal antibody (clone D1-4G2-4-15; 1:250 dilution in PBS; in-house, stock concentration 500 ng/µL) for 2 h at room temperature, followed by incubation with a secondary anti-mouse HRP-conjugated antibody (1:200 dilution in PBS) for 1 h. Foci were visualized by adding carbazole solution and titers calculated using the Spearman–Kaerber method as previously described[71].

## Proteomics sample preparation, data analysis and processing

Confluent monolayers of JEG-3 cells stably expressing HA-tagged 2k-NS4B of each individual orthoflavivirus or controls (n = 4 biological replicates/sample) were collected and processed for HA-immunoprecipitation using anti-HA–conjugated agarose beads (Sigma-Aldrich) as described before[14]. Eluates from affinity-purified complexes (interactomes) and whole-cell lysates (effectomes; 50 µg total) were denatured by incubation in 40 µL U/T buffer (6 M Urea, 2 M Thiourea, 100 mM Tris-HCl, pH 8.5), and reduction and alkylation carried out with 10 mM DTT and 55 mM iodoacetamide in 50 mM ABC buffer (50 mM NH$_4$HCO$_3$ in water pH 8.0), respectively. After digestion with LysC (WAKO Chemicals USA) (IPs: 0.5 µg; whole-cell lysates: 1 µg) at room temperature for 3 h, the suspension was diluted in ABC buffer, and the protein solution was digested with 1 µg trypsin (Promega)(IPs: 0.5 µg; whole-cell lysates: 1 µg) overnight at room temperature. Digestion was stopped with 1% TFA, and peptides were purified on StageTips containing three C18 Empore filter discs (3 M) and analyzed by liquid chromatography coupled to tandem mass spectrometry on a timsTOF Pro (Bruker).

## Ultra-high-performance liquid chromatography and trapped ion mobility spectrometry quadrupole time of flight settings

Samples were analysed on a nanoElute (plug-in v.1.1.0.27; Bruker) coupled to a trapped ion mobility spectrometry quadrupole time of flight (timsTOF Pro) (Bruker) equipped with a CaptiveSpray source. Peptides (500 ng) were injected into a Trap cartridge (5 mm × 300 µm, 5 µm C18; Thermo Fisher Scientific) and next separated on a 25 cm × 75 µm analytical column, 1.6 µm C18 beads with a packed emitter tip (IonOpticks). The column temperature was maintained at 50 °C using an integrated column oven (Sonation GmbH). The column was equilibrated using 4 column volumes before loading samples in 100% buffer A (99.9% Milli-Q water, 0.1% formic acid (FA)). Samples were separated at 400 nl min$^{-1}$ using a linear gradient from 2 to 17% buffer B (99.9% ACN, 0.1% FA) over 60 min before ramping up to 25% (30 min), 37% (10 min) and 95% of buffer B (10 min) and sustained for 10 min (total separation method time, 120 min). The timsTOF Pro was operated in parallel accumulation-serial fragmentation (PASEF) mode using Compass Hystar v.5.0.36.0. Settings were as follows: mass range 100–1700 m/z, 1/K0 start 0.6 V·s/cm2End 1.6 V·s/cm2; ramp time 110.1 ms; lock duty cycle to 100%; capillary voltage 1600 V; dry gas 3 l min−1; dry temperature 180 °C. The PASEF settings were: 10 tandem mass spectrometry (MS) scans (total cycle time, 1.27 s); charge range 0-5; active exclusion for 0.4 min; scheduling target intensity 10,000; intensity threshold 2500; collision-induced dissociation energy 42 eV.

## Raw data processing and analysis

Raw MS data were processed with the MaxQuant software v.1.6.17 using the built-in Andromeda search engine to search against the human proteome (UniprotKB, release 2019_10) containing forward and reverse sequences concatenated with the individual viral open reading frames of orthoflavivirus 2k-NS4B or control baits manually annotated, and searched with the label-free quantitation (LFQ) algorithm. Additionally, the intensity-based absolute quantification (iBAQ) algorithm and match between runs option were used. In MaxQuant, carbamidomethylation was set as fixed and methionine oxidation and N-acetylation as variable modifications. Search peptide tolerance was set at 70 p.p.m. and the main search was set at 30 p.p.m. (other settings left as default). Experiment type was set as TIMS-DDA with no modification to the default settings. Search results were filtered with a false discovery rate of 0.01 for peptide and protein identification. Perseus software versions v.1.6.10.4 and v.1.6.15.0 were used to process the data further. Protein tables were filtered to eliminate the identifications from the reverse database and common contaminants. When analyzing the MS data, only proteins identified on the basis of at least one peptide and a minimum of three quantitation events in at least one

experimental group were considered. The iBAQ protein intensity values were normalized against the median intensity of each sample (using only peptides with recorded intensity values across all samples and biological replicates) and log-transformed; missing values were filled by imputation with random numbers drawn from a normal distribution calculated for each sample (width 0.3, down shift 1.8). Principal component analysis was used to identify and remove outliers. In total, three individual samples were removed from the NS4Bome (JEV_3, POWV_2 and WNV_1), and the Effectome (Gluc_3, JEV_4 and YFV_3).

Significantly up- or down-regulated proteins within NS4B-expressing cell lines (Effectome) were identified using a two-sided FDR-adjusted (n = 250) Student's t-test (|log2(Fold-change)|>2; $p$ value ≤ 0.05; S0 = 1, FDR ≤ 0.05) using *Gaussia* luciferase as control (Supplementary Data 1).

Significant protein-protein interactions for each bait were identified using a two-sided FDR-adjusted (n = 250) Welch's t-test (*P* value ≤ 0.05; S0 = 4, FDR ≤ 0.01) using *Gaussia* luciferase as control (Supplementary Data 2). Only proteins measured in at least 3 biological replicates of each bait were considered. Furthermore, inspection of unimputed intensity profiles of all significant interactors across control baits (ZIKV-capsid, HCV-NS4B and Mock) was used to monitor for subcellular specificity and organellar selectivity.

For identification of high-confidence *core* interactors of ortho-flavivirus NS4B, only proteins displaying a log2 (fold-change) ≥5 for each bait were considered. To identify shared host proteins among different NS4Bs in the extended NS4Bome network (Fig. 1b) and all intersection and functional enrichment analysis (Figs. 2, 3, Supplementary Fig. 1, 2), the high-confidence network was relaxed to include significantly enriched proteins across other NS4B interactors (log2 (fold-change) ≥ 2).

To identify selective, virus-specific host interacting proteins (Fig. 2d–g), the multi-volcano Hawaii analysis available within Perseus, setting all the other baits as background control. The s0 and FDR parameters of the multi-volcano analysis for Class A (higher confidence, s0 = 1, FDR = 0.01) and Class B (lower confidence, s0 = 1, FDR = 0.05) were chosen by visual inspection, aiming for a low number of significantly depleted proteins in any of the experiments (Supplementary Data 1).

For Fisher's exact tests, annotations were extracted from Uni-ProtKB, Gene Ontology (GO), the Kyoto Encyclopedia of Genes and Genomes (KEGG) and CORUM (Supplementary Fig. 2d). For functional enrichment of cellular processes and pathways within the "core" NS4Bome (Supplementary Fig. 2a) STRING network analysis and gene ontology analysis were performed using STRING v11.0[72]. For functional enrichment of cellular processes and pathways within the extended NS4Bome (Fig. 2b, c) GO analysis was performed using Metascape[73]. Complete enrichment analyses are available in Source Data and representative categories are visualized.

## Oxygen consumption rate measurements

The day before the assay, Seahorse XF 96-well sensor cartridges (Agilent) were hydrated overnight at 37 °C in a non-CO₂ incubator using 200 μL of Seahorse XF calibrant (Agilent) per well. Simultaneously, UBA5 knockout JEG-3 (JEG-3_gUBA5) and control JEG-3_gNT cells were seeded at a density of 50,000 cells per well into a Seahorse XF 96-well cell culture microplate (Agilent) and incubated overnight at 37 °C in a 5% CO₂ atmosphere. To evaluate the effect of ZIKV infection on mitochondrial respiration, infected and uninfected JEG-3 cells (50,000 cells/well) were similarly seeded in the Seahorse XF microplate and incubated overnight at 37 °C with 5% CO₂. For experiments involving the UBA inhibitor DKM 2-93, JEG-3 cells were seeded at a lower density of 20,000 cells per well, three days before the assay. One day after seeding, the cells were treated with either 32 μM DKM 2-93 or DMSO (control) and incubated for 48 h under the same conditions. At least 5

replicates per condition were included in each experiment. On the day of the assay, the cells were washed with 200 μL of Seahorse XF RPMI medium supplemented with 1 mM pyruvate, 2 mM glutamine, and 10 mM glucose. Washed cells were then incubated in 180 μL of the supplemented Seahorse XF RPMI medium for 1 h at 37 °C in a non-CO₂ incubator. Oxygen Consumption Rates (OCR) was measured using the Seahorse XF Cell Mito Stress Test kit (Agilent) following the manufacturer's protocol. Briefly, 20 μL of 10 mM oligomycin, 22 μL of 10 mM FCCP, and 25 μL of 5 mM rotenone/antimycin A were loaded into the designated injection ports of the hydrated sensor cartridges. Sensor cartridges were then positioned onto the Seahorse XF 96-well cell culture microplate. Sequential drug addition (10-fold dilution) and time-lapse OCR measurements were carried out using a Seahorse XFe96 analyzer (Agilent). OCR data were analyzed using Wave 2.6.1 software (Agilent) to calculate basal respiration, ATP-linked respiration, and maximal respiration. For each experiment, the values were first normalized to total mean protein concentration, measured with Bradford assay on all replicates (μg per condition), and then to the mean basal OCR of the control condition from each independent experiment.

## Mitochondria morphology analysis

JEG-3 mitochondria were labelled in standard immunofluorescence using anti-COX-IV primary antibodies (Cell Signaling, #4850S). Mitochondria images were acquired using an LSM780 confocal microscope (Carl Zeiss Microimaging) at the Confocal Microscopy Core Facility of INRS, and subsequently analyzed using the Mitochondria Analyzer plugin in ImageJ[74]. For images containing multiple cells, individual cell areas were first selected using the ROI Manager. The single channel containing the mitochondrial staining was then selected and converted into a binary image. Thresholding was performed using the 2D Threshold option in the Mitochondria Analyzer Manager. For each experiment, the thresholding method and C-value were previously optimized. After thresholding, images were analyzed using the 2D Analysis option. For statistical analysis, all values were processed using GraphPad Prism software. Normality was first assessed for each dataset, in which case an unpaired t-test or one-way ANOVA was used. Alternatively, the Mann-Whitney test or Kruskal-Wallis test was applied.

## Zebrafish infection experiments

All the zebrafish experiments were performed in compliance with the guidelines of the Canadian Council for Animal Care (CCAC) and under the approval of institutional ethic committee ("Comité institutionnel de protection des animaux" (CIPA); certificate number: #2005-01) and biosafety committee ("Comité de biosécurité"; certificate number: #2016-12) of Institut National de la Recherche Scientifique (INRS).

Adult zebrafish (*Danio rerio*, TL fish line) were maintained at 28 °C with a 12 h/12 h light/dark cycle in the aquatic facility of the National Laboratory of Experimental Biology at INRS. Fertilized eggs were collected and maintained in petri dishes containing E3 medium at 28.5 °C. At the 2–4 cell stage, embryos were injected with 4 nL containing ZIKV H/PF/2013 particles (~100 PFU in total) or DMEM (vehicle) using the FemtoJet 4i microinjection system (Eppendorf). At 4 h post-infection, embryos were treated with 20 μM DKM 2-93, 100 μM NITD008 (Tocris Small Molecules) or DMSO diluted in E3 medium. Two days post-infection, embryos were dechorionated with forceps if unhatched, then fixed overnight at 4 °C with 4% formaldehyde in PBS. Morphological assessment was performed using a Stemi 305 microscope (Zeiss) as described before[48]. Phenotypes were classified according to the severity of several morphological features (e.g., curved tail, edema, head size and shape), exactly as we previously described[48]. For viral RNA absolute quantification, on the second day post-infection, embryos were pooled by condition, ensuring equal numbers of animals per experimental group (6-10 embryos per group). RNA was

extracted from whole non-fixed larvae using the RNeasy Mini Kit (Qiagen) following the manufacturer's instructions. For cDNA synthesis, 500 ng of RNA was used with the QuantiTect Reverse Transcription Kit (Qiagen). Amplification and absolute quantification of ZIKV cDNA was performd using the QX200 ddPCR EvaGreen Supermix (Bio-Rad) on the QX200 ddPCR System (Bio-Rad). The primers specific to ZIKV H/PF/2013 sequence were 5′-AGATGAACTGATGGCCGGGC-3′ and 5′-AGGTCCCTTCTGTGGAAA TA-3′.

## RNA-seq analysis

Total RNA was isolated from JEG-3 cells using TRIzol (Invitrogen) as per manufacturer instructions. High RNA integrity was confirmed by applying the Bioanalyzer Total RNA 6000 Nano Assay (Agilent, 5067-1511). Per sample, 1 µg total RNA (quantified by Qubit RNA HS Assay, Thermo Fisher, Q32852) was pretreated applying the NEBNext Poly(A) mRNA Magnetic Isolation Module (New England Biolabs) and further processed via the CORALL Total RNA-Seq Library Prep Kit (Lexogen, user guide version 095UG190V0130) in combination with Unique Dual Indexing Add-on Set B1 (Lexogen) according to manufacturer's instructions (applying 16 cycles of Library Amplification PCR, step 4.3, user guide version 117UG228V0201). Libraries were quality controlled on a TapeStation D1000 Assay (Agilent, D1000 ScreenTape 5067-5582 with D1000 Reagents 5067-5583) and were sequenced on an Illumina NextSeq 2000 instrument (NextSeq 2000 P3 Reagents (200 Cycles), product number: 20040560) in a paired-end mode ($2 \times 10^7$ bp). After demultiplexing via bcl2fastq, 13.5 – 23.5 million read pairs assigned to each sample (compare table Supplementary Data 1). All samples passed quality control by fastqc (Andrews S. (2010). FastQC: A Quality Control Tool for High Throughput Sequence Data. http://www.bioinformatics.babraham.ac.uk/projects/fastqc/) and multiqc[75] and were subjected to downstream analysis.

Gene abundance for each sample was quantified with human gene annotations from gencode (version 43) for GRCh38 genome assembly using salmon[76] (v1.10.0), and imported using R package tximport[77]. Counts normalization and multi-factor differential expression (DE) analysis was performed using DESeq2 package[78]. Replicates were evaluated based on PCA plot and outliers were removed. Design matrix with interaction term "~ group + condition + group:condition" for multi-factor DE analysis, where condition refers to (CT/shNT1 and SH/shUBA5, 5 sample in control vs 4 in treated) and group refers to the sub-group in each of the conditions (mock and Infected) was used as experiment design for DESeq2. Null variance of Wald test statistic output by DESeq2 was re-estimated using R package fdrtool[79] to calculate *p* values (and adjusted using Benjamini-Hochburg method) for the final list of differentially expressed gene. FDR (BH-adjusted *p* values) <0.1 was used a criterion for the final DE gene list. Volcano plots were created using EnhancedVolcano R package (Blighe K, Rana S, Lewis M (2024). EnhancedVolcano: Publication-ready volcano plots with enhanced colouring and labeling. R package version 1.24.0, https://github.com/kevinblighe/EnhancedVolcano) and Gene ontology enrichment analysis was performed using R package clusterProfiler[80]. To find overlapping interferon gene signatures in shNT1 and shUBA5 in infection condition, samples were compared with their respective controls using design matrix "~ condition". R statistical computing language (version 4.3.3) was used to perform above mentioned steps, except gene quantification.

## Anisomycin treatment and cell fractionation by sequential detergent extraction

Crude extracts of JEG-3 cells were prepared as recently described[81]. On the day of the treatment, the spent media in the cell culture dish was replaced with fresh media containing 200 nM Anisomycin (ANS, Sigma-Aldrich). ANS is an elongation inhibitor, that at substoichiometric concentrations, stochastically stalls a subset of ribosomes causing uninhibited upstream ribosomes to collide with them,

triggering Ribosome-associated Quality Control (RQC). Following the addition of ANS, the cells were incubated for 20 min at 37 °C to induce RPL26 UFMylation. Post incubation, the cells were harvest by trypsinisation, washed once with ice cold PBS and resuspended in fractionation buffer (50 mM HEPES pH7.5 and 150 mM NaCl; 600 µL of the buffer was used for JEG-3 cells harvested from one confluent 10-cm cell culture dish) freshly supplemented with protease inhibitor cocktail. One-third of the resuspended cells were incubated on ice for 10 min with 1% NP40 followed by centrifugation at 20,000 × g to obtain the whole cell lysate (WCL) fraction. The remaining two-thirds were resuspended in the fractionation buffer containing 0.005% digitonin for 5 min to permeabilize the plasma membrane, followed by centrifugation at 8000 × g for 5 min. The supernatant containing the cytosolic fraction was collected (Cyto), and the pellet was washed once with 500 µL fractionation buffer with no detergents before resuspending in 100 µL fractionation buffer containing 1% NP40. After incubating for 10 min on ice and centrifugation at 20,000 × g for 10 min, the supernatant containing the membrane fraction (ER) was collected. Equal volumes of collected fractions were analyzed by SDS-PAGE and immunoblotted with anti-UFM1 antibodies.

## Luciferase Assay

Luciferase assay was used to measure viral replication in cells infected with the ZIKV H/PF/2013 reporter strain expressing Renilla luciferase (ZIKV-R2A) or electroporated with full-length viral RNA or subgenomic replicons. On the day of harvest, cells were washed once with PBS, lysed in luciferase lysis buffer (1% Triton-X-100, 25 mM Glycyl-Glycine pH7.8, 15 mM MgSO4, 4 mM EGTA, 1 mM DTT added directly prior to use), and plates were frozen at −20 °C until analysis. Plates were eventually thawed on ice and luciferase activity was measured using a M200 Tecan plate reader (Tecan Life Sciences) which was programmed to inject 80 µL of luciferase assay buffer (25 mM glycylglycine, pH 7.8, 15 mM KPO4 buffer pH7.8, 15 mM MgSO4, 4 mM EGTA, 1 mM coelenterazine freshly added before the assay) into each well. The luciferase activity was then measured using default settings.

## Western blotting

Protein samples were harvested in RIPA buffer, and protein concentrations were determined using the Pierce 660 nm Protein Assay kit (Pierce). Normalized samples were denatured in 1X sample buffer (200 mM Tris, pH 8.8, 5 mM EDTA, 0.1% bromophenol blue, 10% sucrose, 3% SDS, and 1 mM DTT) for 5 min at 95 °C. Samples were loaded onto 10%, 12%, or 15% SDS-polyacrylamide gels. After electrophoresis, proteins were transferred to a nitrocellulose membrane using a Mini Trans-Blot Cell wet-blotting apparatus (Bio-Rad). The membrane was blocked with 5% non-fat dry milk in PBST (0.5% Tween-20 in 1X PBS) and incubated with primary antibodies overnight at 4 °C (or for 1 h at room temperature in the case of HRP-conjugated primary antibodies). After three washes with PBST, the membrane was incubated with HRP-conjugated secondary antibodies for 1 h at room temperature. The membrane was developed using Western Lightning Plus-ECL reagent (Perkin Elmer) and imaged using an Intas Chemocam Imager 3.2 (Intas).

## Co-immunoprecipitation

Frozen cell pellets were thawed on ice and incubated with lysis buffer (0.5% NP-40, 150 mM NaCl, 50 mM Tris-Cl pH 8.0, 1X cOmplete protease inhibitor cocktail (Roche)) for 30 min on ice. All subsequent steps were performed at 4 °C unless otherwise stated. The lysates were centrifuged for 15 min at 20,000 × g. The protein concentration was determined, and the normalized lysates were incubated with pre-cleared beads for 4 h. After incubation, the beads were washed three times with lysis buffer and three times with wash buffer (150 mM NaCl and 50 mM Tris-Cl pH 8.0). Immunocomplexes were eluted in 2X sample buffer at room temperature and analyzed by Western blotting.

## Immunofluorescence microscopy

Cells grown on coverslips for immunofluorescence analysis were fixed with 4% paraformaldehyde (PFA) for 20 min at room temperature. The fixed cells were permeabilized with either 0.2% or 0.5% (v/v) Triton X-100 in 1X PBS for 10–15 min and blocked with 5% milk in 1X PBS-Tween20 (0.25%) or 5% bovine serum albumin (BSA) and 10% goat serum (Thermo-Fisher) for 30–60 min at room temperature. Blocked coverslips were incubated with primary antibodies for 2 h at room temperature, followed by incubation with Alexa Fluor–conjugated secondary antibodies. Nuclear DNA was stained with 4',6-diamidino-2-phenylindole (DAPI, Molecular Probes). Coverslips were mounted using Fluoromount-G mounting medium (Southern Biotechnology Associates), and cells were visualized using a ECLIPSE Ti2-E Inverted Microscope with Yokogawa CSU-W1 Borealis Spinning Disk Confocal or an LSM780 confocal microscope (Carl Zeiss Microimaging). Images were analyzed using Fiji/ImageJ software[74].

## Quantitative real-time PCR

Total cellular RNA was extracted using the NucleoSpin RNA II kit (Macherey-Nagel), and cDNA was synthesized using the high-capacity cDNA reverse transcription kit (Applied Biosystems, Life Technologies), following the manufacturer's protocols. Quantitative real-time PCR (qRT-PCR) was performed on 7500 Fast Real-Time PCR System (Applied Biosystems). For each primer set, reactions were conducted in triplicate using SYBR Green master mix (A25742, Applied Biosystems) with the following primers binding to a region within the ZIKV E coding sequence: Fwd: 5'-CCG CTG CCC AAC ACA AG-3' and Rev: 5'-CCA CTA ACG TTC TTT TGC AGA CAT-3', and within the GAPDH coding sequence: Fwd: 5'-GAA GGT GAA GGT CGG AGT C-3' and Rev: 5'-GAA GAT GGT GAT GGG ATT TC-3'.

## Resazurin cell viability assay

Cell viability was assessed by adding resazurin (final concentration 0.167 mg/mL, Sigma-Aldrich) to the cells. After incubation for 1 h at 37 °C, the reduction of the non-fluorescent dye resazurin to the highly fluorescent dye resorufin, resulting from the metabolic activity of living cells, was quantified by measuring absorbance at 598 nm on a Tecan M200 plate reader (Tecan Life Sciences). Absorbance values were normalized to those of the control cells, and cells were considered viable if their absorbance was at least 75% of the control.

## Statistical analysis

Statistical analysis was performed using GraphPad Prism 10 software, unless otherwise stated.

## Reporting summary

Further information on research design is available in the Nature Portfolio Reporting Summary linked to this article.

## Data availability

The mass spectrometry proteomics data generated in this study have been deposited to the ProteomeXchange Consortium via the PRIDE[82] partner repository under accession codes PXD063343 and PXD063404. The RNA sequencing data generated in this study have been deposited in the European Nucleotide Archive (ENA) at EMBL-EBI under accession code PRJEB88617. Source data are provided with this paper.

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

## Acknowledgements

We are grateful to Alessia Ruggieri (Heidelberg University, Germany) for critical reading of the manuscript and to Subhash Vasudevan (Duke-NUS, Singapore) for scientific and experimental support. We thank all the members of the Systems Arbovirology group for helpful discussions, especially Yogy Simanjuntak for the generation of the stable NS4B-expressing cell lines and the help with the preparation of the MS samples, Charlotte Flory for help with UpSet plot visualization, Cordula Grüttner, Dennis Zorndt, Jana Holm and Emily Klaus for experimental support. We acknowledge excellent technical and bioinformatic support with RNA sequencing experiments from Patrick Blümke, Sanamjeet Virdi and Susanne Hoibian of the Technology Platform Next Generation Sequencing (TP-NGS) of the Leibniz Institute of Virology. We thank Ralf Bartenschlager (Heidelberg University, Germany) for the gift of the ZIKV molecular clones and the Vero E6 and Huh7 cell lines, and Michael Diamond (Washington University, USA) for the gift of JEG-3 cells. We acknowledge the European Virus Archive (EVAg) for sharing wild-type isolates of DENV-2, ZIKV, WNV, JEV, Jonas Schmidt-Chanasit (Bernard Nocht Institute of Tropical Medicine, Hamburg, Germany) for the YFV Asibi isolate and Wolfram Brune (Leibniz Institute of Virology, Hamburg, Germany) for the HSV-1. Work in P.S. laboratory is supported by the Free and Hanseatic City of Hamburg, the German Ministry of Research (DFG; grants number 499961789 and 528559282), the Joachim Herz Stiftung foundation (Innovate Academy), the German Research Center of Infection (DZIF; grant number TTU 01.812) and Federal Ministry of Research Technology and Space (BMFTR, VirMScan, grant number 13GW0622). P.S. is associated with the CRC1648 (DFG; grant number 512741711). L.C.-C. is receiving a research scholar (Senior) salary support from Fonds de Recherche du Québec-Santé (FRQS). The Seahorse analyzer was acquired through a young investigator infrastructure grant from the Armand-Frappier Foundation and Institut National de la Recherche Scientifique to L.C.-C. S.P. holds an FRQS Junior 2 research scholar award and the Anna Sforza Djoukhadjian Research Chair. This study was supported by the Canada Foundation for Innovation (CFI; #37512, to S.P.) and by the Canadian Institutes for Health Research (CIHR; PJT 190064) to L.C.-C. and S.P. Y.C.P.G. is supported by a Quebec-Brazil postdoctoral fellowship from Fonds de Recherche du Québec (FRQ; 2025-2026 - 2B - 370123).

## Author contributions

S.R. performed most of the formal investigation and analysis and conceptualized the study. V.A.B.T., Y.C.P.G., L.W., K.O. and Y-C.L. contributed to formal analysis, investigation, methodology development, validation, and visualization. K.O. and Y-C.L. provided methodological expertise and contributed to manuscript revisions. S.W. and S.A.P. provided resources and contributed to manuscript editing. P.S. conceptualized the study, curated data, performed formal analysis and visualization and wrote the original draft and subsequent revisions. S.R., L.C-C., also contributed to writing the original draft and revisions. P.S. and L.C.-C. acquired funding, curated data, developed methodology, supervised the project, provided resources, and contributed to writing both the original draft and subsequent revisions.

## Funding

## Competing interests

P.S. and S.R. are listed as inventors in a patent application that describe the use of UFMylation inhibitors for antiviral purposes. The other authors declare no competing interests. The funders had no role in study design, data collection and analysis, decision to publish, or preparation of the manuscript.
