## [Transparent Peer Review file · Nature Communications]

A genus-wide interaction atlas across NS4B orthologues identifies a conserved role for UFMylation in orthoflavivirus replication

Corresponding Author: Professor Pietro Scaturro

Version 0:

Reviewer comments:

Reviewer #1

(Remarks to the Author)

In this study, Rajasekharan et al. take a multiomics approach to better understand the role of the orthoflavivirus non structural protein NS4B in viral replication. By expressing NS4B of many orthoflaviviruses in JEG3 cells, authors generate a map of virus-specific and common interacting partners. To gain further insight into the role of NS4B binding, authors perform an RNAi screen in which a number of identified binding partners are depleted, and ZIKV infectivity is assayed. Authors then focus more specifically on one interacting host factor, UBA5, an E1 activating enzyme specific for UFM1. Authors find no major modifications to the UFMylation pathway upon ZIKV challenge, however do notice some relocalization of UFMylation proteins to the NS3 and NS4B positive replication sites. In further characterizing binding of UFMylation host factors to orthoflavivirus NS4B, authors find NS4B of DENV and YFV also bind UBA5, while NS4B of many other viruses bind additional UFMylation machinery. Pharmacological inhibition of UBA5 and therefore the UFMylation pathway, inhibit ZIKV infection, specifically during the virion production/egress phase. While innate immune signaling was not altered by UFMylation perturbation in ZIKV challenged cells, it was determined that UFMylation is altering mitochondrial respiration rates, and shape, which may play a role in ZIKV infection. Lastly, in zebra fish treated with an UFMylation inhibitor during ZIKV challenge, a reduction in the pathology is observed. In all, authors have identified novel NS4B binding partners, across multiple orthoflaviviruses, and provided insight into a potential druggable host factor for the therapeutic treatment of patients with orthoflavivirus infections. This study is thoughtfully designed, and conclusions are strongly supported by experimental evidence. In addition, this study is a valuable resource providing multiomics data that may prove useful to many in the orthoflavivirus field.

Major points:

- 1) A major finding of this study is the involvement of UBA5, and the UFMylation pathway, in ZIKV infection. Through a series of logical and well-controlled experiments, the authors conclude that UBA5's function appears to be related to later steps in the viral life cycle; post entry, translation and RNA replication. The most well-characterized role of UFMylation is in ribosome quality control. Flavivirus infection activates the UPR and extensively remodels the ER to stimulate RNA replication and assembly. While the replicon experiments suggest that translation and replication are not affected by knockout, it is still conceivable that there is a link between flavivirus translation at the ER and viral assembly. It is possible that polyprotein biogenesis and ER-membrane insertion requires the UFMylation machinery to deal with stalled ribosomes, possibly stalled on the large, structurally complex viral RNA. Addressing this in the discussion or, ideally, experimentally, by testing whether ribosome stalling occurs during flavivirus infection e.g. by looking at UFM1 conjugation to RPL26 in the ER fraction, would strengthen the manuscript.
- 2) The authors demonstrate a defect in mitochondrial respiration in UBA5 KO cells and show that ZIKV replication stimulates respiration. While intriguing, it remains rather speculative to claim that this correlation is relevant for the viral assembly phenotype. It is also unclear how specific interactions with NS4B would mediate this. The authors should be more nuanced in their discussion toning down some conclusions.
- 3) Authors state that ZIKV infection causes components of the UFMylation machinery to redistribute to replication sites. This is based on colocalization in infected cells using immunofluorescence. It is unclear if this is an active recruitment and if a larger proportion of these proteins are now at the ER compared to mock infected cells, or whether this simply reflects the fact that the ER-derived replication organelle takes up a larger area in the cell. This data should not be overinterpreted.

- 4) In figure 3 silencing of SEL1L resulted in an increase of viral infection and RPN2 has a variable effect. This contrasts with prior literature where they were identified as strong proviral factors (PMID: 31273220, PMID: 28720733, PMID: 33980593). Could the authors please discuss this.
- 5) The authors should mention and discuss another recent study independently identifying Ufm1ylation factors involved in flavivirus infection (PMID: 40459261).
- 6) Line 155: "suggesting an evolutionary divergence of NS4B-host-interactions within the Orthoflavivirus genus." This is an overstatement. The proteomic data uses arbitrary cutoffs, so a false negative does not mean much. Also, the affinity purifications were performed only in one cancer cell line.

Minor points:

- 1) Figure 1B is a very nice figure- it should be made larger (perhaps its own figure) so that readers can really dive into all the interactions.
- 2) The definition of "viral replication" varies among virologist- either it refers to a specific stage (copying of the RNA genome) in viral infection, or it can refer to the entire viral lifecycle. Throughout the manuscript authors use "replication" to describe the viral life cycle and to describe RNA replication. This is confusing.
- 3) Line 226: Protein levels are altered as a result of two major changes to the cell- changes to mRNA expression, or changes in degradation kinetics. Authors use the phrase "targeted for degradation" without monitoring mRNA expression or preventing protein degradation with a lysosomal or proteasomal inhibitor, and therefore this is not supported by the data.
- 4) "Virus replication" is used for multiple axis titles (figure 4b-d, and others), as well as "Virus titer" (fig. 5F). Please use consistent and more specific naming, and please refer back to minor point 2. "virus titer", "infectious viral particles", or "virions" seem like appropriate axis titles for an assay that measures PFUs. "Viral RNA" for qPCR, "luciferase activity" for luciferase assays.
- 5) Figure 4b: UBA5 addback doesn't appear to restore ZIKV replication in the RT-qPCR experiment. Could the authors please address this.
- 6) There doesn't appear to be much of a change in the number of severe pathology cases in ZIKV infected Zebra fish, with or without the UFM1ylation inhibitor, however this is never addressed.
- 7) If available, it would be great to see longer exposures for the western blots, and for the western blot sizes to be increased.
- 8) Figure 4h: With one of the biological replicates, it appears as though one of the UBA5 addbacks did not restore ZIKV replication, and is significantly different than the other two replicates. This should be repeated for an additional biological replicate.
- 9) Please make the methods section clearer. For example, I was unclear what the delta SC construct was, or how the UBA5-FLAG construct was made shRNA resistant. How was the NS4B mutant with the internal HA tag made?
- 10) Figure 5b: Western blots have been cut or boxed in a way that prevents the reader from viewing them, specifically the UBA5 and UFC1 blots.
- 11) Figure 6C: Samples from the drug treated cells are not loaded equally to the DMSO treated cells. Please load equal amounts to make conclusions about NS1 expression and viral inhibition under drug treatment.
- 12) Figure 6d: There could be a concern that the drugs are still present in the harvested supernatant, but this depends on the specific dilutions used for the plaque assay. Please comment on the dilutions.
- 13) Figure 7a: please discuss in greater detail in manuscript.

Reviewer #2

(Remarks to the Author)

Reviewer #3

(Remarks to the Author)

This study by Rajasekharan et al investigates the role of the UFM1ylation pathway in flavivirus restriction using a combination of genetic knockout, chemical inhibition, and in vivo validation approaches. The authors propose that UFM1ylation functions as host dependency pathway for these viruses and characterize the small molecule UBA5 inhibitor DKM 2-93 as a potential therapeutic compound. While the proteomics in this study are rigorously performed, the mechanism of UFM1ylation's role in NS4B-dependent virus replication is less convincing. Some of the concerns are outlined below:

General comments:

1. The primary weakness in this style of proteomics is the reliance on viral protein overexpression studies conducted outside the context of infection. This approach has fundamental limitations for flavivirus research, as demonstrated by the poor reproducibility between studies examining NS4B interactions through overexpression approaches (PMID: 30037810, PMID: 28317380, PMID: 30177828). Multiple studies have now shown that NS4B interaction partners differ substantially when the protein is expressed alone versus in the presence of other viral proteins such as NS4A (PMID: 27524440, PMID: 29902443).

A better experimental design would involve proteomics analysis in the context of authentic viral infection, as demonstrated in previous studies (PMID: 25926641; bioRxiv doi: 10.1101/2024.04.03.587877). The authors should compare their NS4B interaction data with findings from these infection-based studies and their own previous work (PMID: 30177828) to assess the degree of overlap and discuss potential differences. If UFM1ylation machinery was not identified in previous infection-based studies, this could indicate that the interactions observed here are cell type specific effects or potential artifacts of the overexpression system.

2. Another critical gap in the manuscript is the lack of clarity regarding the mechanism by which UFMylation supports viral replication. Is NS4B itself modified by UFMylation, or does the interaction with UBA5 facilitate modification of other viral or cellular proteins? How do UFMylation substrates change during infection? The authors propose that UFMylation enhances mitochondrial respiration to support viral replication, but this hypothesis requires more direct experimental validation through alternative approaches to modulate respiration. Additionally, if mitochondrial function is the key mechanism, the authors should explain why NS4B specifically needs to interact with UBA5 and whether this interaction occurs at mitochondrial membranes.

3. The findings would be significantly strengthened by validation in multiple cell types beyond JEG3 cells, particularly given the potential for cell-type-specific UFMylation interactions. The restriction to a single cell type limits the generalizability of the conclusions and raises questions about the broader relevance of the UFMylation-flavivirus interaction.

Specific comments:

Figure 1 and 2: Key validation data are missing regarding NS4B expression levels and subcellular localization. The authors should show that expressed NS4B achieves physiologically relevant expression levels comparable to those observed during infection and maintains proper subcellular distribution. The resolution of Supplementary Figure 1 is insufficient to assess localization patterns. Additionally, the temporal dynamics of the NS4B-UBA5 interaction during the viral lifecycle are not characterized.

Figure 4: In Figure 4a-c, the extent of both the reduction and the restoration of viral titers are very modest. The level of NS2B in the UBA5-depleted samples show minimal reduction in NS2B levels and that does not change in the reconstituted samples. Similarly, the infectious virus production is between 2-4 fold and rescue seems to be partial. The authors should move this to the supplementary section, since the KO data (Figure 4h) seems more convincing. What is also puzzling is that while reconstitution of the WT UBA5 in the depleted (shRNA) and deleted (gRNA) background seem equivalent (Figure 4f), the viral titres are rescued in the gRNA background (Figure 4h) but not in the shRNA background (Figure 4b). What is the explanation for this?

The authors should determine which step in the viral lifecycle is affected by systematically measuring entry, replication and release in the KO and reconstituted cells.

Figure 5: The imaging data (panel c) lacks sufficient resolution for meaningful colocalization analysis. The authors to provide higher resolution data and quantify colocalization between UFMylation components and viral proteins. It would also be useful to see whether this is happening at mitochondrial membranes, given the effect of UFMylation on increased mitochondrial respiration.

Figure 6: While the effects of the chemical inhibitor on viral proteins and infectious particles are clear (panels 6c-d), the time-of-addition experiments show much more modest effects (<2-fold reduction compared to DMSO control). What is the explanation for this discrepancy?

Figure 7: The analysis of Figure 7a should be performed by co-staining for NS4B and UFMylation components to directly visualize their spatial relationship in the context of the proposed mechanism.

Reviewer #4

(Remarks to the Author)

This manuscript provides a comprehensive analysis of cellular proteins with viral NS4B of multiple orthoflaviviruses and incorporates information from the host factors that are perturbed during orthoflaviviral infection. They focus on the role of UBA5 and the UFMylation pathway in promoting infection, with emphasis on ZIKV, in in vitro and in vivo model systems, using genetic ablation as well as a pharmacological inhibitor. In addition, they also describe a role for UFMylation in regulating mitochondrial morphodynamics that they postulate could potentially be hijacked or mediated by viral proteins during infection. Overall, the study identifies a new role for a post-translation modification pathway during orthoflavivirus infection that could serve as a potential target for the design of pan-orthoflaviviral drugs.

The methods are well-described and the supplementary tables cover required information. While the manuscript provides sufficient evidence for what the current title describes, there are a few suggestions/ queries that, if addressed, would greatly benefit the work overall.

Major comments:

1. As the authors might already be aware, a recent article published in JVI addresses a very similar question (<https://doi.org/10.1128/jvi.00654-25>). While both manuscripts focus on the UFMylation pathway, they differ in their approaches and breadth and also their targets. The JVI manuscript would need to be cited here and this manuscript would also benefit from additional discussion from the author's perspectives on a parallel UFMylation mechanism in orthoflaviviruses.

2. Readability – the abstract and introduction are concise and well written. However, the first and second results sections require constant back and forth, especially between Figures 1b and 2a, affecting the readability of the manuscript. The first part of the results section could also benefit from some language editing to make it easier for readers to comprehend.

3. The transition from Figure 2 to Figure 3 results is a big jump, going from a pan-orthoviral proteomics study to performing functional follow-up only on ZIKV. What was the reason to choose ZIKV as the virus of choice for the mitochondrial studies? While a narrower scope for a RNAi screen is understandable on a technical level, the manuscript should give a rationale for

the choice of virus for functional follow-up. Additionally, the figure and section title should more accurately describe the scope of the work in this section; as written, it reads like an over-statement of the presented work. This is also true for the discussion of the RNAi results in the Discussion.

4. Authors haven't shown that the UBA5 inhibitor used in the study, DKM 2-93, successfully inhibits UFMylation in their experiments. Specificity of the inhibitor in affecting UFMylation downstream by lack of formation of UFM-conjugates would need to be demonstrated in both in vitro and in vivo experiments.

5. Are there other E1 enzymes known for UFMylation in humans? The UBA5 inhibitor used is effective in micromolar concentrations which is higher than what is conventionally considered 'low'. The authors do show that viability isn't affected but what could be the reason for such high amounts of inhibitor required to show an antiviral effect?

6. While the authors have shown that pharmacological inhibition of UBA5 restricts ZIKV replication in zebrafish, demonstrating that genetic ablation or UBA5 knockdown has the same effect would further substantiate their in vivo results.

Minor comments:

1. Incorrect figure cited in Line 130 (should be Supplementary Figure 1b).

2. Looks like incorrect virus may be mentioned in Line 214, (JEV instead of YFV?) based on Figure 2g.

3. Authors use the term 'replication' to denote RNA copies as well as viral titers interchangeably in the y-axes of figures and also in the text (for example – Figure 4c, g, h v/s Figure 5f). Maintaining consistency throughout the manuscript will prevent confusion.

4. The previous point is especially important in the title of Figure 6 – the title reads "Pharmacological inhibition of the UFMylation pathway impairs ZIKV replication" while 6f and 6g clearly indicate that the viral replication process is not hindered by DKM 2-93 treatment. The results section for this figure explains this clearly. Please modify this figure title.

5. Text referring to Figure 3c and Supplementary Figure 3a mentions '5 pro-viral genes' throughout the manuscript (Lines 247, 470) while the figure has 6 genes indicated. Kindly clarify.

6. Figure 1b – only the high confidence interactors have been denoted. While this could be a stylistic choice by the authors to emphasize the specific interactions, having all the interactions shown here would make the figure a lot more appealing (the empty circles, especially prominent in the "Mitochondria" bubble, don't convey much information and therefore don't substantially help the figure). Additionally, this Figure has more than 200 interactors labelled and the figure legend indicates only high confidence interactors have been denoted. Kindly clarify the discrepancy in numbers.

7. Figure 2a and Line 189 indicate different numbers of interactors.

8. Figure 4b – shUBA5 v/s exogenous supplementation does not show significant difference in viral RNA levels, while Figure 4c shows successful rescue in viral titers. Can the authors comment on this?

9. Can the authors provide a better rationale in the manuscript how they chose UBA5 from the proteomics dataset? The extended NS4B effectome and the extended NS4Bome indicate that UBA5 was only picked up in ZIKV samples (Supp Tables 1 and 2, and Figure 1b).

Reviewer #5

(Remarks to the Author)

Version 1:

Reviewer comments:

Reviewer #1

(Remarks to the Author)

The authors have satisfactorily addressed my concerns.

Reviewer #2

(Remarks to the Author)

Reviewer #3

(Remarks to the Author)

The revised version has largely addressed the concerns raised.

What is still missing from the study is that the defect in the late stage of the viral lifecycle is inferred. Assembly/release/spread is not directly measured by intracellular vs extracellular infectious units, particle:PFU ratio, intracellular E/prM accumulation with reduced release etc.

Fig 6a legend describes DKM 2-93 as “competitively” binding the catalytic cysteine. DKM 2-93 is described in the primary literature to covalently modify the catalytic cysteine of UBA5 (Cys250). That's not the same thing.

Reviewer #4

(Remarks to the Author)

The authors have largely amended their manuscript to address the reviewer's comments. However, there are a few areas that can still be improved.

1. Figure 1b is now much more interpretable and interesting to go through. This change is great.
2. While the authors have addressed the issue of readability by combining results sections 1 and 2 that makes it easier, they might want to consider rearranging their figures to make it easier for readers.
3. The justification for doing follow up studies with ZIKV is still weak. The use of JEG-3 for their APMS study is then confusing as this background is not ideal for 4/8 viruses chosen by the authors. While the experiments that follow make sense, how they address choosing ZIKV is still not convincing.
4. The authors have provided western blotting data to confirm inhibition of UFMylation pathway. This definitely helps but the use of 16uM for their experiments is tricky as it doesn't seem to inhibit the pathway as effectively as 32uM, but it does inhibit viral protein synthesis.
5. The authors have explained the difficulties with generating a UBA5 KO zebrafish model, but they say they have included these considerations in the discussion, which doesn't seem to be the case.
6. The revised manuscript largely addresses the queries of all reviewers. However, some justifications still seem like overinterpretations. I would like to ask the authors to revise the manuscript to make sure not exaggerate some of their findings.

Reviewer #5

(Remarks to the Author)

Point-by-point response to reviewers' comments

Manuscript Number: **NCOMMS-25-34251-T**

Re.: Rebuttal to reviewer's comments for

"*A genus-wide interaction atlas across NS4B orthologues identifies a conserved role for UFMylation in orthoflavivirus replication*" by Rajasekharan S. et al.

We thank the 5 reviewers for their constructive feedback and suggestions that helped to further improve the manuscript. To address their comments, we have now performed a series of additional experiments and analyses in addition to revisions of results, discussion, and methods.

These experiments include: 1) a fractionation-based approach to analyze the impact of viral infection on RPL26 UFMylation at the ER, 2) the requested additional biological repetitions of selected experiments, 3) drug-testing in additional cell lines, 4) extensive additional experiments to address the role of NS4B in cellular respiration and the differential UFMylation landscape in viral-infected cells and 5) validation of UFMylation inhibition in drug-treated cells. Just to contextualize our efforts, these new analyses also included generation and characterization of novel knock-out cell lines and over 20 hours of additional mass-spectrometry experiments, including manual inspection of MS/MS spectra of modified peptides.

We feel that the new data are highly valuable and have strengthened our manuscript, confirming our previous results. However, in our view, the new observations on differential UFMylation rather form the basis for an entirely new study, and are beyond the scope of this study which has its main focus on the characterization of the NS4B interaction landscape across multiple orthoflaviviruses and already includes:

- a) a genus-wide interaction and effectome atlas of orthoflavivirus NS4Bs,
- b) a targeted RNAi screen identifying novel host-restriction and -dependency factors,
- c) generation and characterization of a novel replication-competent ZIKV NS4B-tagged virus,
- d) validation of key protein-protein interactions in infection,
- e) detailed analysis of UFMylation pathway with respect to transcriptional landscape (RNAseq), mitochondrial morphology, cellular respiration and overall pathway activity using RNAi and CRISPR/Cas9-based systems, co-IP-WB and immunofluorescence analysis
- f) *in vitro* and *in vivo* drug-repurposing experiments demonstrating the antiviral potential of UFMylation inhibitors and characterizing the replication step(s) affected.

In light of this, we suggest not to include the new UFMylation data in this manuscript as albeit extremely interesting they require further additional validation and mechanistic experiments which will be the basis of future studies.

Please find uploaded in the online submission system the merged document highlighting the changes to the first version of the manuscript. Point-by-point responses to reviewers' comments are indicated below in dark blue.

Reviewer #1 (Remarks to the Author):

In this study, Rajasekharan et al. take a multiomics approach to better understand the role of the orthoflavivirus non structural protein NS4B in viral replication. By expressing NS4B of many orthoflaviviruses in JEG3 cells, authors generate a map of virus-specific and common interacting partners. To gain further insight into the role of NS4B binding, authors perform an RNAi screen in which a number of identified binding partners are depleted, and ZIKV infectivity is assed. Authors then focus more specifically on one interacting host factor, UBA5, an E1 activating enzyme specific for UFM1. Authors find no major modifications to the UFMylation pathway upon ZIKV challenge, however do notice some relocalization of UFMylation proteins to the NS3 and NS4B positive replication sites. In further characterizing binding of UFMylation host factors to orthoflavivirus NS4B, authors find NS4B of DENV and YFV also bind UBA5, while NS4B of many other viruses bind additional UFMylation machinery. Pharmacological inhibition of UBA5 and therefore the UFMylation pathway, inhibit ZIKV infection, specifically during the virion production/egress phase. While innate immune signaling was not altered by UFMylation perturbation in ZIKV challenged cells, it was determined that UFMylation is altering mitochondrial respiration rates, and shape, which may play a role in ZIKV infection. Lastly, in zebra fish treated with an UFMylation inhibitor during ZIKV challenge, a reduction in the pathology is observed. In all, authors have identified novel NS4B binding partners, across multiple orthoflaviviruses, and provided insight into a potential druggable host factor for the therapeutic treatment of patients with orthoflavivirus infections. This study is thoughtfully designed, and conclusions are strongly supported by experimental evidence. In addition, this study is a valuable resource providing multiomics data that may prove useful to many in the orthoflavivirus field.

We thank the reviewer for their supportive comments.

Major points:

1) A major finding of this study is the involvement of UBA5, and the UFMylation pathway, in ZIKV infection. Through a series of logical and well-controlled experiments, the authors conclude that UBA5's function appears to be related to later steps in the viral life cycle; post entry, translation and RNA replication. The most well-characterized role of UFMylation is in ribosome quality control. Flavivirus infection activates the UPR and extensively remodels the ER to stimulate RNA replication and assembly. While the replicon experiments suggest that translation and replication are not affected by knockout, it is still conceivable that there is a link between flavivirus translation at the ER and viral assembly. It is possible that polyprotein biogenesis and ER-membrane insertion requires the UFMylation machinery to deal with stalled ribosomes, possibly stalled on the large, structurally complex viral RNA. Addressing this in the discussion or, ideally, experimentally, by testing whether ribosome stalling occurs during flavivirus infection e.g. by looking at UFM1 conjugation to RPL26 in the ER fraction, would strengthen the manuscript.

We thank the reviewer for this important comment. Indeed, the potential connection between ribosome quality control (RQC) and viral morphogenesis at the ER is an interesting and relevant one which we had not yet explored. We have now performed additional experiments specifically designed to assess the levels of RPL26 UFMylation at the ER following ZIKV infection.

To do this we leveraged a recently described method combining crude ER fractionation and western-blotting with anti-UFM1-specific antibodies (PMID: 37036982). As seen below (Figure R1), isolation of crude ER fractions fully recapitulates observations from the original publication, and as expected we also observed an identical higher molecular weight ladder of UFM1-positive RPL26 conjugates as described in the original publication (PMID: 37036982). As expected, we could confirm the presence of ZIKV non-structural proteins (NS2B) only in ER, (Calnexin-positive) fractions. Importantly, to investigate the relationship between UFMylation and RQC, we additionally exposed cells to Anisomycin (ANS), a translation

elongation inhibitor that stochastically stalls a subset of ribosomes causing noninhibited upstream ribosomes on translating RNA to collide with them, triggering RQC. Also in our hands, ANS concentrations previously reported to be optimal to induce collisions on cytosolic ribosomes (200 nM, 20 min) (PMID: 37036982), stimulated an increase in UFMylated RPL26 at the ER. Conversely, ZIKV-infected cells did not display any upregulation in RPL26 UFMylation at the ER, and only a very mild increase of RPL26 UFMylation was observable on whole cell lysates upon very long exposure of the membranes.

Collectively, these data suggest that ZIKV infection does not upregulate UFMylation of ribosomes at the ER and only very mildly increase RPL26 UFMylation in whole cell lysates (visible higher molecular UFM⁺ species detectable only upon very long exposure, with levels far below those observed in ANS-treated samples). We have also included this new figure in the manuscript (Supplementary Fig. 5a; lines 314-316) as it's an important observation to share with the community in light of future studies.

Figure R1: ZIKV infection doesn't cause stalling-induced RPL26 UFMylation on ER-bound ribosomes.

JEG-3 cells were either infected with ZIKV H/PF/2013 wild-type strain (MOI 1) for 24 h or treated with 200 nM Anisomycin (equivalent concentration of DMSO as control) for 20 min prior to cell fractionation. Anisomycin treatment and cell fractionation was performed following the protocol described by Scavone *et. al.*, 2023 (PMID: 37036982). Following cell fractionation equal volumes of collected fractions were analyzed by SDS-PAGE and immunoblotted with anti-UFM1 antibodies. The absence of increased RPL26 UFMylation in the ER fraction of ZIKV infected JEG-3 cells suggests the absence of stalling-induced RPL26 UFMylation on ER-bound ribosomes. Calnexin and basal UFM1 were used as loading control for ER or Cytosolic fractions respectively. Data shown are representative of n=3 independent experiments.

2) The authors demonstrate a defect in mitochondrial respiration in UBA5 KO cells and show that ZIKV replication stimulates respiration. While intriguing, it remains rather speculative to claim that this correlation is relevant for the viral assembly phenotype. It is also unclear how specific interactions with NS4B would mediate this. The authors should be more nuanced in their discussion toning down some conclusions.

We completely agree that the causal relation between mitochondrial respiration and viral fitness in UBA5 KO cells is rather circumstantial at this stage. We have now further toned down our statements accordingly, and discuss the reciprocal effects of viral infection and

UFMylation inhibition on respiration as very interesting and worth of additional experimental investigations (lines 434, 559-560 and 562-564).

3) Authors state that ZIKV infection causes components of the UFMylation machinery to redistribute to replication sites. This is based on colocalization in infected cells using immunofluorescence. It is unclear if this is an active recruitment and if a larger proportion of these proteins are now at the ER compared to mock infected cells, or whether this simply reflects the fact that the ER-derived replication organelle takes up a larger area in the cell. This data should not be overinterpreted.

We agree that the immunofluorescence analysis as such does not allow us to rule out that redistribution of UFMylation components to replication sites in infected cells is partly driven by ER remodeling. However, we note that this hypothesis is further substantiated by additional lines of evidence, namely: a) the observation that NS4B of different orthoflaviviruses, as well as the ZIKV capsid interact with different members of the UFMylation machinery when ectopically expressed (UBA5, UFL1, UFM1, ODR4, UFSP2 and CDK5RAP3), i.e. in the absence of viral infection (Fig. 5e); b) the interacting pairs (viral or cellular) reside on both cytosolic (capsid) and ER luminal side; and c) the partial enrichment of additional members of the viral replicase machinery (NS2B-NS3) in UFM1 pull-downs (Fig. 5d). Collectively, these data point towards a recruitment or at least an enrichment of multiple UFMylation components through multiple viral proteins, independent from the massive ER remodeling observed in infected cells. However, further experiments are needed to establish the chronological and causal relations of these events, and we now further discuss this element of caution in the revised manuscript (lines 322; 559-560; 562-564).

4) In figure 3 silencing of SEL1L resulted in an increase of viral infection and RPN2 has a variable effect. This contrasts with prior literature where they were identified as strong proviral factors (PMID: 31273220, PMID: 28720733; PMID: 33980593). Could the authors please discuss this.

With respect to SEL1L, we note that only one of the cited studies reports opposite effects upon SEL1L silencing (PMID: 33980593), but these experiments were performed under substantially different conditions: a) different viral species (JEV rather than ZIKV), b) different silencing approach (CRISPR/Cas9 knock-out rather than shRNA) and c) different cellular background (HCT116 cells derived from colorectal carcinoma rather than JEG-3 cells derived from gestational choriocarcinoma). In our hands silencing of SEL1L increases ZIKV replication with 3/3 shRNAs tested in JEG-3 cells. Such differences could result from cell-type-dependent specificities or underscore viral species-specific effects (i.e. JEV vs. ZIKV). Additional experiments comparatively assessing JEV replication upon SEL1L silencing in JEG-3 cells would be required to assess the most likely cause for these differences.

Conversely, RPN2 was shown to exert a pro-viral role in Huh7 cells upon DENV infection (siRNA-, PMID: 31273220; or CRISPR/Cas9-mediated knock-out, PMID: 28720733), while we observe variable effects upon silencing with different shRNAs in JEG-3 cells upon ZIKV infection. Analogously to many other studies (and some of the ones cited above), our screening approach does not include any validation of silencing efficiency, which was performed only for host factors shortlisted for further follow-up experiments. Therefore, we cannot exclude that RPN2 might indeed exert similar pro-viral functions for ZIKV in JEG-3 cells, and the observed variability results from differences in silencing efficiencies (i.e. only 1 out of 3 shRNA specifically silenced RPN2 expression). As it did not pass our shortlisting criteria it was excluded from further analysis. We have now included this aspect in the discussion (lines 482-485).

5) The authors should mention and discuss another recent study independently identifying UFMylation factors involved in flavivirus infection (PMID: 40459261).

We thank the reviewer for this comment, indeed shortly after first submission of this manuscript, another study reported that silencing of other UFMylation components affect orthoflavivirus replication. We now included relevant discussion points and refer to this study in the revised manuscript (lines 503-513 and 581).

6) Line 155: “suggesting an evolutionary divergence of NS4B-host-interactions within the Orthoflavivirus genus.” This is an overstatement. The proteomic data uses arbitrary cutoffs, so a false negative does not mean much. Also, the affinity purifications were performed only in one cancer cell line.

We agree that any cut-off criteria are unavoidably arbitrary and the study was performed in only 1 cell line, however, we respectfully disagree on the conclusion that no species-specific differences in interactors can be drawn. In our view, the fact that the study was performed with the exact same cellular background and expression system is one of its major strengths, overcoming the inherent heterogeneity resulting from comparisons across studies (different cells lines, expression systems, epitope tags, replication kinetics, LC-MS/MS protocols, etc.). This aspect and its experimental implications are carefully addressed and discussed in the manuscript (see Fig. 2a: bait abundance vs. set size). Notably, even within the UFMylation pathway, we observe that NS4B proteins from different viral species exhibit binding specificity to different components of the same pathway (Fig. 5e), highlighting at least some degree of divergence in host-binding specificity. For these reasons we feel that “suggesting” binding differences across species is a fair statement fully substantiated by the data. However, we have now removed the wording “evolutionary” and included a further element of caution (“...*suggesting a certain degree of divergence of NS4B-host-interactions within the Orthoflavivirus genus*”) (line 157). Furthermore, we discuss the use of an individual cell line as element of caution in the context of species-specificity binding inference.

Minor points:

1) Figure 1B is a very nice figure- it should be made larger (perhaps its own figure) so that readers can really dive into all the interactions.

In response to another Reviewer’s comment, we have now increased the size and fonts of displayed nodes as much as possible. We hope that high-resolution figure on screen will now allow full exploration of the data.

2) The definition of “viral replication” varies among virologist- either it refers to a specific stage (copying of the RNA genome) in viral infection, or it can refer to the entire viral lifecycle. Throughout the manuscript authors use “replication” to describe the viral life cycle and to describe RNA replication. This is confusing.

We apologize for the confusion, following this and the comment below, we have now revised the reference in text and figures accordingly, and now refer to viral RNA (qRT-PCR), luciferase activity (luciferase assays) or virus titer (PFU or TCID50 assays).

3) Line 226: Protein levels are altered as a result of two major changes to the cell- changes to mRNA expression, or changes in degradation kinetics. Authors use the phrase “targeted for degradation” without monitoring mRNA expression or preventing protein degradation with a lysosomal or proteasomal inhibitor, and therefore this is not supported by the data.

We have now removed the reference to “targeted degradation” accordingly (line 230 and respective figure legend).

4) “Virus replication” is used for multiple axis titles (figure 4b-d, and others), as well as “Virus titer” (fig. 5F). Please use consistent and more specific naming, and please refer back to minor point 2. “virus titer”, “infectious viral particles”, or “virions” seem like appropriate axis titles for as assay that measures PFUs. “Viral RNA” for qPCR, “luciferase activity” for luciferase assays.

In line with the comment above, we have now changed axis names and text references accordingly.

5) Figure 4b: UBA5 addback doesn't appear to restore ZIKV replication in the RT-qPCR experiment. Could the authors please address this.

We think that these differences could be due to the milder effects observed on viral RNA replication when compared to effects on infectious particle production (in agreement with the proposed mechanism of inhibition at later stages of the viral replication cycle). Furthermore, as discussed in response to Reviewer #1's comment, milder effects in shRNA-based experiments might be due to the presence of leftover UBA5 in shUBA5-expressing cell lines (Fig 4f: under shRNA-mediated silencing there is still residual UBA5 activity, as inferred by the presence of UFC1-UFM1 conjugates by WB).

Under these conditions (residual endogenous UBA5 + UBA5-FLAG overexpression) there is likely an excessive amount of functional UBA5 (Fig. 4a, last lane), leading to a misbalance in the UFMylation pathway. Indeed, similar to the qRT-PCR, also rescue of viral replication or viral titers as assessed by luciferase activity and PFU assays (Fig. 4c and 4d) was milder when compared to experiments with KO cells.

Although seemingly counterintuitive, UBA5 overexpression has been reported to mimic its ablation, reducing the levels of charged UFC1 in the cell (PMID: 35806453).

In agreement with this hypothesis, UBA5 knock-out cells (no residual UBA5 or UFC1-UFM1 conjugates; Fig. 4f) displayed a much stronger viral phenotype and an overall stronger magnitude of rescue upon overexpression of UBA5 (Fig. 4h).

6) There doesn't appear to be much of a change in the number of severe pathology cases in ZIKV infected Zebra fish, with or without the UFMylation inhibitor, however this is never addressed.

We apologize for the confusion: full images of mock- or inhibitor-treated animals were originally included to actually support a lack of qualitative differences. Indeed, we observed statistically significant quantitative differences in the relative occurrence of all pathology classes (Severe cases: 22.97% -> 18.18%; Mild cases: 58.65->31.17) as well as a significant improvement in normal morphologies (28.38% -> 50.65%) and reduction in viral loads upon UFMylation inhibition. The additional panels have been removed to avoid misunderstandings (were originally included to display representative pictures across all conditions, not to support qualitative differences in the phenotypes).

7) If available, it would be great to see longer exposures for the western blots, and for the western blot sizes to be increased.

We present multiple or additional exposures when available or in many panels where we felt it was necessary to highlight weak bands or background binding. We did our best to fit available space as much as possible, and included now revised blots with higher resolution (600 dpi), but if a specific panel is particularly unclear, we would be happy to modify as suggested. Please note that all raw, uncropped blots will be available as part of the policy of the publishing group.

8) Figure 4h: With one of the biological replicates, it appears as though one of the UBA5 addbacks did not restore ZIKV replication, and is significantly different than the other two replicates. This should be repeated for an additional biological replicate.

We thank the reviewer for raising this point. We now performed the additional biological repetition as recommended. As expected, we now see more robust effects in the extent of phenotypic rescue.

9) Please make the methods section clearer. For example, I was unclear what the delta SC construct was, or how the UBA5-FLAG construct was made shRNA resistant. How was the NS4B mutant with the internal HA tag made?

We have now included detailed description of cloning strategies for the overexpression constructs and NS4B internal tagging in the material and methods section (lines 1094-1107 and 1134-1141).

10) Figure 5b: Western blots have been cut or boxed in a way that prevents the reader from viewing them, specifically the UBA5 and UFC1 blots.

We apologize for the assembly of this blot: cuts were made to optimize incubation with the high number of antibodies needed to probe for the multiple host proteins. We have now repeated this experiment one additional time, and blotted on multiple gels the same host proteins, allowing for more relaxed cropping of the membranes. For this reason, we now also included 2x GAPDH membranes as loading control for each gel (new updated Fig. 5b).

11) Figure 6C: Samples from the drug treated cells are not loaded equally to the DMSO treated cells. Please load equal amounts to make conclusions about NS1 expression and viral inhibition under drug treatment.

We have now repeated this gel, and also included UFM1 as additional loading control (updated Fig. 6C).

12) Figure 6d: There could be a concern that the drugs are still present in the harvested supernatant, but this depends on the specific dilutions used for the plaque assay. Please comment on the dilutions.

We present here on the right side the unnormalized viral titers (PFU/ml) underlying Fig. 6d. In this experiment overall viral titers were above 10^4 PFU/ml.

Under these conditions, cell culture supernatants containing virus particles (and thereby drug leftover) were diluted up to 1000-fold in the dilution where plaques are counted (equivalent to 16nM or 32nM of leftover drug). At these concentrations the drug does not exhibit any antiviral activity (Fig. 6b), and therefore we can rule out that inhibition occurs due to effects on target cells (in this case VeroE6 cells were used for titration).

13) Figure 7a: please discuss in greater detail in manuscript.

We have now added additional sentences on the interpretation of this panel in the discussion section (lines 546-551).

Reviewer #2 (Remarks to the Author):

Reviewer #3 (Remarks to the Author):

This study by Rajasekharan et al investigates the role of the UFMylation pathway in flavivirus restriction using a combination of genetic knockout, chemical inhibition, and in vivo validation approaches. The authors propose that UFMylation functions as host dependency pathway for these viruses and characterize the small molecule UBA5 inhibitor DKM 2-93 as a potential therapeutic compound. While the proteomics in this study are rigorously performed, the mechanism of UFMylation's role in NS4B-dependent virus replication is less convincing. Some of the concerns are outlined below:

General comments:

1. The primary weakness in this style of proteomics is the reliance on viral protein overexpression studies conducted outside the context of infection. This approach has fundamental limitations for flavivirus research, as demonstrated by the poor reproducibility between studies examining NS4B interactions through overexpression approaches (PMID: 30037810, PMID: 28317380, PMID: 30177828).

We acknowledge that in the ideal case, all studies should be performed under conditions fully recapitulating productive viral infections, possibly in primary cells or tissues, and that ectopic expression of viral proteins does not fully recapitulate virus-host interactions observed upon productive viral infection. However, we also note that this approach has also fundamental advantages and is the least biased for comparative studies aiming to dissect PPI across multiple viral species (discussed in more details below).

With respect to the poor reproducibility across the studies mentioned above, we would like to highlight that the three studies present substantial experimental differences, far beyond the ectopic expression aspect. Specifically (order of studies cited above):

- a) considerably different LC-MS/MS set-ups with different biochemical separation methods and sensitivity (BioID, a proximity-based assay; 1DE-LC-MS/MS: SDS-PAGE based isolation; label-free AP-MS/MS: affinity-based isolation)
- b) highly diverse cellular backgrounds (293T cells, HeLa cells, SK-N-BE2 cells).
- c) different tagging strategies (BioID-tag, GST-tag, HA-tag).

While we cannot pinpoint to which extent each of these individual differences contributed to the low reproducibility across these studies, arguably all these aspects profoundly affected the quantity and the identity of the reported interactors.

Multiple studies have now shown that NS4B interaction partners differ substantially when the protein is expressed alone versus in the presence of other viral proteins such as NS4A (PMID: 27524440, PMID: 29902443). A better experimental design would involve proteomics analysis in the context of authentic viral infection, as demonstrated in previous studies (PMID: 25926641; bioRxiv doi: 10.1101/2024.04.03.587877).

With respect to the central question of our manuscript (identification of NS4B-interacting host proteins across 9 viral species), and as carefully discussed in the text (lines 86-93) we note that this presents major experimental hurdles as it would require:

- a) identification of suitable tolerated insertion sites for each of the viral species;
- b) thorough characterization of virus mutants;
- c) the availability of molecular clones for the desired species;
- d) highly comparable replication kinetics across the different viral species;
- e) a cell line equally permissive to the 9 flaviviruses object of the study;
- e) *alternatively*: availability of species-specific antibodies for affinity enrichment from wild-type-infected cells.

In agreement with these substantial challenges, to the best of our knowledge only 1 peer-reviewed study to date has reported NS4B interactions in the context of a replication-

competent NS4B-tagged virus, and this was performed on DENV-2 (16681 strain) (cited by the reviewer; PMID: 25926641, coauthored by ourselves). We were not aware of the other study cited above (bioRxiv doi: 10.1101/2024.04.03.587877), however at close inspection we note that this study presents important caveats: a) has not yet been peer-reviewed; b) does not provide easily retrievable lists of interactors for intersections, c) has been performed under conditions where different NS4B species are present at once (cells constitutively expressing TBEV NS4B, additionally infected with Langkat virus), d) has been performed in 293T cells (we have used JEG-3 cells in our study).

Collectively, in spite of the unavoidable and known limitations associated with studies using ectopic expression systems, our experimental design is the only one allowing to assess species-specific differences and similarities across NS4B-interactors independent of potential differences arising from differential replication kinetics by different viral species, as it relies on a consistent cellular background, lack of broad immune or cell-death responses and ER remodeling (infection specific, but viral protein unspecific), the same LC-MS/MS set-up and an identical enrichment strategy.

We would like to emphasize that in our study we do not simply report a list of interactors, but present extensive orthogonal validation experiments, substantiating the functional relevance of the identified interactors. These include:

- a) Engineering and characterizing a novel ZIKV clone carrying an epitope tag within NS4B (novel and not previously described; the second one of the orthoflavivirus genus available in pubmed after the DENV clone) (Supplementary Fig. 4) which was used to validate novel key interactors in virus-infected cells (Fig. 3e)
- b) Performing a custom RNAi screen on newly identified host interacting proteins, which identified 19 novel host-dependency and host-restriction factors (Fig. 3b,c)
- c) Extensive characterization of a completely new interactor (UBA5), and cellular pathway (UFMylation) using several orthogonal validation methods and model systems (PPI+WB in ectopic expression and infection, KD, KO, pharmacological inhibition, *in vitro* and *in vivo*).

The authors should compare their NS4B interaction data with findings from these infection-based studies and their own previous work (PMID: 30177828) to assess the degree of overlap and discuss potential differences. If UFMylation machinery was not identified in previous infection-based studies, this could indicate that the interactions observed here are cell type specific effects or potential artifacts of the overexpression system.

As requested by the reviewer, we present below intersections of our data with the only infection-based study which can be retrieved from Pubmed, and was performed on DENV (DENV infection in Huh7 cells, PMID: 30177828). This study identified only 20 significant interacting proteins; mostly found also in our NS4B interactome. This is not surprising as the methods used 12 years ago were by far less sensitive (the study was published in 2016 but the dataset was collected 3 years earlier). In that study we did not identify UBA5 (which we also don't identify here as a significant DENV-interactor, but report it as mildly enriched in the DENV-NS4B interactome). We also observe that other independent studies identified members of the UFMylation pathway as interactors of other viral proteins (UFL1 as ZIKV-NS4A and DENV-NS4A interacting protein in 293T cells; PMID: 30550790).

Overlap with *Chatel-Chaix L. et al. Cell Host & Microbes. 2016 – DENV-NS4B interactome in infected cells (PMID: 27545046)* (in parenthesis, binding specificities observed in our study): ATP5A1, ATP5B, CHCHD3 (WNV and YFV), CANX (DENV), GHITM (POWV, TBEV, WNV, YFV), SLC3A2 (USUV, WNV, YFV, ZIKV), ATP2A2, ATP5F1 (WNV, YFV, ZIKV), ATP5O (YFV), HM13, DHCR7 (pan-orthoflavi), ATP5H (YFV, WNV), CLTC (JEV, POWV, WNV, YFV), IMMT (WNV), HSPD1 (WNV), ATP5C1 (WNV, YFV).

Overlap with *Scaturro P. et al. Nature 2018 –ZIKV-NS4B interactome upon ectopic expression in SK-N-BE2 cells (PMID: 30177828)*:

This intersection revealed substantial overlap, with 73 proteins identified as significantly enriched in both studies, including 9 proteins which were even ultimately shortlisted for validation in our RNAi screen (STOML2, RPN2, VAPB, PHB2, SLC2A1, VDAC1, PHB, TECR, DHCR24).

These intersections, alongside intersections with other interactomes from other studies on ectopically expressed DENV-NS4B and WNV-NS4B (PMID: 30833725, PMID: 30550790) are now presented in a fully searchable format in new *Supplementary Table 5*.

Re. the interactions observed being “*cell type specific effects or potential artifacts of the overexpression system*”, we note that this can be unequivocally ruled out since:

- a) The functional relevance of UBA5 and the UFMylation pathway is extensively supported by data in the manuscript showing pro-viral phenotypes in JEG-3 (trophoblast-like), Huh7 (hepatoma-derived), MRC-5 (epithelial-derived) and whole organisms *in vivo* (zebrafish), using either shRNA, gRNA or pharmacological inhibition, unequivocally demonstrating that this is not a cell-type or overexpression-related artefact.
- b) We purposely engineered in this study a novel ZIKV molecular clone carrying an epitope tag within NS4B, and characterized its replication fitness. Using this recombinant virus, we validate interactions with 3 newly discovered host factors displaying (FPGS, HSD17B7 and UBA5) in virus-infected cells upon reciprocal pull-downs.
- c) We identify 19 host factors which exert significant pro- or anti-viral effects on viral replication.

In conclusion:

- 1) to our knowledge there are no published interactome studies performed in ZIKV-infected cells;
- 2) intersection with the only DENV-NS4B interactome performed in infected cells revealed substantial overlap;
- 3) as expected intersection with other studies using ectopic expression (by us and others) reveal both similarities and differences. As discussed above, rather than validating or invalidating either of these studies, these differences reflect fundamental heterogeneity (including differential expression patterns in different tissues, hit-calling criteria, negative controls, LC/MS instruments, etc.).

Our study has the competitive advantage of providing for the first time highly comparable experimental conditions across a large number of orthoflavivirus species, allowing to infer species-specific interaction profiles and determine much more accurately ER-related background.

2. Another critical gap in the manuscript is the lack of clarity regarding the mechanism by which UFMylation supports viral replication. Is NS4B itself modified by UFMylation, or does the interaction with UBA5 facilitate modification of other viral or cellular proteins? How do UFMylation substrates change during infection?

We agree and openly discuss in the manuscript that the exact mechanism through which UFMylation supports viral replication is still unclear and further studies are needed to identify the responsible potentially UFMylated targets (lines 559-560 and lines 590-592). However, we report here multiple novel experimentally-supported observations including:

- 1) the impact of viral infection on UFMylation activity,
- 2) multipartite interactions between UFMylation pathway members and viral proteins, and
- 3) the impact of UFMylation inhibition on global transcriptional responses, mitochondria morphology and cellular respiration. Furthermore, we demonstrate that
- 4) UFMylation plays a role at late stages of viral replication and
- 5) provide for the first time proof-of-principle that pharmacological inhibition of UFMylation has broad-spectrum antiviral potential *in vitro* and anti-ZIKV activity *in vivo*.

Nonetheless, we took very seriously this comment by performing a distinct and challenging set of experiments (Fig R2, for reviewers only). UFMylation is a notoriously labile and low-abundant post-translational modification. Indicative of these challenges and despite multiple international efforts, only a handful of cellular targets have been reported to date as bona fide UFMylated substrates (reviewed in PMID: 41079645).

To address this reviewer's question, we therefore decided to generate an *ad hoc* experimental system, following the most recent literature on the topic.

This system relies on the following cellular tools that we have generated:

a) A novel background cell line where UFMylated substrates accumulate:

JEG-3 UFSP2/UFM1 double knock-out cells (complete KO for UFSP2, the recycling enzyme of the UFMylation pathway; and displaying strongly reduced expression of endogenous UFM1).

b) Reconstituted cell lines expressing UFM1 variants which can or cannot be conjugated to target proteins:

- JEG-3 UFSP2/UFM1 KO cells + **FLAG-UFM1 Δ SC** (FLAG-tagged UFM1 mutant which can be conjugated to target substrates)
- JEG-3 UFSP2/UFM1 KO cells + **FLAG-UFM1 Δ GSC** (FLAG-tagged UFM1 mutant which cannot be conjugated to target substrate proteins as it lacks an additional c-terminal essential Gly residue).

This approach has been successfully employed to identify some of the known *bona fide* UFMylated cellular proteins (PMID: 39085203), as it allows to enrich for putatively UFMylated proteins by differential interactome analysis (only UFMylated proteins should be enriched in UFM1 Δ SC pull-downs when compared to UFM1 Δ GSC) and additionally allows to obtain direct biochemical evidence of UFMylation (UFMylated proteins carry a prototypic +156.08 Da shift on target lysines as a result of a glycine-valine dipeptide being covalently bound to lysine residues in tryptic peptides) (Fig. R2a).

We validated complete ablation of UFSP2 and significant reduction of UFM1 expression in our "founder" cell line (Fig. R2b), and confirmed robust and comparable expression of different UFM1 variants and their ability or inability of being conjugated to target proteins (see UFC1-UFM1 conjugates, Fig. R2c). In agreement with previous reports, using this approach we could obtain biochemical evidence of UFMylation on target lysines for abundant substrates, identifying modified peptides within some of the known abundant cellular proteins previously reported as UFMylated (RPL26, RPL26L, CYB5R3) as well as 3 novel cellular substrates (TRBV18, LARS, FLJ36925). As expected, these substrates were all exclusively enriched in UFM1 Δ SC when compared to UFM1 Δ GSC, confirming the validity of our approach. However, none of the identified modified peptides was significantly altered upon ZIKV infection (Fig. R2d).

To increase further our chances to map UFMylated proteins, we additionally performed differential analysis of UFM1 Δ SC-enriched proteins (i.e. putative UFMylated targets) when compared to UFM1 Δ GSC in both mock- or ZIKV-infected cells (Fig. R2e,f). This approach confirmed significant enrichment of other known UFMylation substrates (i.e. VCP, PMID: PMID: 38762759; MAVS, PMID: PMID: 37311461), alongside multiple UFMylation pathway members. In agreement with other experiments presented in our manuscript, none of the other most abundant UFMylation targets or UFMylation pathway members, was differentially modulated in ZIKV infection, corroborating the lack of general remodeling of UFMylation in viral infection (Fig. R2f). Interestingly, systematic analysis of selective UFM1 Δ SC-enriched proteins in mock or ZIKV-infected cells (Fig. R2g), and further manual inspection of most-differentially enriched proteins (Fig. R3h), revealed a handful of so far unknown cellular targets putatively hyper-UFMylated (ZCCHC9, C8Orf33, DDX54, ASF1B, KMT2A) or hypo-UFMylated (DNAJB6) only in viral-infected cells.

Furthermore, a few of the viral proteins (Capsid, NS5, prM) also appeared slightly enriched in UFM1 Δ SC when compared to UFM1 Δ GSC in ZIKV-infected cells, albeit all below the significance threshold (Fig. R2f).

Fig R2 – Systematic characterization of differentially UFMylated proteins upon viral infection. **a.** JEG-3 double KO cells (UFSP2/UFM1) were reconstituted with UFM variants which can (UFM1 Δ ASC) or cannot (UFM1 Δ GSC) be conjugated to target proteins. As such, differential interactome analysis can differentiate potentially ufmylated substrates (proteins only binding to UFM1 Δ ASC) from UFM1-interacting proteins (proteins also binding to UFM1 Δ GSC). **b.** Characterization of founder cell line (UFSP2/UFM1 KO) and **c.** reconstituted variants. **d.** Identification of UFMylated peptides in naive or ZIKV-infected JEG-3 cells. Only the most abundant known UFMylated proteins have been identified. **e.** Analysis of selective UFM1 Δ ASC enriched proteins in naive JEG-3 cells confirms identification of known UFMylated proteins and pathway members, which are not lost upon ZIKV-infection (f.; MOI=0.1, 48h.p.i.). Furthermore, mild (non-significant) enrichment of some of the viral proteins is observed. **f.;** MOI=0.1, 48h.p.i.). Furthermore, mild (non-significant) enrichment of some of the viral proteins is observed. **g.** Differential analysis of putative UFMylated substrates, and subsequent curation of prioritized host factors (h.) identifies putatively hyper- or hypo-UFMylated host proteins in ZIKV-infected cells.

Collectively, these results corroborate the lack of profound or generalized activation/inhibition of UFMylation in ZIKV-infected cells, and point towards a subset of putative cellular factors which could be differentially UFMylated in viral infection.

Although this was an experimental tour-de-force, we feel that the conclusions drawn from these data are still premature as: a) we cannot unequivocally rule out that some of the viral proteins might be UFMylated and b) the host factors identified as putatively hyper- or hypo-UFMylated in virus-infected cells would require extensive functional and biochemical validation. Additional follow-up experiments, using even more sensitive detection methods, such as alternative MS analysis pipelines and further functional validations would be needed to unequivocally ascertain the identity and functional relevance of identified substrates and challenge the UFMylation of viral proteins.

In light of this, we are in favor not to include the new UFMylation data in this manuscript as albeit extremely interesting, they would require further extensive validation and mechanistic experiments which are in our view beyond the scope of this study which focuses on the orthoflavivirus NS4B-interactomic landscape and already presents a large body of work.

The authors propose that UFMylation enhances mitochondrial respiration to support viral replication, but this hypothesis requires more direct experimental validation through alternative approaches to modulate respiration. Additionally, if mitochondrial function is the key mechanism, the authors should explain why NS4B specifically needs to interact with UBA5 and whether this interaction occurs at mitochondrial membranes.

This is an important question and we agree that the causal relation between the observed antiviral phenotype and the effects of UFMylation inhibition of viral replication is currently hypothetical and would require direct experimental validation.

With respect to the interaction of UBA5-NS4B on mitochondrial membranes, we would like to point out that as already shown in previous work, NS4B is a transmembrane protein which distributes exclusively in ER-derived replication factories, and not within mitochondria (PMID: 27545046; PMID: 33432690). For the benefit of the reviewer, we are including at the bottom of this paragraph, a snapshot of published confocal microscopy data from each of these studies, clearly demonstrating that in both ZIKV and DENV-infected cells NS4B does not colocalize with mitochondria.

Indeed, in the manuscript we do not propose that the NS4B-UBA5 interaction occurs on mitochondria membranes, but conversely exclude that changes in mitochondria morphologies are directly connected to the pro-viral phenotypes (lines 419-421).

The novelty aspect of our manuscript lies in the observation that genetic or pharmacological inhibition of UFMylation has a reciprocal effect on respiration to what observed in ZIKV infection. While we find this observation highly interesting, at the current stage we can only speculate about its functional consequences for infection since it's extremely difficult to experimentally disentangle the effects of NS4B expression (and UBA5 interaction) in ZIKV-infected cells upon UFMylation inhibition. Indeed, in cells deficient of a functional UFMylation pathway (UBA5 KO or inhibitor-treated cells) there is a substantial reduction in viral replication, which in turn would reduce virus-mediated alterations of mitochondrial respiration.

For this reason in the original version of the manuscript, we assessed the two effects separately (UBA5 inhibition or ZIKV infection), and described these reciprocal effects only as a potential link, not a direct evidence that mitochondrial morphology changes are the primary drivers of viral replication inhibition (carefully discussed in lines 559-564).

Nonetheless, following the reviewer's suggestion, we now performed additional experiments, aiming to disentangle the causal relation between NS4B-UBA5 interaction and the effects on mitochondrial metabolism. To this end, we assessed the impact of ectopic ZIKV-NS4B expression on mitochondrial respiration in JEG-3 cells and evaluated the impact of UFMylation inhibition on respiration under these conditions.

In agreement with the previous experiments described in the manuscript, inhibition of UFMylation strongly reduced basal, maximal and ATP-linked respiration (Fig R3; red columns); however ectopic expression of ZIKV NS4B had no significant effects under these conditions regardless of the treatment of the cells. To rule out potential effects results deriving from kinetics difference, we also tested shorter NS4B expression regimens (two days instead of three days) but observed very similar phenotypes (data not shown).

Figure R3 – Effect of ectopic ZIKV-NS4B expression on cellular respiration.

Naïve JEG-3 cells were transduced with lentiviruses expressing ZIKV-NS4B (NS4B-Z) or empty controls (pWPI) (MOI=5) and after short puromycin selection, cells were seeded on Seahorse plates and treated with DMSO or UBA5 inhibitor (DKM2-39; 32 µM). Twenty-four hours after treatment (72h.p.t.) metabolic activity was measured at the Seahorse. Results from n=3 independent experiments are shown.

We hypothesize that the expression levels, the exact timing or the impact of additional cellular or viral proteins as well as remodeled ER (whose contacts with mitochondria are altered upon viral infection; PMID: 39834870), might play an important role in these phenomena and contribute to UFMylation-dependent changes in virus respiration. However, we cannot experimentally address these hypotheses. Indeed, it still possible that the UBA5-NS4B interaction requires infection-specific changes to modulate respiration (i.e. other viral or

cellular proteins) or that is needed to increase local UFMylation of an additional viral or cellular protein which ultimately modulates respiration. Noteworthy, while DENV NS4B was shown to stimulate mitochondrial elongation, the identity of the viral protein actually responsible for changes in mitochondrial respiration was never determined in any of the previous studies and is still elusive (PMID: 39834870, PMID: 33245857).

Additional experiments, would require systematic testing of all the individual viral proteins, alongside the use of expression systems independent from active viral replication which could allow to discriminate the impact of UFMylation inhibition on viral replication from that on respiration.

Following the recommendation of Reviewer#1 and alongside our initial cautious conclusions, we have further tuned down our statements in the corresponding sections (lines 559-564); but we are determined to follow up these reciprocal effects through extensive additional experimentation in follow-up studies.

3. The findings would be significantly strengthened by validation in multiple cell types beyond JEG3 cells, particularly given the potential for cell-type-specific UFMylation interactions. The restriction to a single cell type limits the generalizability of the conclusions and raises questions about the broader relevance of the UFMylation-flavivirus interaction.

As extensively discussed above (see point 1), multiple lines of evidence allow us to unequivocally rule out that these observations are potential artefacts or that conclusions cannot be generalized. Nonetheless, following this reviewer's suggestion, we now included an additional primary, non-immortalized epithelial cell line (MRC-5) confirming significant antiviral activity, in line of observations already presented in the manuscript (Fig. R4).

Fig. R4 – DKM2-93 exhibit antiviral activity in MRC-5 cells. A reduction in ZIKV titers was also observed in MRC-5 cells. MRC-5 cells were infected with ZIKV H/PF/2013 wild-type strain (MOI 0.25), and treated with either 32 μM of DKM 2-93 or equivalent concentration of DMSO. Supernatants were harvested at 72 hpi and the virus titers were measured by plaque assay. n=3 biological replicates; Panel c, bars represent the mean and error bars represent the standard deviation of the mean. **p<0.01; ns, not significant as determined by two-way ANOVA with Sidak's multiple comparisons test.

These results are now included in the manuscript as well (new Supplementary Fig. 6b), extending the validation of UFMylation relevance to hepatic (Huh7 cells), trophoblasts (JEG-3), epithelial (MRC-5) cells, alongside a living organism (*Danio rerio/Zebrafish*).

Specific comments:

Figure 1 and 2: Key validation data are missing regarding NS4B expression levels and subcellular localization. The authors should show that expressed NS4B achieves physiologically relevant expression levels comparable to those observed during infection and maintains proper subcellular distribution. The resolution of Supplementary Figure 1 is insufficient to assess localization patterns. Additionally, the temporal dynamics of the NS4B-UBA5 interaction during the viral lifecycle are not characterized.

We now provide additional characterization of NS4B expression levels in ZIKV-infected or NS4B-expressing cells; alongside detailed subcellular distribution of all cell lines used in the study including cellular markers for the endoplasmic reticulum (Fig. R5). These data confirm that expression levels of ZIKV-NS4B stable JEG-3 cell lines are comparable to those observed in JEG-3 cells infected with ZIKV at 24 h.p.i. (MOI=1) (Fig R5a) and that all our cell lines express NS4B at the endoplasmic reticulum (Fig. R5b). These data have now been included in updated Supplementary Fig. 1.

Fig. R5 - The expression levels of NS4B in ZIKV infected JEG-3 cells and JEG3 cells stably expressing 2k-NS4B^{HA} and subcellular distribution of NS4B-expressing cell lines.

a. Wildtype JEG-3 cells were infected with ZIKV H/PF/2013 (MOI 1) wildtype strain, and the cell lysates were harvested in RIPA buffer 24 hpi. The JEG3 cells stably expressing 2k-NS4B^{HA} were plated alongside the wildtype cells, and the lysates harvested together. Western blot analysis confirms the comparable expression levels of NS4B in both the cells. **b.** Subcellular distribution of all JEG-3 stable cells lines used in the study. HA and Calnexin were used to assess localization at the ER using confocal microscopy. Scale bar=10 μm.

Figure 4: In Figure 4a-c, the extent of both the reduction and the restoration of viral titers are very modest. The level of NS2B in the UBA5-depleted samples show minimal reduction in NS2B levels and that does not change in the reconstituted samples. Similarly, the infectious virus production is between 2-4 fold and rescue seems to be partial. The authors should move this to the supplementary section, since the KO data (Figure 4h) seems more convincing. What is also puzzling is that while reconstitution of the WT UBA5 in the depleted (shRNA) and deleted (gRNA) background seem equivalent (Figure 4f), the viral titres are rescued in the gRNA background (Figure 4h) but not in the shRNA background (Figure 4b). What is the explanation for this?

We were also puzzled at first by the magnitude of the rescue effects, and as also discussed in response to reviewer's 1 comment above, we think that these differences could be due to the presence of leftover UBA5 in shUBA5-expressing cell lines (Fig 4f: under shRNA-mediated silencing, there is still residual UBA5 activity, as inferred by the presence of UFC1-UFM1 conjugates by WB). Under these conditions (residual UBA5 + UBA5 overexpression) there is likely an excessive amount of functional UBA5 (Fig. 4a, last lane), leading to a misbalance in the UFMylation pathway. Although seemingly counterintuitive, UBA5 overexpression has indeed been reported to mimic its ablation, reducing the levels of charged UFC1 in the cell (PMID: 35806453).

In agreement with this hypothesis, also in our hands UBA5 knock-out cells (completely devoid of residual UBA5 or UFC1-UFM1 conjugates) display a much stronger viral phenotype and an overall stronger magnitude of rescue upon overexpression of UBA5 (Fig. 4h). We decided to leave both assays in the manuscript as they still provide data complementary to the KO.

The authors should determine which step in the viral lifecycle is affected by systematically measuring entry, replication and release in the KO and reconstituted cells.

We genuinely do not feel that this experiment would add any additional information to the manuscript as we already demonstrate this point using:

- pharmacological inhibition (with an inhibitor which shows antiviral activity in vitro and in vivo) on time-of-addition studies
- a very clean genetic system allowing as to study viral RNA translation and RNA replication, independent of viral entry and particle production. This is already the best possible system where we can unequivocally rule-out effects on viral RNA translation or viral RNA replication in the absence of viral entry or viral spread.

Furthermore, we note that:

- In our hands, electroporation of JEG-3 cells is associated with significant cytotoxicity and therefore we would need to rely on time of addition experiments or very high MOIs
- Pharmacological inhibition allows for better-controlled experiments as can be added at specific times (while in KO multiple cycles of viral replication would influence the readout, making it difficult to dissect effects on specific replication step(s)).
- Independent studies published during the revision of this paper, and now also referenced in the revised text, also clearly point to a post-entry/late stage effect on replication for ZIKV virus (PMID: 40459261; now discussed in the manuscript).

Collectively, the data from us presented in the study, alongside other independent studies published during the revision of this manuscript, fully substantiate the manuscript conclusions.

Figure 5: The imaging data (panel c) lacks sufficient resolution for meaningful colocalization analysis. The authors to provide higher resolution data and quantify colocalization between UFMylation components and viral proteins. It would also be useful to see whether this is happening at mitochondrial membranes, given the effect of UFMylation on increased mitochondrial respiration.

We apologize for the low resolution of the uploaded figures, likely resulting from the automatic .pdf conversion. We now re-uploaded high-quality .tiff files (also available as separate files and retrievable from the online portal) from the confocal microscopy pictures already presented.

We feel that these data, are of the highest resolution and quality, and together with the other experiments presented in the manuscript (co-IP-WB, interaction with other components of the UFMylation machinery by mass-spectrometry, enrichment of NS2B-3 in UFM1 pull-down by co-IP WB) fully support the original statements. We note that we do not propose or suggest these events to occur on mitochondrial membranes (see point 2 above). As we have discussed above and reported before, NS4B is an integral ER protein and does not associate with mitochondrial membranes. Conversely, we clearly propose in the manuscript that effects on mitochondria morphology are likely not connected to the observed changes in viral respiration (discussed in point 2 above).

Figure 6: While the effects of the chemical inhibitor on viral proteins and infectious particles are clear (panels 6c-d), the time-of-addition experiments show much more modest effects (<2-fold reduction compared to DMSO control). What is the explanation for this discrepancy?

We think that these milder effects might be due to the presence of 1 outlier experiment (upper repeat across all conditions), as well as the excessive media change unique to this experiment compared to the others (across all conditions media is changed 4 times, to ensure comparability with the most "treated" condition, *i.e.* the "pre-" condition, were media is changed before virus infection, upon virus inoculation, directly after virus-absorption, and again 3 hours post-infection). Following also another reviewer's recommendation, to consolidate this experiment further we have now performed 2 additional biological repetitions of this experiment (included in new Fig. 6e), which fully recapitulate the previous experiments and show even stronger and significant effects.

Figure 7: The analysis of Figure 7a should be performed by co-staining for NS4B and UFMylation components to directly visualize their spatial relationship in the context of the proposed mechanism.

As mentioned above, we did not perform this experiment, as drug-treatment, KO or knock-down of UBA5 would inhibit viral replication and thus viral protein production. Hence, we would expect reduced NS4B staining which would complicate the interpretation of the analysis. Furthermore, as discussed above and in the manuscript, we clearly state that we do not consider effects on mitochondrial network morphology to be responsible for viral phenotypes

(lines 415-417) also in light of previous literature (PMID: 27545046), or expect the UBA5-NS4B interaction to take place on mitochondria. These data are provided to the community, as descriptive of an impact of UBA5 depletion on mitochondrial network morphology. This could be of potential relevance for other UFMylation-regulated processes. We believe instead, that the effects on mitochondrial respiration (Fig. 7h-o) might contribute to flavivirus replication, through mechanisms we still do not fully understand.

Reviewer #4 (Remarks to the Author):

This manuscript provides a comprehensive analysis of cellular proteins with viral NS4B of multiple orthoflaviviruses and incorporates information from the host factors that are perturbed during orthoflaviviral infection. They focus on the role of UBA5 and the UFMylation pathway in promoting infection, with emphasis on ZIKV, in in vitro and in vivo model systems, using genetic ablation as well as a pharmacological inhibitor. In addition, they also describe a role for UFMylation in regulating mitochondrial morphodynamics that they postulate could potentially be hijacked or mediated by viral proteins during infection. Overall, the study identifies a new role for a post-translation modification pathway during orthoflavivirus infection that could serve as a potential target for the design of pan-orthoflaviviral drugs.

The methods are well-described and the supplementary tables cover required information. While the manuscript provides sufficient evidence for what the current title describes, there are a few suggestions/ queries that, if addressed, would greatly benefit the work overall.

We thank the reviewer for their supportive comments.

Major comments:

1. As the authors might already be aware, a recent article published in JVI addresses a very similar question (<https://doi.org/10.1128/jvi.00654-25>). While both manuscripts focus on the UFMylation pathway, they differ in their approaches and breadth and also their targets. The JVI manuscript would need to be cited here and this manuscript would also benefit from additional discussion from the author's perspectives on a parallel UFMylation mechanism in orthoflaviviruses.

We thank the reviewer for this comment, indeed shortly after first submission of this manuscript, another study reported that silencing of other UFMylation components affect orthoflavivirus replication. We now discuss and refer to this study in the revised manuscript (lines 492-502 and 574-576).

2. Readability – the abstract and introduction are concise and well written. However, the first and second results sections require constant back and forth, especially between Figures 1b and 2a, affecting the readability of the manuscript. The first part of the results section could also benefit from some language editing to make it easier for readers to comprehend.

We thank the reviewer for this comment. We have now rephrased large parts of this section to improve readability. Changes are indicated in the tracked-change version online.

3. The transition from Figure 2 to Figure 3 results is a big jump, going from a pan-orthoviral proteomics study to performing functional follow-up only on ZIKV. What was the reason to choose ZIKV as the virus of choice for the mitochondrial studies? While a narrower scope for a RNAi screen is understandable on a technical level, the manuscript should give a rationale for the choice of virus for functional follow-up. Additionally, the figure and section title should more accurately describe the scope of the work in this section; as written, it reads like an over-statement of the presented work. This is also true for the discussion of the RNAi results in the Discussion.

We have now added additional justification explaining the rationale behind the use of ZIKV as follow-up virus. This was mostly due to the selection criteria used to prioritize hits from the MS data. In consideration of the cellular background used (JEG-3 are choriocarcinoma derived, and thereby of placental origin), we included more hits related to ZIKV which is one of the few orthoflaviviruses vertically transmitted. Follow-up studies were dictated by the results of the the observation that UBA5-NS4B was amongst the validated interactions in infected cells and one of the strongest pro-viral factor newly identified by the RNAi screen. We now comment on

these choices more specifically (lines 249-251). We also revised figure and title sections to reflect follow-up on ZIKV (line 245 and line 271).

4. Authors haven't shown that the UBA5 inhibitor used in the study, DKM 2-93, successfully inhibits UFMylation in their experiments. Specificity of the inhibitor in affecting UFMylation downstream by lack of formation of UFM-conjugates would need to be demonstrated in both in vitro and in vivo experiments.

We thank the reviewer for this comment. We had characterized this during the study, but since the compound was published in the context of other studies and in light of the large number of figures, we had not included this in the original manuscript.

As seen below (Fig. R6), treatment of JEG-3 cells with increasing doses of DKM 2-93, significantly and dose-dependently reduces the abundance of higher molecular weight UFM1-conjugates (most visible at 32 and 64uM).

Figure R6 - Dose-response curve of DKM 2-93 inhibition of UFMylation in JEG-3 cells.

JEG-3 cells were treated with increasing concentrations of the DKM 2-93 inhibitor for 72h. Western blot analysis confirmed the inhibition of the UFMylation pathway as indicated by the absence or decreased intensity of UFM1 conjugates at higher concentrations of the inhibitor (multiple species between the 30 and 60 kDa bands).

As reference, we also included cells persistently silenced for UFMylation (shUBA5). Basal UFM1 levels are shown as loading control. We have now included this figure in Supplementary Fig. 6a.

5. Are there other E1 enzymes known for UFMylation in humans? The UBA5 inhibitor used is effective in micromolar concentrations which is higher than what is conventionally considered 'low'. The authors do show that viability isn't affected but what could be the reason for such high amounts of inhibitor required to show an antiviral effect?

We completely agree that the EC50 of this inhibitor is not low, also in the original paper where it was described for cancer treatment the Selective Index (SI) was rather modest. Indeed, we and others are currently developing improved derivatives of the compound (see very recent publication from September 2025: PMID: 40994290).

We believe that the low potency of the compound might be simply due to its inherent activity, rather than redundancy of the target (to our knowledge, no other E1 enzyme is known in mammals). This conclusion stems from the *in vitro* data of the original publication showing high selectivity (PMID: 28186401), but also the very high consistency of effects observed in cells in our hands: virus replication phenotypes, effects on mitochondria morphology and mitochondria respiration fully recapitulates those observed upon genetic inhibition of the pathway, further arguing for a high selectivity of this compound.

6. While the authors have shown that pharmacological inhibition of UBA5 restricts ZIKV replication in zebrafish, demonstrating that genetic ablation or UBA5 knockdown has the same effect would further substantiate their in vivo results.

The suggestion of the reviewer is interesting but it's associated with a high technical challenge. First, generating a UBA5 KO zebrafish line is not feasible as the animals would not survive past 40 days post-fertilization (PMID: 38046095). F₀ knockout of UBA5 (by injection of the embryo with Cas9/gRNA complexes) is possible, but the animals would develop very rapidly peripheral nerve and cerebellar axonal damage, resulting in impairment in motor function and delayed growth. Under these conditions, the phenotypes observed upon ZIKV infection could be pleiotropic and not directly related to the loss of UBA5-NS4B interaction *in vivo*. Finally, such an approach would require the co-injection of ZIKV particles and gRNA-Cas9 complexes at the 1-4 cell embryonic stage. Beyond a possible reduced efficiency of viral infection and KO, this would imply that ZIKV starts replicating while the KO is not yet established.

Taking into account all these factors, we believe that the pharmacological approach we used in the manuscript was the best suited to address the role of UBA5 in ZIKV pathogenesis *in vivo*, especially considering that we always obtained comparable phenotypes between the UBA5 KO and DKM treatments in all our human cell-based experiments. That said, we want to emphasize that the UBA5-dependent UFMylation pathway is functional in zebrafish (PMID: 38046095) and was shown to impact mitochondria functions, in line with our observations in human cells. We have now included these considerations in the discussion.

Minor comments:

1. Incorrect figure cited in Line 130 (should be Supplementary Figure 1b).

Changed accordingly.

2. Looks like incorrect virus may be mentioned in Line 214, (JEV instead of YFV?) based on Figure 2g.

Changed accordingly.

3. Authors use the term 'replication' to denote RNA copies as well as viral titers interchangeably in the y-axes of figures and also in the text (for example – Figure 4c, g, h v/s Figure 5f). Maintaining consistency throughout the manuscript will prevent confusion.

Changed accordingly.

4. The previous point is especially important in the title of Figure 6 – the title reads "Pharmacological inhibition of the UFMylation pathway impairs ZIKV replication" while 6f and 6g clearly indicate that the viral replication process is not hindered by DKM 2-93 treatment. The results section for this figure explains this clearly. Please modify this figure title.

Changed accordingly.

5. Text referring to Figure 3c and Supplementary Figure 3a mentions '5 pro-viral genes' throughout the manuscript (Lines 247, 470) while the figure has 6 genes indicated. Kindly clarify.

Thank you for noticing this mistake. The figure has been changed accordingly, it indeed contained one additionally labeled gene.

6. Figure 1b –only the high confidence interactors have been denoted. While this could be a stylistic choice by the authors to emphasize the specific interactions, having all the interactions shown here would make the figure a lot more appealing (the empty circles, especially prominent in the "Mitochondria" bubble, don't convey much information and therefore don't substantially help the figure). Additionally, this Figure has more than 200 interactors labelled and the figure legend indicates only high confidence interactors have been denoted. Kindly clarify the discrepancy in numbers.

All the nodes displayed in the figure (labelled or unlabeled) represent significant, high-confidence interactors. The choice of displaying only some of the labels was made to improve readability. Following this reviewer's suggestion, we have now labelled all the displayed nodes, and rearranged them within the network to improve readability. The apparent discrepancy in number results from the decision to display in Fig.1b also those connections

below the hit-calling criteria if a certain interactor was highly significant for one species and less enriched - but still significant - for others (explained in more details in lines 1366-1371). We note that instead, in the supplementary tables (Supplementary Table 1) we present both high-confidence and extended interactors, labelled accordingly and fully searchable using different identifiers (GeneID, UniprotID). This is also accompanied by a full summary of specificity across species and hopefully allows the reader to search better her/his desired node.

7. Figure 2a and Line 189 indicate different numbers of interactors.

Please note that the n=76 intersection refers to the shared interactors between YFV, WNV and ZIKV (not the n=74 pan-viral baits). We rephrased the sentence to clarify this point.

8. Figure 4b – shUBA5 v/s exogenous supplementation does not show significant difference in viral RNA levels, while Figure 4c shows successful rescue in viral titers. Can the authors comment on this?

We think that these differences could be due to the milder effects observed on viral RNA replication when compared to effects on infectious particle production (in agreement with the proposed mechanism). Furthermore, as discussed in response to Reviewer #1's comment, milder effects in shRNA-based experiments might be due to the presence of leftover UBA5 in shUBA5-expressing cell lines (Fig 4f: under shRNA-mediated silencing there is still residual UBA5 activity, as inferred by the presence of UFC1-UFM1 conjugates by WB). Under these conditions (residual endogenous UBA5 + UBA5-FLAG overexpression) there is likely an excessive amount of functional UBA5 (Fig. 4a, last lane), leading to a misbalance in the UFMylation pathway. Although seemingly counterintuitive, UBA5 overexpression has been reported to mimic its ablation, reducing the levels of charged UFC1 in the cell (PMID: 35806453).

In agreement with this hypothesis, UBA5 knock-out cells displayed a much stronger viral phenotype (no residual UBA5 or UFC1-UFM1 conjugates) and an overall stronger magnitude of rescue upon overexpression of UBA5 (Fig. 4h).

9. Can the authors provide a better rationale in the manuscript how they chose UBA5 from the proteomics dataset? The extended NS4B effectome and the extended NS4Bome indicate that UBA5 was only picked up in ZIKV samples (Supp Tables 1 and 2, and Figure 1b).

As part of the strategy outlined in Fig. 3a, we included more ZIKV unique interactors in our hit selection (ZIKV was given a higher priority as it was the most relevant virus for the cellular background used; JEG-3 trophoblast-like cells). Indeed, the reason to further pursue UBA5 was the a) exclusive binding profile to ZIKV-NS4B (Fig. 1b); b) the validated binding to NS4B also in ZIKV-infected cells (Fig. 3e) and c) it exhibited one of the strongest pro-viral phenotypes in the RNAi screen performed with ZIKV (Fig. 3e). We also had other candidates matching these criteria, but because we were intrigued by its function in a poorly characterized post-translational modification, we eventually prioritized it for follow-up studies. We now included a few sentences to clarify further this choice (line 249-251 and 274-278).

Reviewer #5 (Remarks to the Author):

Point-by-point response to reviewers' comments

Manuscript Number: **NCOMMS-25-34251A**

Re.: Rebuttal to reviewer's comments for

"A genus-wide interaction atlas across NS4B orthologues identifies a conserved role for UFMylation in orthoflavivirus replication" by Rajasekharan S. et al.

We thank the 5 reviewers for their constructive feedback and suggestions that helped to further improve the manuscript. Please find below a point-by-point reply to their remaining comments.

Please find uploaded in the online submission system the merged document highlighting the changes to the second version of the manuscript. Point-by-point responses to reviewers' comments are indicated below in dark blue.

Reviewer #1 (Remarks to the Author):

The authors have satisfactorily addressed my concerns.

We thank the reviewer for their supportive comments.

Reviewer #2 (Remarks to the Author):

We thank the reviewer for their supportive comments.

Reviewer #3 (Remarks to the Author):

The revised version has largely addressed the concerns raised. What is still missing from the study is that the defect in the late stage of the viral lifecycle is inferred. Assembly/release/spread is not directly measured by intracellular vs extracellular infectious units, particle:PFU ratio, intracellular E/prM accumulation with reduced release etc. We thank the reviewer for their supportive comments. We agree that these experiments would be of great interest, and we are currently addressing them in the laboratory. We note that while we do not provide any experimental evidence allowing us to dissect whether effects are on assembly, release or specific infectivity, we do provide genetic and pharmacological evidence that the defect is at late stage. Namely, the reduction on infectious particles (PFU/ml) observed upon drug-treatment (as well as KD and KO), alongside the lack of effect on viral entry (time of addition experiment), viral RNA translation or viral RNA replication (sub-genomic replicon experiment), strongly support this claim. We intentionally refer to this defect as "infectious particle production", which would be reduced as a result of defects on either assembly, release or specific infectivity.

We further toned down our statements to clarify this aspect in the results (lines 371-373): "Collectively, these results indicate that functional UFMylation is critically required for late stages of the viral replication cycle, independent from viral entry, viral RNA translation and viral RNA replication."

Additionally we now underlined the necessity for further studies in this direction in the discussion (lines 580-584): "Arguably, in agreement with a recent report (Schmidt et al., 2025), these effects seem particularly relevant at late stages of infection, as they affect infectious particle production, independently of viral entry, viral RNA translation and viral RNA replication (Fig. 6c-g). More work is needed to dissect whether assembly, trafficking or specific infectivity of virus particles is affected.")

Fig 6a legend describes DKM 2-93 as “competitively” binding the catalytic cysteine. DKM 2-93 is described in the primary literature to covalently modify the catalytic cysteine of UBA5 (Cys250). That’s not the same thing.

We thank the reviewer for noticing this inaccuracy. We have now modified the sentence in the figure legends accordingly (“... covalently modifies...” instead of “...covalently binds...”).

Reviewer #4 (Remarks to the Author):

The authors have largely amended their manuscript to address the reviewer’s comments. However, there are a few areas that can still be improved.

We thank the reviewer for their supportive comments.

1. Figure 1b is now much more interpretable and interesting to go through. This change is great.

We thank the reviewer for their supportive comments.

2. While the authors have addressed the issue of readability by combining results sections 1 and 2 that makes it easier, they might want to consider rearranging their figures to make it easier for readers.

We removed the subheading to facilitate interpretation of the first paragraphs. We hope it will be of sufficient clarity to allow readers to follow the results narrative.

3. The justification for doing follow up studies with ZIKV is still weak. The use of JEG-3 for their APMS study is then confusing as this background is not ideal for 4/8 viruses chosen by the authors. While the experiments that follow make sense, how they address choosing ZIKV is still not convincing.

We made our best to clarify our rationale within the text, but we note that the emphasis on ZIKV as a follow up model virus species, was indeed driven by the cellular background used (using all the other viruses as controls in such “not ideal” background was an intentional choice).

4. The authors have provided western blotting data to confirm inhibition of UFMylation pathway. This definitely helps but the use of 16uM for their experiments is tricky as it doesn’t seem to inhibit the pathway as effectively as 32uM, but it does inhibit viral protein synthesis. We agree that the inhibition of UFMylation might appear weak by WB, however we note that this readout is unavoidably insensitive as it allows inferring global UFMylation only indirectly, through the most abundant species (RPL26 and UFM1-UFC1 or UFM1-UBA5 conjugates) heavily UFMylated and thereby detectable by WB. For this reason, we included also the shUBA5-treated cells as control. Also in this case the extent of inhibition might appear weak, and higher molecular weight UFM-reactive species can still be detected. While we cannot provide an exact quantitative measure for the overall degree of UFMylation inhibition using this approach, the dose-dependent inhibition of UFM-reactive species observable at 16-64 uM range supports the specific inhibition of the pathway. Any toxicity on protein synthesis at 16-32 uM can be ruled out from data on cell viability *in vitro*, as well as *in vivo* data (using 32uM) where no overt toxic effects on the animals could be observed. Collectively, these data together with the observations that DKM2-93 phenocopies KD and KO data with respect to viral phenotypes, effect on mitochondrial morphology and effect on cellular respiration supports the notion that the pathway inhibition is specific, dose-dependent and non-cytotoxic at the concentrations 16-32 uM.

5. The authors have explained the difficulties with generating a UBA5 KO zebrafish model, but they say they have included these considerations in the discussion, which doesn’t seem to be the case.

We apologize for overlooking this addition, we were actually referring to the part of the text related to the mitochondrial abnormalities and involvement of central nervous system. We have now included in the text a succinct version of the explanations provided in the reviewers' response at the end of the discussion. *"... Interestingly, while the use of UBA5 ^{-/-} lines for further mechanistic experiments is associated with technical challenges (it would require the co-injection of ZIKV particles and gRNA-Cas9 complexes at the 1-4 cell embryonic stage), UBA5 deficiency in zebrafish was shown to impact mitochondria functions and affect both peripheral and central nervous systems development (PMID: 38046095). Improved chemotypes or alternative strategies selectively targeting this pathway might serve as highly potent broad-spectrum antiviral inhibitors and provide valuable model systems to characterize the relevance of UFMylation in ZIKV pathogenesis."*

6. The revised manuscript largely addresses the queries of all reviewers. However, some justifications still seem like overinterpretations. I would like to ask the authors to revise the manuscript to make sure not exaggerate some of their findings.

We have made further amendments to some of the conclusions from the most heavily discussed points during the revision (i.e. mitochondrial respiration, effect on assembly/release and potency of the antiviral effect).

Reviewer #5 (Remarks to the Author):

I co-reviewed this manuscript with one of the reviewers who provided the listed reports. This is part of the Nature Communications initiative to facilitate training in peer review and to provide appropriate recognition for Early Career Researchers who co-review manuscripts. We thank the reviewer for their supportive comments.